# Local and systemic responses to SARS-CoV-2 infection in children and adults

Masahiro Yoshida[1,2,16], Kaylee B. Worlock[1,16], Ni Huang[3,16], Rik G. H. Lindeboom[3,15], Colin R. Butler[4,5], Natsuhiko Kumasaka[3], Cecilia Dominguez Conde[3], Lira Mamanova[3], Liam Bolt[3], Laura Richardson[3], Krzysztof Polanski[3], Elo Madissoon[3,6], Josephine L. Barnes[1], Jessica Allen-Hyttinen[1], Eliz Kilich[7], Brendan C. Jones[4,5], Angus de Wilton[7], Anna Wilbrey-Clark[3], Waradon Sungnak[3], J. Patrick Pett[3], Juliane Weller[3], Elena Prigmore[3], Henry Yung[1,7], Puja Mehta[1,7], Aarash Saleh[8], Anita Saigal[8], Vivian Chu[8], Jonathan M. Cohen[7], Clare Cane[8], Aikaterini Iordanidou[8], Soichi Shibuya[4], Ann-Kathrin Reuschl[9], Iván T. Herczeg[1], A. Christine Argento[10], Richard G. Wunderink[10], Sean B. Smith[10], Taylor A. Poor[10], Catherine A. Gao[10], Jane E. Dematte[10], NU SCRIPT Study Investigators*, Gary Reynolds[11], Muzlifah Haniffa[3,11], Georgina S. Bowyer[12], Matthew Coates[12,13], Menna R. Clatworthy[3,12], Fernando J. Calero-Nieto[14], Berthold Göttgens[14], Christopher O'Callaghan[4,5], Neil J. Sebire[4,5], Clare Jolly[9], Paolo De Coppi[4,5], Claire M. Smith[4], Alexander V. Misharin[10], Sam M. Janes[1,7], Sarah A. Teichmann[3,15], Marko Z. Nikolić[1,7,17✉] & Kerstin B. Meyer[3,17✉]

It is not fully understood why COVID-19 is typically milder in children[1–3]. Here, to examine the differences between children and adults in their response to SARS-CoV-2 infection, we analysed paediatric and adult patients with COVID-19 as well as healthy control individuals (total $n = 93$) using single-cell multi-omic profiling of matched nasal, tracheal, bronchial and blood samples. In the airways of healthy paediatric individuals, we observed cells that were already in an interferon-activated state, which after SARS-CoV-2 infection was further induced especially in airway immune cells. We postulate that higher paediatric innate interferon responses restrict viral replication and disease progression. The systemic response in children was characterized by increases in naive lymphocytes and a depletion of natural killer cells, whereas, in adults, cytotoxic T cells and interferon-stimulated subpopulations were significantly increased. We provide evidence that dendritic cells initiate interferon signalling in early infection, and identify epithelial cell states associated with COVID-19 and age. Our matching nasal and blood data show a strong interferon response in the airways with the induction of systemic interferon-stimulated populations, which were substantially reduced in paediatric patients. Together, we provide several mechanisms that explain the milder clinical syndrome observed in children.

SARS-CoV-2 infection in children presents with milder disease severity compared with infection in adults[1,2]. The overall risk of severe COVID-19 in children is even lower than originally believed[3], with around two deaths per million. The molecular basis of the differences in disease progression between children and adults is not understood and may hold clues for better treatment of severe SARS-CoV-2 infection.

SARS-CoV-2 uses a host cell-surface protein, angiotensin-converting enzyme 2 (ACE2), as a receptor for cellular entry[4]. Studies suggested that *ACE2* expression is both tissue and age dependent[5,6], with the highest expression found in nasal epithelium of healthy adults[7] and

comparatively lower expression in paediatric upper[8] and lower airways[6,9]. These differences were proposed to contribute to reduced disease severity in children, although recent studies have found no correlation with age or infection[10,11].

During the initial antiviral immune response, interferon (IFN) is important in inhibiting viral replication, contributing to both innate and cell-intrinsic immunity[12,13]. Severe COVID-19 in adults has been linked to an impaired antiviral response in the nasal epithelium and blood[14–16], whereas several other studies highlight the contribution of the IFN response to the pathogenesis[17,18].

[1]UCL Respiratory, Division of Medicine, University College London, London, UK. [2]Division of Respiratory Diseases, Department of Internal Medicine, Jikei University School of Medicine, Tokyo, Japan. [3]Wellcome Sanger Institute, Cambridge, UK. [4]NIHR Great Ormond Street BRC and UCL Institute of Child Health, London, UK. [5]Great Ormond Street Hospital for Children NHS Foundation Trust, London, UK. [6]European Molecular Biology Laboratory, European Bioinformatics Institute, Cambridge, UK. [7]University College London Hospitals NHS Foundation Trust, London, UK. [8]Royal Free Hospital NHS Foundation Trust, London, UK. [9]UCL Division of Infection and Immunity, University College London, London, UK. [10]Division of Pulmonary and Critical Care Medicine, Northwestern University Feinberg School of Medicine, Chicago, IL, USA. [11]Biosciences Institute, Newcastle University, Newcastle upon Tyne, UK. [12]Department of Medicine, University of Cambridge, Cambridge, UK. [13]Cambridge University Hospitals NHS Foundation Trust, Cambridge, UK. [14]Wellcome, MRC Cambridge Stem Cell Institute, University of Cambridge, Cambridge, UK. [15]Department of Physics, Cavendish Laboratory, University of Cambridge, Cambridge, UK. [16]These authors contributed equally: Masahiro Yoshida, Kaylee B. Worlock, Ni Huang, Rik G. H. Lindeboom. [17]These authors jointly supervised this work: Marko Z. Nikolić, Kerstin B. Meyer. *A list of authors and their affiliations appears at the end of the paper. ✉e-mail: m.nikolic@ucl.ac.uk; km16@sanger.ac.uk

As the virus spreads, 14% of symptomatic, unvaccinated adults develop progressive respiratory failure displaying a strong inflammatory immune response[19]. Single-cell analysis of this response in adults demonstrated the involvement of various immune cell types, including proinflammatory monocytes/macrophages[20], clonally expanded cytotoxic T cells[21–23] and neutrophils[21]. However, the cell-specific immune responses in children have not been comprehensively characterized. Studies comparing bulk RNA-sequencing (RNA-seq) and cytokine profiles between children and adults suggest a more robust immune response, such as increased levels of IFNγ and interleukin-17 (IL-17A) in the plasma[24], and a reduced antibody response and neutralizing activity against SARS-CoV-2 in children[25]. The most recent single-cell transcriptional study analysing the upper airways of children with mild COVID-19 revealed that higher expression of pattern recognition receptor pathways was related to a stronger innate immune response[11]. However, differences in the coordination of local and systemic immune responses to SARS-CoV-2 between children and adults including patients with severe COVID-19 remain to be elucidated.

To address these questions and identify paediatric-specific responses in COVID-19, we collected matched nasal, tracheal, bronchial and blood samples from healthy individuals and patients with COVID-19 from infancy to adulthood and analysed them using single-cell transcriptomics combined with protein profiling.

## Study cohort and experimental overview

Using single-cell RNA-seq and cellular indexing of transcriptomes and epitopes by sequencing (CITE-seq), we examined the effects of COVID-19 in children versus adults, comparing the airway and systemic responses. We recruited 19 paediatric and 18 adult patients with COVID-19, ranging from asymptomatic to severe, and 41 healthy children and adults, to profile the cellular landscape in the airways (nasal, tracheal and bronchial brushings) and in matching peripheral blood mononuclear cells (PBMCs) (Fig. 1a and Extended Data Fig. 1a, b). For 6 patients with COVID-19, blood was also taken at hospital discharge. Furthermore, 15 patients contributed nasal and/or blood samples 3 months after having severe COVID-19. A summary of patient characteristics and metadata is provided in Extended Data Table 1.

In total, we generated a dataset of 659,217 cells (an easy-to-use interactive analysis is provided at https://www.covid19cellatlas.org/). We characterized the epithelial and immune cell compartments at a high granularity, identifying 59 cell types and states in airways including previously undescribed ones (Fig. 1b, c and Extended Data Fig. 2a, b) and 34 cell types in blood, mostly based on established markers[23,26].

## New cell subtypes in airway epithelia

The detailed cell type annotation is described in the Supplementary Note, with marker genes and comparison to existing datasets in Extended Data Figs. 2c and 3a–d. Multiple basal, goblet, ciliated and transit epithelial 1 and 2 (secretory to ciliated) cell types reflect the plasticity of the airway compartment[26–28], with the main differentiation pathways visualized in Fig. 1d. Notably, transit epithelial 1 cells occur mostly in patients with COVID-19, but also in healthy children (Extended Data Fig. 2a) suggesting a function in development and tissue regeneration. Compared with published adult nasal datasets[14,28], we annotated cell types with greater granularity, especially for B and T lymphocytes, and we identified three Hillock-like populations[14,26,27]. The latter are all marked by *KRT14*, *KRT6A* and *KRT13*, which form a distinct differentiation trajectory (Fig. 1d) similar to the one reported in mice[27]. Moreover, monocytes fall into clearly distinct clusters, annotated by their highly expressed markers, IL-6[+] monocytes, GPBAR1[+] monocytes and CXCL10[+] monocytes, and were mostly derived from neonates with COVID-19 (Fig. 1c and Extended Data Figs. 2a and 3a).

## SARS-CoV-2 reads in airway epithelium

In COVID-19-positive nasal samples, we detected viral reads ($n \geq 10$) in 10 out of 28 patients, with the highest levels found in patients who were sampled closest to the estimated onset of infection (Fig. 1e). After filtering ambient RNA, the cell types with the highest proportion of viral reads were goblet 2 inflammatory cells, followed by cycling basal, transit epithelial and ciliated cells (Fig. 1f), largely mirroring *ACE2* expression (Extended Data Fig. 4). Viral reads were also detected in lymphocytes and myeloid cells (mostly macrophages), reflecting either active infection in macrophages[29] or merely uptake of virions or infected cells. The expression of SARS-CoV-2 viral entry and associated factors, including *ACE2*, was similar between children and adults, with few genes correlating with active viral infection (Extended Data Fig. 4a, b). In adults, *ACE2* expression is induced by IFN[30] and in response to infection[28], but we observed no significant increase of *ACE2* expression in children with COVID-19 (Extended Data Fig. 4c), consistent with recent bulk RNA-seq comparisons[10]. As reported[31,32], no SARS-CoV-2 viral reads were detected in peripheral blood.

## Airway cell type proportions in COVID-19

We next examined changes in cell type proportions for location, age group and COVID-19 status in all of the airway cell populations (Fig. 2a and Extended Data Fig. 5a, b). To test significance, we used a Poisson linear mixed model (Methods), enabling us to test the whole cohort in a single analysis while taking into account clinical metadata and technical factors (Extended Data Fig. 5b). Airway epithelial cell type composition showed trends of decreasing basal 1 and increasing secretory and goblet cells with age (Extended Data Fig. 3e), reflecting developmental trajectories from progenitors to differentiated cells (Fig. 1d). Notably, there were significant changes with location, as previously reported[33].

Contrasting epithelial cells in COVID-19 versus healthy adults, the most highly enriched cell types are transit epithelial 1 and goblet 2 inflammatory cells (Fig. 2a; all of the cell types are shown in Extended Data Fig. 5a). We hypothesized that the increased transit epithelial cell numbers reflects a compensatory replacement of dying ciliated cells[14,34] by their precursors, to maintain homeostasis after infection as seen in the lower airways[35,36], and consistent with trajectory analysis (Fig. 1d). This is further supported by the return to healthy cell population levels in patients after COVID-19 (Fig. 2a). In adults, the proportions of nasal immune cells were not significantly changed in COVID-19.

In children, epithelial cell proportions did not change but in the immune compartment IL6[+] monocytes were significantly enriched in COVID-19, with a trend towards higher CXCL10[+] monocytes and neutrophils. We also observed changes in immune cell populations over healthy childhood (Fig. 2a), such as high monocytes and low CD8 T cell levels in infants, and expansion of B cell populations in young children, reflecting a switch from innate to adaptive immunity[37].

## Distinct changes in children and adults

We next examined gene expression changes in children versus adults, in healthy individuals, patients with COVID-19 and patients after COVID-19. In nasal epithelial cells, the biggest changes were observed for gene expression signatures associated with IFNα signalling (Fig. 2b). Healthy adults had the lowest IFNα response that was strongly induced in COVID-19 and returned to preinfection levels in patients after COVID-19. In children, this gene signature was already activated and increased only slightly after infection. These patterns were repeated for signatures of IFNγ response, TNF signalling and neutrophil migration, albeit with smaller fold changes. For nasal immune cells, the induction

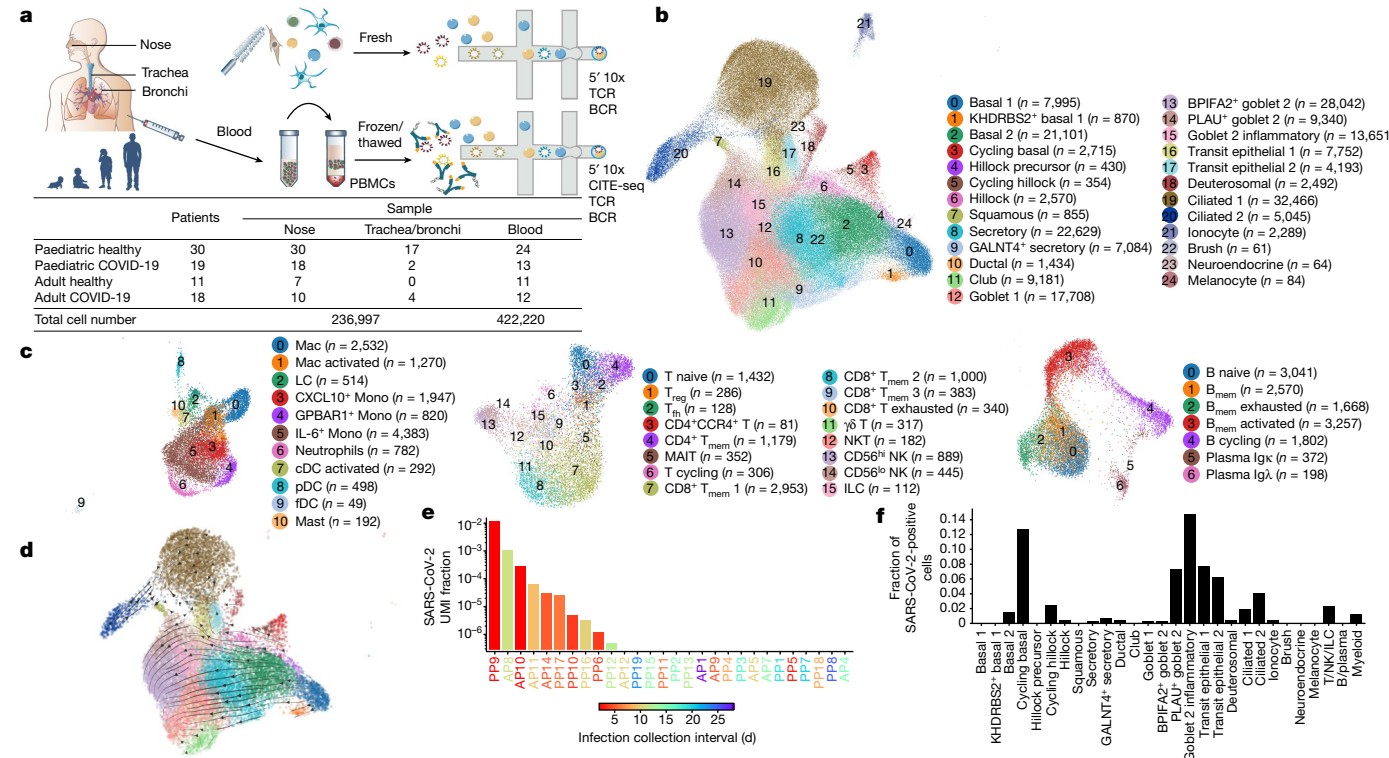

**Fig. 1 | Experimental outline and overview of results. a**, Visual overview of the experimental design, numbers of patients, samples taken and single cells sequenced. **b, c**, Uniform manifold approximation and projection (UMAP) visualization of annotated airway epithelial cells (**b**) and immune cells (**c**), with the cell numbers per cell type shown in parentheses. $B_{mem}$, memory B cells; cDCs, conventional dendritic cells; fDCs, follicular dendritic cells; ILCs, innate lymphoid cells; LCs, Langerhans cells; Mac, macrophages; MAIT, mucosal-associated invariant T cells; Mono, monocytes; NKT, natural killer T cells; pDCs, plasmacytoid dendritic cells; $T_{fh}$, T follicular helper cells; $T_{mem}$, memory T cells; $T_{reg}$, regulatory T cells. A full list of abbreviations is provided in the Supplementary Note. **d**, Airway epithelial cells in the same UMAP as **a** with RNA velocity of major epithelial cell types. **e**, The fraction of SARS-CoV-2 viral unique molecular identifiers (UMI) (where ≥10 were detected per donor) relative to total UMI per donor, before filtering out of ambient RNA, in descending order coloured by infection collection interval (days). This was calculated as the days between sample collection and estimated onset of infection, based on the first symptom onset or a positive SARS-CoV-2 RT−qPCR test, whichever was reported first for symptomatic patients, and the latter for asymptomatic patients. **f**, The fraction of airway cells with detected SARS-CoV2 mRNA in each cell type (with immune cells in broad categories) in patients with COVID-19 with detected viral RNA (≥5 viral UMI per donor following filtering out ambient RNA). $n = 9$.

---

of the IFNα response signature was higher in children than in adults. The other signatures examined also showed greater induction in children than in adults.

Examining these responses by cell types in healthy children versus adults, the IFN response signatures were already activated in children across several epithelial cell types, with the highest levels in goblet inflammatory cells, Hillock precursors and rare melanocytes (Fig. 2c; absolute values per cell type are shown in Extended Data Fig. 5c). However, SARS-CoV-2-induced IFN responses were higher in adults across many epithelial cell types. Many immune cell types in healthy children had elevated IFN response signatures compared to adults, particularly $CD56^{lo}$ natural killer cells, natural killer T cells, neutrophils, $CXCL10^+$ monocytes and some $CD8^+$ T cell subsets for IFNα, and a wider range for IFNγ (Fig. 2d). After infection, we observed a greater induction of these responses in immune cells in children, most prominently in monocytes, including in the already expanded $IL6^+$ monocytes, CD4 $CCR4^+$ T cells and T follicular helper cells.

In adults with COVID-19, a higher systemic IFN response has been reported for non-severe disease[14,38,39]. We confirmed this across disease severity in our adult cohort for the local response, finding a higher IFNα response in asymptomatic/mild versus moderate/severe disease in both epithelial and immune cells (Fig. 2e). In children, this phenomenon was much stronger in immune versus epithelial cells. These data suggest that, in both children and adults, a strong local IFN response is associated with a milder disease severity, presumably because interferons inhibit viral replication[13]. However, in children, this local response is preactivated in epithelial cells and stronger in immune cells, providing better protection against the virus.

We next examined differential gene expression patterns in healthy versus COVID-19 samples, followed by Gene Ontology term enrichment, in cell types that are particularly associated with disease: transit epithelial 1 and goblet 2 inflammatory cells upregulated in adult COVID-19, and IL-6 monocytes upregulated in children, as strong IFNα responders (Fig. 2f). For transit epithelial cells, this highlighted the IFN type I and II response as well as neutrophil chemotaxis, a notable finding given that neutrophil infiltration is linked to COVID-19 severity[40]. The neutrophil recruitment signature was driven by *S100A8* and *S100A9* expression (calprotectin) (Extended Data Fig. 5d), which is also a key correlate of disease severity[41]. For goblet 2 inflammatory cells and IL-6+ monocytes, the top two terms were type I IFN signalling and negative viral replication. Enrichment of motile cilium assembly is consistent with our observation that in disease there seems to be a higher cell turnover with precursors such as secretory cells differentiating to replace dying ciliated cells.

As calprotectin expression has primarily been associated with myeloid cells, we validated expression at the protein level in epithelial cells. Figure 2g shows double-positive cells, staining for both calprotectin subunit S100A9 and the epithelial marker EPCAM, in a posterior nasal space biopsy of an adult patient with COVID-19. At the RNA level, calprotectin is expressed across different secretory cell types (Extended Data Fig. 3b).

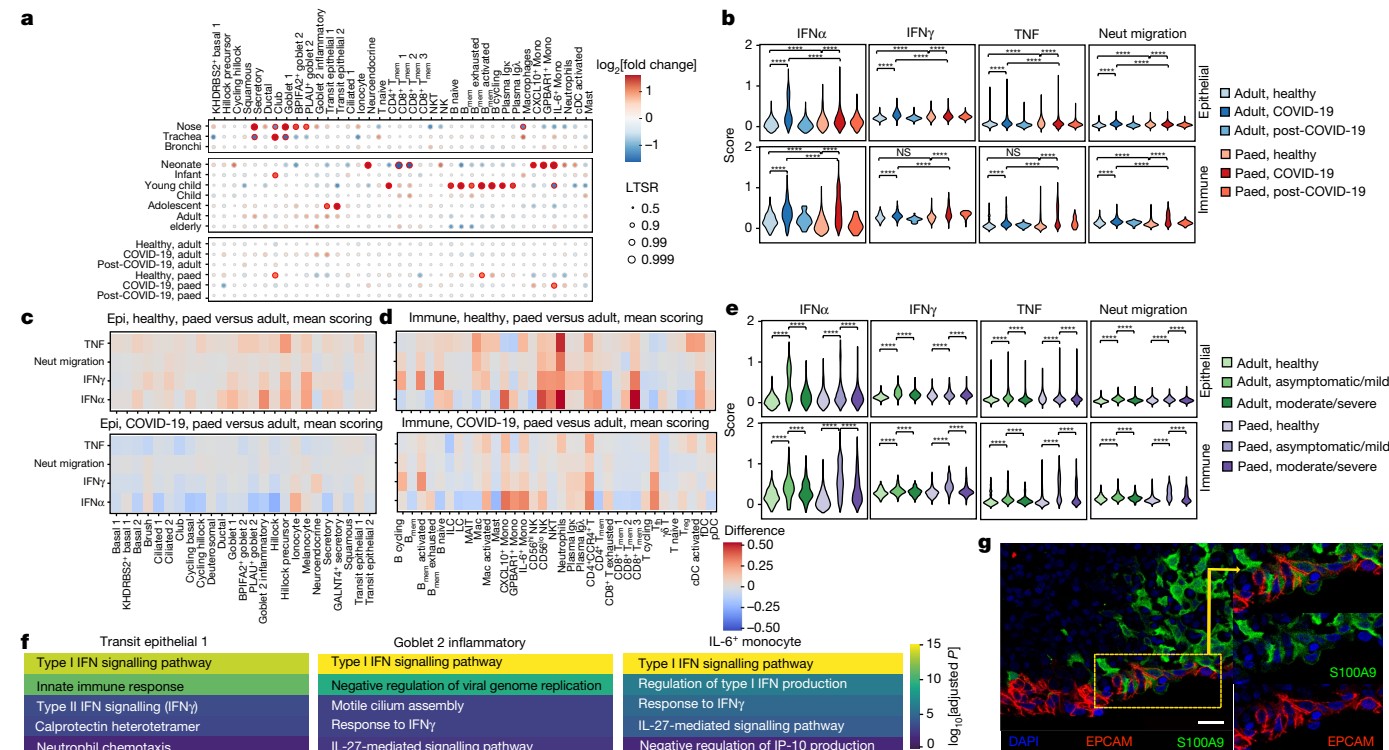

**Fig. 2 | Differences in airway epithelial and immune cells between paediatric and adult patients with COVID-19. a**, The fold change in and statistical significance of major airway cell type proportions across location of sampling, age group and COVID-19 status, estimated by fitting Poisson generalized linear mixed models taking into account other technical and biological variables (Methods). The red circles indicate local true sign rate (LTSR) > 0.95. Paed, paediatric. **b**, Comparison of the expression signature of cellular response to IFNα, IFNγ, TNF signalling and neutrophil migration signalling across COVID-19 status and age groups. Neut, neutrophil. **c**, **d**, Heat maps comparing these expression signatures in healthy paediatric versus adult individuals, and in paediatric versus adult patients with COVID-19 in epithelial cells (**c**) (the colours indicate difference in scoring) and in immune cells (**d**). **e**, Comparison of expression signatures across COVID-19 severity and age groups. **f**, Representative five enriched Gene Ontology terms in genes that are upregulated in COVID-19 samples in transit epithelial 1 cells, goblet 2 inflammatory cells and IL-6⁺ monocytes. **g**, Immunohistochemistry confocal microscopy image showing S100A9 expression (green) by epithelial cells (EPCAM, magenta) in the nasal epithelium. Nuclei were stained with DAPI (blue). Scale bar, 20 μm. One representative section out of four technical replicates is shown. For **b**, **e**, pairwise comparisons were performed using two-sided Wilcoxon rank-sum tests; NS, $P > 0.05$; ****$P < 2.2 \times 10^{-16}$.

## Multi-omic blood immune landscape

Using CITE-seq and single-cell profiling of blood from paediatric and adult patients with COVID-19, we annotated 422,220 high-quality single-cell transcriptomes from healthy donors, donors with COVID-19 or donors who had recovered from COVID-19, into 34 blood cell types (Fig. 3a; marker expression and annotation validation is shown in Extended Data Fig. 6a–c). To investigate how the immune system responds to SARS-CoV-2, and how age affects this response, we calculated the fold changes in the proportions of cell types that can be attributed to disease state and age (Fig. 3b and Extended Data Fig. 6d–g). Importantly, our Poisson linear mixed model enabled us to distinguish the immune dynamics that can be attributed to technical effects, ageing and COVID-19. Furthermore, we included an interaction between adulthood and disease status to uncover paediatric-specific immune responses to COVID-19 (Fig. 3b). We observed higher plasma cell and plasmablast proportions, as well as a reduction in the monocyte and dendritic cell compartment in the blood of both adult and paediatric patients with COVID-19, as previously reported in adults[21,23].

## Reduced cytotoxic response in children

In contrast to the aforementioned cell types that change consistently in adults and children in response to COVID-19, we observed opposing changes in the abundance of many other immune cell types (Fig. 3b).

The circulating immune system of adult patients with COVID-19 is characterized by an increased cytotoxic compartment, in which CD8⁺ cytotoxic T lymphocytes and effector memory cells re-expressing CD45RA are significantly more abundant in adults. Notably, the latter populations, natural killer cells and CD4⁺ cytotoxic T lymphocytes are reduced in paediatric patients with COVID-19. Together, this could reflect a more systemic infection and inflammation in adults, whereas the infection in paediatric patients remains more restricted to airways.

## Naive T cells in children with COVID-19

In addition to a reduced cytotoxic cellular composition, we observed a striking increase in naive lymphocytes in the blood of paediatric patients with COVID-19 (Fig. 3b). High numbers of naive cells may be attributed to an increased release of immature B and T lymphocytes from the bone marrow and thymus, respectively, or due to migration of more mature cells to the site of infection. With our statistical model and large cohort of healthy individuals, the strong effects of age on the immune landscape were deconvoluted from the COVID-19 effects into independent age effects and quantified in Fig. 3b. The strong maturation patterns and shift from innate to adaptive immunity observed over healthy childhood amplifies some of the paediatric-specific COVID-19 responses; that is, not only do children have a more naive and reduced cytotoxic response to COVID-19, but they also start off with an immune state that is already skewed towards this response.

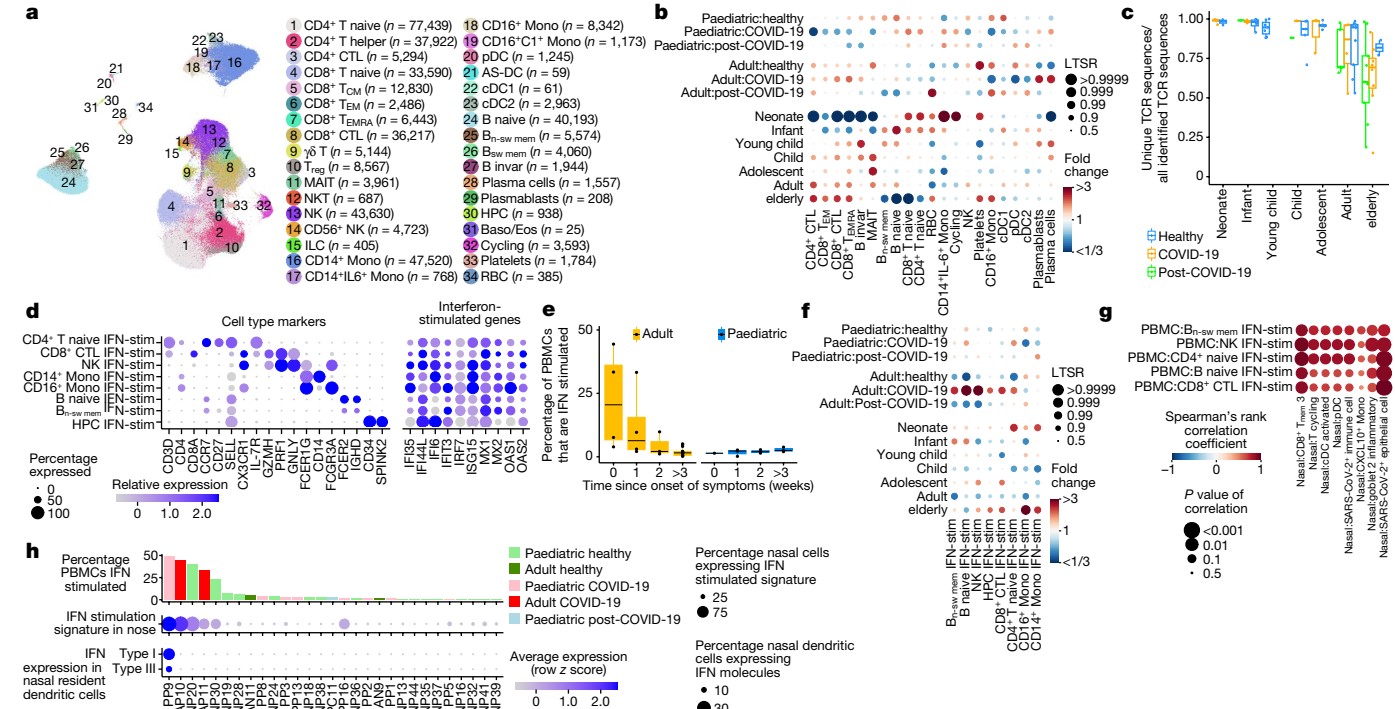

**Fig. 3 | Differences in immune response between paediatric and adult patients with COVID-19. a**, UMAP visualization of 422,220 PBMCs incorporating both protein- and RNA-expression data. AS-DC, AXL⁺SIGLEC6⁺ dendritic cells; Baso/Eos, basophil/eosinophil; CM, central memory; CTL, cytotoxic T lymphocytes; EM, effector memory; EMRA, effector memory re-expressing CD45RA; invar, invariant; n-sw, non-switched; RBCs, red blood cells; sw, switched. **b**, The fold changes in the proportions of immune cell types across age group and disease status, taking into account confounding factors (Methods). Only cell types that change with a local true sign rate of >0.90 in the disease status groups are shown (all of the cell types are shown in Extended Data Fig. 6d, e). This analysis does not include the cells analysed in **f**. **c**, The fraction of unique TCR sequences in different age groups. **d**, Cell-type-marker expression alongside IFN-stimulated genes. The colour is scaled to all other cell types (Extended Data Fig. 6a). HPC, haematopoietic progenitor cell. **e**, The

percentage of IFN-stimulated PBMCs of each symptomatic patient with COVID-19, grouped by the weeks since the onset of symptoms. **f**, Dot plot as in **b** showing the IFN-stimulated subpopulations (IFN-stim) across age and disease status. **g**, Correlation analysis comparing the blood and nose, using a Spearman rank-order correlation coefficient between relative proportion of PBMC subtypes (*y* axis) and nasal cell types (*x* axis) (Extended Data Fig. 7d, e.) **h**, IFN stimulation in PBMCs and nasal cells, and nasal IFN production in individuals with matched nasal and PBMC data (detailed gene expression dynamics are shown in Extended Data Fig. 8). Dots in **c, e** represent independent patient samples. For **e**, the box plots show the median (centre line), the first and third quartiles (box limits), and the whiskers extend to the lowest and highest values within 1.5 × interquartile range. All cell type abbreviations are provided in the Supplementary Note.

## Diverse immune repertoire in children

As we detected more naive immune cells in children, we hypothesized that this could affect the amount of unique T and B cell receptors (TCRs and BCRs) that are available to detect new pathogens. Indeed, we observed that the pool of detected TCRs becomes increasingly dominated by expanded clones over age (Fig. 3c and Extended Data Fig. 7a), reducing the amount of unique TCRs that are available to detect unseen pathogens. It is therefore conceivable that a higher TCR repertoire diversity in children could contribute to a faster, more efficient adaptive immune response to SARS-CoV-2.

## IFN-stimulated cell subtypes in blood

When annotating our PBMC dataset, we noticed further cell type heterogeneity that generated distinct clusters within all major immune cell types due to high expression of IFN-stimulated genes (Fig. 3d and Extended Data Fig. 7b). Activation of IFN signalling is a key hallmark of COVID-19, acting both as an important protective pathway that can equally be associated with severe COVID-19 (refs. [15,42,43]). Although we and others reported an association between global changes in IFN related gene expression and COVID-19 (ref. [23]), our increased granularity enabled us to distinguish multiple distinct stimulated and unstimulated populations alongside each other within donors. Importantly, this shows that IFN stimulation of PBMCs does not lead

to a global activation of gene expression, but is restricted to a subset of circulating cells.

## IFN response in early COVID-19

When investigating the COVID-19 IFN response, we found that IFN-stimulated natural killer, B, T and haematopoietic progenitor cell subpopulations are much more abundant in adult patients compared with paediatric patients with COVID-19 (Fig. 3e, f). In adults, the amount of IFN-stimulated PBMCs is strongly correlated with sampling time since onset of symptoms (Fig. 3e). This suggests that IFN-stimulated PBMCs are a characteristic of the acute phase of infection, when the innate immune response is trying to control the viral infection. In children, the correlation with onset of symptoms is completely absent (Fig. 3e) but IFN-stimulated cells were abundant in some asymptomatic children (Extended Data Fig. 7c), suggesting a much faster induction and clearance of IFN-stimulated cells. Together, these observations support our hypothesis that COVID-19-induced inflammation and cytotoxicity in the blood is more abundant in adults than in children.

## Dendritic cells initiate IFN response

To investigate the connection between the local and systemic immune response to SARS-CoV-2, we compared cell type proportions in the blood and nose for multi-tissue donors and observed strong correlations

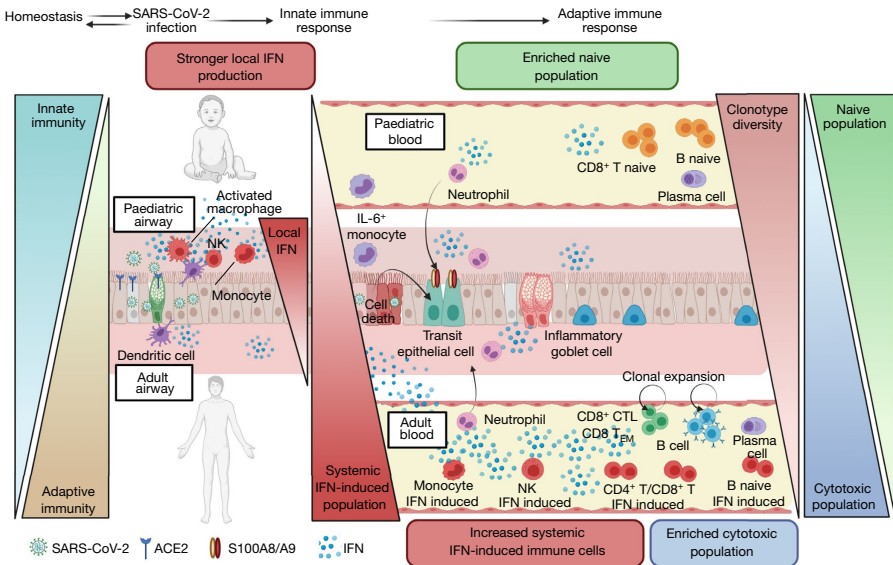

**Fig. 4 | The local and systemic response to SARS-CoV-2 infection in children and adults.** Schematic of the difference in the airway and systemic immune response to SARS-CoV-2 infection between children and adults, reflecting the maturation of the immune landscape throughout childhood to adulthood. The key points are (1) immune cell proportions display strong maturation patterns throughout healthy childhood and adulthood, with a notable innate to adaptive immunity switch. (2) In the airways, the local innate IFN response to SARS-CoV-2 is stronger in paediatric airway immune cells compared with adult airway immune cells. (3) In the blood, the systemic innate IFN response to SARS-CoV-2 is stronger in adults, with a notable increase in IFN-stimulated subpopulations, whereas the adaptive immune response is characterized by expanded cytotoxic populations in adults compared with naive populations in children. (4) Epithelial cells with an inflammatory gene expression (S100A8/S100A9) are found enriched in patients with COVID-19. (5) Clonotype diversity decreases with age. The figure was generated using BioRender.com.

(Fig. 3g; all comparisons are shown in Extended Data Fig. 7d, e). Particularly, SARS-CoV-2-infected and inflammatory nasal epithelial cells, and nasal plasmacytoid and conventional dendritic cells correlated with IFN stimulation in the blood. This is interesting as dendritic cells are known for their viral-sensing and IFN-production capacities[44], but this has not been directly observed in COVID-19. Although dendritic cells protect against severe disease[45], most COVID-19 studies that analysed blood reported a depletion of dendritic cells[46]. However, here we provide evidence that, at the earliest stages of infection, type I and type III IFNs are detectable (Fig. 3h) and are produced by plasmacytoid and conventional dendritic cells, but not other immune or epithelial cells (Extended Data Fig. 8b and Supplementary Note).

## Discussion

Here we focused on why children are generally protected from severe COVID-19 and propose multiple mechanisms (Fig. 4). First, we show that the airway epithelium has a higher steady-state expression of IFN-response genes in children. SARS-CoV-2 has been reported to be highly sensitive to prestimulation with interferons[47], and preactivation may restrict viral spread in children. Second, the systemic immune response in blood is characterized by a more naive state. By contrast, adults display a highly cytotoxic immune compartment in the blood, probably due to a failure to restrict viral spreading. This elevated systemic response in adults can lead to widespread immune-related organ damage[48]. A third feature that we observed was the higher TCR repertoire diversity in children versus adults. The acquisition of memory T and B cells during childhood and adulthood, combined with reduced thymic output, shifts the adaptive immune system into a more memory-based compartment in aged individuals[49]. This reduces the pool of unique immune receptors within naive lymphocytes[50], making it less probable that a high-affinity immune receptor is directly available against SARS-CoV-2 antigens. Finally, we found previously undescribed IFN-stimulated cell states in multiple blood cell lineages that are highly abundant in early disease in adults. This presents an added inflammatory feature of the already cytotoxic immune compartment in adult patients with COVID-19, and possibly amplifies any pathological effects of the systemic immune response. The identification of both IFN-stimulated and unstimulated blood cells within donors underscores that activation is cell specific rather than, as noted by others, systemic, possibly caused by either close proximity to the site of infection or an associated secondary lymphoid organ, or cell-to-cell variability in responsiveness as we have shown in fibroblasts and phagocytes[51].

SARS-CoV-2 infection frequently starts in the upper airways, in which we found the highest total viral load in surface epithelial goblet, ciliated and differentiating cells. Viral infections are cleared by cell death and the removal of the infected cells[52], leading to a highly dynamic restructuring of the airway epithelium with a marked increase in developmental intermediates, most notably the transit epithelial populations, which are rebalanced after infection. We also observed a strong neutrophil recruiting signature, driven by the expression of calprotectin in epithelial cell types, highlighting the key role of epithelial cells in initiating an innate immune response.

Overall, our study provides multiple insights using paired multi-omics profiling of both airway epithelium and peripheral blood to fill the gap in our understanding of paediatric epithelial and immune responses to COVID-19, while also identifying previously undescribed cell states in both airway epithelium and blood. These insights could contribute to pinpointing the triggers of severe disease in adults with a view towards risk stratification and therapeutic intervention.

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

**NU SCRIPT Study Investigators**

A. Christine Argento[10], Catherine A. Gao[10], Alexander V. Misharin[10], G. R. Scott Budinger[10], Jane E. Dematte[10], Helen K. Donnelly[10], Nikolay S. Markov[10], Richard G. Wunderink[10], Sean B. Smith[10], Taylor A. Poor[10] & Ziyan Lu[10]

## Methods

### Study participants and design

**The UK cohort.** Participants were included from five large hospital sites in London, UK, namely Great Ormond Street Hospital NHS Foundation Trust, University College London Hospitals NHS Foundation Trust, Royal Free London NHS Foundation Trust (Royal Free Hospital and Barnet Hospital) and Whittington Health NHS Trust from March 2020 to February 2021. Ethical approval was given through the Living Airway Biobank, administered through the UCL Great Ormond Street Institute of Child Health (REC reference: 19/NW/0171, IRAS project ID: 261511, North West Liverpool East Research Ethics Committee), REC reference 18/SC/0514 (IRAS project: 245471, South Central Hampshire B Research Ethics Committee) administered through the University College London Hospitals NHS Foundation Trust and REC reference 18/EE/0150 (IRAS project ID: 236570, East of England Cambridge Central Research Ethics Committee) administered through Great Ormond Street Hospital NHS Foundation Trust, REC reference 08/H0308/267 administered through the Cambridge University Hospitals NHS Foundation Trust, as well as by the local R&D departments at all hospitals. All of the study participants or their surrogates provided informed consent. At daily virtual COVID-19 coordination meetings, suitable patients were chosen from a list of newly diagnosed patients who were admitted within the preceding 24 h. Only patients with COVID-19 who tested positive for SARS-CoV-2 by a quantitative PCR with reverse transcription (RT–qPCR) nasopharyngeal test were enrolled in the study; a summary of symptom onset relative to RT–qPCR testing and sampling is shown in Extended Data Fig. 1b. Patients with typical clinical and radiological COVID-19 features but with a negative screening test for SARS-CoV-2 were excluded. Other excluding criteria included active haematological malignancy or cancer, known immunodeficiencies, sepsis from any cause and blood transfusion within 4 weeks. Two cases of paediatric multisystem inflammatory syndrome (PIMS-TS, named by the Royal College of Paediatrics and Child Health) were included (airway samples only), which is also referred to as multisystem inflammatory syndrome in children (MIS-C) by the World Health Organization, with little to no MIS-C-specific difference detected after analysis in the nasal mucosa compared with equivalent samples from paediatric patients with COVID-19 (ref. [53]). Maximal severity of COVID-19 was determined retrospectively by determining the presence of symptoms, the need for oxygen supplementation and the level of respiratory support (mild, symptomatic without oxygen requirement or respiratory support; moderate, requiring oxygen without respiratory support; severe, requiring non-invasive or invasive ventilation). Brushings and peripheral blood sampling were performed by trained clinicians before inclusion in any pharmacological interventional trials, with the exception of 3 paediatric patients with COVID-19 (noted in Extended Data Table 1) and ideally within 48 h of a positive SARS-CoV-2 test. All of the participants for our paediatric healthy cohort were recruited from Great Ormond Street Hospital NHS Foundation Trust and were eligible for inclusion if they were <18 years old and asymptomatic for respiratory viral infections at time of sampling. At the start of the study, initiated in March 2020 it was not standard practice for hospitals to test healthy asymptomatic patients. Therefore 8 (out of 30) of the earliest recruited participants were untested and assumed negative. To confirm this assumption and to look for any other undetected asymptomatic infections metagenomic analysis on the entire dataset was performed (see 'Metagenomic analysis'; Extended Data Fig. 9). Participants for our adult healthy cohort were recruited from University College London Hospitals and associated research laboratories at University College London and were eligible for inclusion if >18 years and asymptomatic with a current negative SARS-CoV-2 test (RT–qPCR or rapid-antigen testing). Exclusion criteria for the cohort included active haematological malignancies or cancer, known immunodeficiencies, sepsis from any cause and blood transfusions within 4 weeks, known bronchial asthma, hay fever, diabetes and other known chronic respiratory diseases such as cystic fibrosis, interstitial lung disease and chronic obstructive pulmonary disease. There were three exceptions to these criteria in our paediatric cohort; NP28 who was later discovered to have asthma and NP10 who was reported to have a immunocompromised status in underlying comorbidities, but for whom only nasal brushes were included; and NP27, who did not have any respiratory problem but was subsequently diagnosed with endocarditis. Exclusion of these individuals did not alter any of our conclusions. For some patients included in our COVID-19 cohort, matched convalescent blood was taken on the day of hospital discharge and analysed separately in our post-COVID-19 cohort alongside symptomatic patients recruited from University College London Hospitals outpatient COVID-19 follow-up clinic, who were recalled around 3 months after recovering from severe COVID-19 using the exclusion criteria as stated for our COVID-19 cohort. Participants were further divided into subgroups to enable us to look at age-specific effects. These were classified based on the World Health Organisation; neonates (0–30 days), infants (1–24 months), young children (2–5 years), children (6–11 years), adolescents (12–17 years) and adults (≥18 years); adults were further broken down into adults (18–64 years) and elderly (≥65 years).

**Chicago cohort (adult bronchial samples).** Ethical approval for sample collection from patients with severe pneumonia was given by Northwestern Institutional Review Board, study STU00204868 (PI Richard Wunderink). Samples from patients with COVID-19, viral pneumonia and other pneumonia, and non-pneumonia controls were collected from participants enrolled in the Successful Clinical Response in Pneumonia Therapy (SCRIPT) study STU00204868 and admitted to the ICU at Northwestern Memorial Hospital, Chicago. All of the study participants or their surrogates provided informed consent. Individuals aged at least 18 years with suspicion of pneumonia based on clinical criteria (including but not limited to fever, radiographic infiltrate and respiratory secretions) were screened for enrolment into the SCRIPT study. The inability to safely perform bronchoalveolar lavage or non-bronchoscopic bronchoalveolar lavage was considered to be an exclusion criteria. In our centre, patients with respiratory failure are intubated on the basis of the judgement of bedside clinicians for worsening hypoxaemia, hypercapnia or work of breathing refractory to high-flow oxygen or non-invasive ventilation modes. Bronchial brushings were performed during the diagnostic bronchoalveolar lavage procedure and the samples were collected from representative sites at the lobar bronchi.

### Sample collection

**The UK cohort.** Samples were collected and transferred to a category level 3 facility at University College London and processed within 2 h of sample collection. Nasal, tracheal and bronchial brushings were enzymatically digested to a single-cell suspension and processed further immediately. Peripheral blood was centrifuged after adding Ficoll Paque Plus and PBMCs, serum and neutrophils were separated, collected and frozen for later processing. A local anaesthetic endoscopically guided biopsy of the postnasal space mucosa was collected from a 19-year-old female patient three weeks after onset of mild COVID-19 symptoms (REC: 08/H0308/267). SARS-CoV-2 virus was confirmed by RT–PCR testing at the time of symptom onset.

**Chicago cohort (adult bronchial samples).** Samples were collected in the ICU at Northwestern Memorial Hospital, transferred to a research laboratory in the Simpson Querrey Biomedical Research Center, Feinberg School of Medicine, Northwestern University, and processed within 1 h of sample collection in biological safety level 2 facility using biological safety level 3 practices. After collection, bronchial brushings were stored in Hypothermosol (Stem Cell Technologies, 07935) at 4 °C.

### Nasal and tracheal brushing tissue dissociation

**The UK cohort.** Nasal brushing was performed on the inferior nasal concha zone with a cytological brush (Scientific Laboratory Supplies, CYT1050). All of the samples were processed fresh

according to a previously described protocol[26] with minor modifications[54]. The brushes were immediately placed in a 15-ml sterile Falcon tube containing 4 ml of transport medium (αMEM supplemented with 1× penicillin–streptomycin (Gibco, 15070), 10 ng ml[−1] gentamicin (Gibco, 15710) and 250 ng ml[−1] amphotericin B (Thermo Fisher Scientific; 10746254)) on ice. Once in the category level 3 facility, the tube was shaken vigorously to collect cells in suspension. The brushes were then carefully transferred into a new Falcon tube containing HBSS and shaken to remove residual cells from the brush. This was repeated until all of the cells looked like they had been collected from the brush. All of the Falcon tubes were centrifuged at 400g for 5 min at 4 °C. The cell pellet was collected from each tube and then put in a dissociation buffer consisting of 10 mg ml[−1] protease from *Bacillus licheniformis* (Sigma-Aldrich, P5380) and 0.5 mM EDTA in HypoThermosol (Stem Cell Technologies, 07935) for dissociation on ice for 30 min. Every 5 min, cells were gently triturated using a 21 G and 23 G needle. After incubation, protease was inactivated by adding 200 µl of inactivation buffer (HBSS containing 2% BSA). The suspension was centrifuged at 400g for 5 min at 4 °C and the supernatant was discarded. Cells were resuspended in 1 ml wash buffer (HBSS containing 1% BSA) and centrifuged again. Red blood cell lysis was performed if needed, followed by an additional wash. The single-cell suspension was forced through a 40-µm Flowmi Cell Strainer. Finally, the cells were centrifuged and resuspended in 30 µl of resuspension buffer (HBSS containing 0.05% BSA). Using Trypan Blue, total cell counts and viability were assessed. The cell concentration was adjusted for 5,000 targeted cell recovery according to the 10x Chromium manual before loading onto the 10x chip (between 700–1,000 cells per µl) and processing immediately for 10x 5′ single-cell capture using the Chromium Single Cell V(D)J Reagent Kits V1.0 (Rev J Guide), the newer chromium Next GEM Single Cell V(D)J Reagent Kit v1.1 (Rev E Guide) or the chromium Next GEM Single Cell 5′ V2 (Dual index) kit (Rev A guide).

For a small subset of nasal samples (PP5_NB_2, PP6_NB_2, AP11_NB, AP12_NB, AP13_NB and AP14_NB_2) 1 µl viral RT oligo (at 5 µM, PAGE) was spiked into the master mix (at step 1.2.b in the 10x guide; giving a final volume of 75 µl) to help with the detection of SARS-CoV-2 viral reads. The samples were then processed according to the manufacturer's instructions, with the viral cDNA separated from the gene expression libraries (GEX) by size selection during step 3.2. Here the supernatant was collected (159 µl) and transferred to a new PCR tube and incubated with 70 µl of SPRI beads (0.6× selection) at room temperature for 5 min. The SPRI beads were then washed according to the guide and the viral cDNA was eluted using 30 µl of EB buffer. No changes to the transcriptome were observed between samples, which were run both with and without the viral oligo and only a small increase in the overall number of SARS-CoV-2 reads detected. The RT oligo sequence was as follows: 5′-AAGCAGTGGTATCAACGCAGAGTACTTACTCGTGTCCTGTCAACG-3′

**Chicago cohort (adult bronchial samples).** Samples were processed using a previously reported protocol[26] with minimal modifications. Specifically, dissociation was performed without EDTA and trituration was performed by pipetting using a regular-bore 1,000 µl tip every 5 min. Dissociation was visually confirmed by inspecting an aliquot of the single-cell suspension using phase contrast on an inverted microscope. Cell count was performed using the AO/PI reagent on the K2 Cellometer (Nexcelom). Approximately 300,000–500,000 cells were obtained per brush with a viability of 97% and above. Cells were captured on a 10x Chromium Single Cell Controller using the Chromium Single Cell V(D)J Reagent Kits V1.0 (Rev J Guide).

**PBMC isolation from peripheral blood**
Peripheral blood was collected in EDTA immediately after the nasal brushing procedure. The blood was diluted with 5 ml of PBS containing 2 mM EDTA (Invitrogen, 1555785-038). Diluted blood (10–20 ml) was carefully layered onto 15 ml of Ficoll-Paque Plus (GE healthcare, 17144002). If the sample volume was less than 5 ml, blood was diluted

with an equal volume of PBS-EDTA and layered onto 3 ml Ficoll. The sample was centrifuged at 800g for 20 min at room temperature. The plasma layer was carefully removed and the PBMC layer was collected using a sterile Pasteur pipette. The PBMC layer was washed with 3 volumes of PBS containing EDTA by centrifugation at 500g for 10 min. The pellet was suspended in PBS-EDTA and centrifuged again at 300g for 5 min. The PBMC pellet was collected, and then both cell number and viability were assessed using Trypan Blue. Cell freezing medium (90% FBS, 10% DMSO) was added dropwise to PBMCs slowly on ice and the mixture was then cryopreserved at −80 °C until further full sample processing.

**CITE-seq staining for single-cell proteogenomics**
Frozen PBMC samples were thawed quickly at 37 °C in a water bath. Warm RPMI1640 medium (20–30 ml) containing 10% FBS was added slowly to the cells before centrifuging at 300g for 5 min. This was followed by a wash in 5 ml RPMI1640-FBS. The PBMC pellet was collected, and the cell number and viability were determined using Trypan Blue. PBMCs from four different donors were then pooled together at equal numbers: $1.25 \times 10^5$ PBMCs from each donor were combined with the other PBMCs to make up $5.0 \times 10^5$ cells in total. The remaining cells were used for DNA extraction (Qiagen, 69504). The pooled PBMCs were resuspended in 25 µl of cell staining buffer (BioLegend, 420201) and blocked by incubation for 10 min on ice with 2.5 µl Human TruStain FcX block (BioLegend, 422301). The PBMC pool was then stained with TotalSeq-C antibodies (BioLegend, 99814) according to the manufacturer's instructions. For a full list of TotalSeq-C antibodies, refer to ref. [23]. After incubating with 0.5 vials of TotalSeq-C for 30 min at 4 °C, PBMCs were washed three times by centrifugation at 500g for 5 min at 4 °C. PBMCs were counted again and processed immediately for 10x 5′ single cell capture (Chromium Next GEM Single Cell V(D)J Reagent Kit v1.1 with Feature Barcoding technology for cell Surface Protein-Rev D protocol). Two lanes of 25,000 cells were loaded per pool onto a 10x chip.

**Library generation and sequencing**
The Chromium Single Cell 5′ V(D)J Reagent Kit (V1.0 chemistry), Chromium Next GEM Single Cell 5′ V(D)J Reagent Kit (V1.1 chemistry) or Chromium Next GEM Single Cell 5′ v2 kit (V2.0 chemistry) was used for single-cell RNA-seq library construction for all airway samples, and the Chromium Next GEM Single Cell V(D)J Reagent Kit v1.1 with Feature Barcoding technology for cell surface proteins was used for PBMCs. GEX and V(D)J libraries were prepared according to the manufacturer's protocol (10x Genomics) using individual Chromium i7 Sample Indices. The cell surface protein libraries were created according to the manufacturer's protocol with slight modifications that included doubling the SI primer amount per reaction and reducing the number of amplification cycles to 7 during the index PCR to avoid the daisy chains effect. GEX, V(D)J and cell surface protein indexed libraries were pooled at a ratio of 1:0.1:0.4 and sequenced on a NovaSeq 6000 S4 Flowcell (paired-end, 150 bp reads) aiming for a minimum of 50,000 paired-end reads per cell for GEX libraries and 5,000 paired-end reads per cell for V(D)J and cell surface protein libraries.

**Single-cell RNA-seq computational pipelines, processing and analysis**
The single-cell data were mapped to a GRCh38 ENSEMBL 93 derived reference, concatenated with 21 viral genomes (featuring SARS-CoV-2), of which the NCBI reference sequence IDs are: NC_007605.1 (EBV1), NC_009334.1 (EBV2), AF156963 (ERVWE1), AY101582 (ERVWE1), AY101583 (ERVWE1), AY101584 (ERVWE1), AY101585 (ERVWE1), AF072498 (HERV-W), AF127228 (HERV-W), AF127229 (HERV-W), AF331500 (HERV-W), NC_001664.4 (HHV-6A), NC_000898.1 (HHV-6B), NC_001806.2 (herpes simplex virus 1), NC_001798.2 (herpes simplex virus 2), NC_001498.1 (measles morbillivirus), NC_002200.1 (mumps rubulavirus), NC_001545.2 (rubella), NC_001348.1 (varicella zoster virus), NC_006273.2 (cytomegalovirus) and NC_045512.2 (SARS-CoV-2).

When examining viral load per cell type, we first removed ambient RNA by SoupX and only included SARS-CoV-2-positive donors where ≥5 viral reads were still detected. Antibody-derived tag counts and gene expression counts in CITE-seq data were jointly quantified using Cell Ranger v.3.0.2. The alignment, quantification and preliminary cell calling of airway samples were performed using the STARsolo functionality of STAR v.2.7.3a, with the cell calling subsequently refined using the Cell Ranger v.3.0.2 version of EmptyDrops[55]. This algorithm has been made available as emptydrops on PyPi. Initial doublets were called on a per-sample basis by computing Scrublet[56] scores for each cell, propagating them through an over-clustered manifold by replacing individual scores with per-cluster medians, and identifying statistically significant values from the resulting distribution, replicating the approach of refs. [57,58]. The clustering was performed with the Leiden[59] algorithm on a k-nearest neighbour graph of a principal component analysis (PCA) space derived from a log[counts per million/100 + 1] representation of highly variable genes, according to the SCANPY protocol[60], and overclustering was achieved by performing an additional clustering of each resulting cluster. The primary clustering also served as an input for ambient RNA removal using SoupX[61].

### Metagenomics analysis

To ensure that the patients in our cohort did not carry undiagnosed infections, we carried out a metagenomic analysis using mg2sc (https://github.com/julianeweller/mg2sc). The metagenomic tool Kraken 2 (ref. [62]) was installed according to the standard instructions on GitHub[63,64]. The prebuilt standard Kraken 2 database was downloaded from https://benlangmead.github.io/aws-indexes/k2 (standard from 12 February 2020, 36 GB). Only reads that were not aligned to *Homo sapiens* with STARsolo[65] were extracted from the STARsolo and converted into FASTQ using bedtools (v.2.30)[66,67] for subsequent metagenomic analysis. This was performed using python scripts available on GitHub (https://github.com/julianeweller/mg2sc) and the command 'scMeG-kraken.py --input [bamfile, e.g. starsolo/Aligned.sortedByCoord.out.bam] --outdir [output directory] / --DBpath [path to kraken database] --threads [#, e.g. 8] --prefix [prefered file prefix] --verbosity [error/warning/info/debug]' resulting in a matrix of cell barcodes with assigned taxonomy transcript counts. Organisms shown are highly variable between samples with min_mean = 0.08, max_mean = 10 and min_disp = 0.05. The results are shown in Extended Data Fig. 9.

### Confocal microscopy method

Nasal epithelial biopsies were placed in Antigenfix (Microm Microtech) for 1–2 h at 4 °C, then 30% sucrose in PBS for 12–24 h at 4 °C, before cryopreservation in OCT (Cell Path). Sections (30 μm) were permeabilized and blocked in PBS containing 0.3% Triton X-100 (Sigma-Aldrich), 1% normal goat serum, 1% normal donkey serum and 1% BSA (R&D) for 1–2 h at room temperature. The samples were stained with anti-human S100A9 antibodies conjugated to FITC (1 in 50 dilution, MRP 1H9, BioLegend) and anti-human EPCAM antibodies conjugated to APC (1 in 50 dilution, MRP14, BioLegend, 350703) in blocking buffer overnight and washed three times for 10 min in PBS before mounting with Fluoromount-G containing DAPI (Invitrogen). Images were acquired using a Leica SP8 confocal microscope. Raw imaging data were processed using Imaris (Bitplane).

### Airway single-cell RNA-seq data processing

**Quality control, normalization and clustering.** To account for large quality variance across different samples, quality control was performed on SoupX-cleaned expression matrixes for each sample separately. Quality control thresholds were automatically established by fitting a 10-component Gaussian mixture model to the log-transformed UMI count per cell and to the percentage of mitochondrial gene expression and finding the lower or higher bounds where the probability density falls under 0.05. We also excluded cells with haemoglobin expression >0.1% of total expression and genes expressed in fewer than 3 cells. Expression values were then normalized to a sum of $1 \times 10^4$ per cell and log-transformed with an added pseudocount of 1. Highly variable genes were selected within each sample and then merged with the top 3,000 most commonly found genes chosen using the Scanpy[60] function scanpy.pp.highly_variable_genes(). After removing mitochondrial and ribosomal genes from the list of highly variable genes, principle component analysis was performed and the top 30 principle components were selected as input for BBKNN[68] to correct for batch effects between donors and compute a batch-corrected k-nearest neighbour graph. Leiden clustering was performed on this graph with a resolution of 0.2 to separate broad cell types (epithelial cells, B/plasma cells, T/natural killer/innate lymphoid cells and myeloid cells). For each broad cell type, clustering was then repeated, starting from highly variable gene discovery to achieve a higher resolution and a more accurate separation of refined cell types. Subclusters were manually examined and further reclustered when necessary.

**Quantifying SARS-CoV-2 viral expression.** For donor-level quantification, we took the data before ambient RNA removal by SoupX, as ambient viral RNA still reflects totally viral load. For cell-type-level quantification, we used the data after ambient RNA removal, as ambient viral RNA cannot be assigned to specific cells.

**Developmental trajectory inference.** RNA velocity analysis was performed to infer developmental trajectory for the major epithelial cell types (excluding melanocytes, ionocytes, brush cells and neuroendocrine cells). Spliced and unspliced UMI counts were generated using the STARsolo functionality of STAR v.2.7.3a. scvelo was used to fit a dynamical model as previously described[69] on the basis of the top 2,000 highly variable genes with at least 20 UMI for both spliced and unspliced transcripts across all cells.

**Expression signature analysis.** The gene sets GOBP_response_to_interferon_alpha, GOBP_response_to_interferon_gamma, GOBP_response_to_tumor_necrosis_factor and GOBP_neutrophil_migration were retrieved from the Molecular Signature Database (http://www.gsea-msigdb.org)[70] and the Scanpy function scanpy.tl.score_genes() was used to score the signature for each cell.

### CITE-seq data processing

**Demultiplexing and doublet removal of PBMC samples.** For pooled donor CITE-seq samples, the donor ID of each cell was determined by genotype-based demultiplexing using souporcell (v.2)[71]. Souporcell analyses were performed with skip_remap enabled and a set of known donor genotypes given under the common_variants parameter. The donor ID of each souporcell genotype cluster was annotated by comparing each souporcell genotype to the set of known genotypes. Droplets that contained more than one genotype according to souporcell were flagged as 'ground-truth' doublets for heterotypic doublet identification. Ground-truth doublets were used by DoubletFinder (v.2.0.3)[72] to empirically determine an optimal pK value for doublet detection. DoubletFinder analysis was performed on each sample separately using 10 principal components, a pN value of 0.25, and the nExp parameter estimated from the fraction of ground-truth doublets and the number of pooled donors.

**CITE-seq background and ambient RNA subtraction.** Background and non-specific staining by the antibodies used in CITE-seq was estimated using SoupX (v.1.4.8)[61], which models the background signal on near-empty droplets. The soupQuantile and tfidfMin parameters were set to 0.25 and 0.2, respectively, and lowered by decrements of 0.05 until the contamination fraction was calculated using the autoEstCont function. Gene expression data were also corrected using SoupX to remove cell-free mRNA contamination using the default SoupX parameters.

**CITE-seq quality control and normalization.** CITE-seq data were filtered by removing droplets with fewer than 200 genes expressed or with more than 10% of the counts originating from mitochondrial genes. Gene expression data were normalized with a log + 1 transformation (log1p), and 2,000 hyper variable genes were selected using the vst algorithm in Seurat (v.3.9.9.9024)[73]. Antibody-derived tag counts were normalized with the centred log-ratio transformation.

**Integrated embedding and clustering of CITE-seq data.** PCA was run separately on gene expression and antibody-derived tag count data, followed by batch correction using harmony[74] on the sequencing library identifier. Nearest neighbour graphs and UMAP visualizations were generated based on the first 30 harmony-adjusted principal components. The first 30 harmony-adjusted principal components of both gene expression and antibody-derived tag count data were used to compute a weighted nearest neighbour graph[75] with Seurat and embedded using UMAP. Cells were clustered with the Leiden algorithm using the igraph R package, with a resolution of 4. After initial clustering of all PBMCs, subsets of all T and natural killer cells, all B and plasma cells, and all monocytes and dendritic cells were reclustered after hypervariable gene selection within each subset. Cells in weighted-nearest-neighbour-based clusters with less than 100 members were reassigned on the basis of the closest multimodal neighbour.

**Comparison of PBMCs using Azimuth.** The manual blood cell type annotation was validated using the Azimuth tool (https://azimuth. hubmapconsortium.org). A randomly sampled subset of 100,000 PBMCs were uploaded to predict their cell type identity.

**Differential expression analysis in airway data.** In addition to the differential expression analysis, correcting for various metadata, that was performed on the whole-airway and PBMC datasets as described below, results shown for subsets of the data were obtained with a simpler method. After subsetting cell types and/or age groups, a Wilcoxon rank-sum test (implemented in Scanpy[60]) was performed to compare gene groups. The sets of differentially expressed genes were further analysed using the g:Profiler toolkit[76] (g:Profiler version e102_eg49_p15_7a9b4d6, database updated on 15 December 2020) for functional enrichment analysis. The expression of SARS-CoV-2 viral entry factors, including ACE2 and secondary entry receptors (*NRP1* (refs. [77,78]), *BSG*[79], *TFRC*[80]), along with other viral-entry-associated factors, were analysed in each cell type (Extended Data Fig. 4a).

**Defining the interferon-stimulated signature in blood.** The genes that make up the interferon-stimulated signature in blood were defined by performing Wilcoxon rank-sum tests in Seurat between each interferon-stimulated subpopulation and its matched unstimulated population. The genes that were most significant (false-discovery rate not distinguishable from 0) in all comparisons were included in the interferon-stimulated signature shown in Fig. 3. This list includes *BST2*, *CMPK2*, *EIF2AK2*, *EPSTI1*, *HERC5*, *IFI35*, *IFI44L*, *IFI6*, *IFIT3*, *ISG15*, *LY6E*, *MX1*, *MX2*, *OAS1*, *OAS2*, *PARP9*, *PLSCR1*, *SAMD9*, *SAMD9L*, *SP110*, *STAT1*, *TRIM22*, *UBE2L6*, *XAF1* and *IRF7*.

**Inference of ethnicity from single-cell RNA-seq data.** The latest biallelic single-nucleotide polymorphism (SNP) genotype data (GRCh38) was obtained from the 1000 Genomes Project (ftp://ftp.1000genomes. ebi.ac.uk/vol1/ftp/data_collections/1000_genomes_project/release/20181203_biallelic_SNV/). Allele-specific counts of RNA-seq reads at the SNP location in 1000 Genomes Project data were generated for each airway sample. As the read coverage from the single-cell RNA-seq data was strongly enriched around the 5' end of a gene, SNP loci covered at least 20 reads for more than 90% of samples that were used (19,733 genome-wide SNP loci in total). The SNP genotype from allele-specific

expression was determined as a maximum posterior genotype after fitting a beta-binomial mixture distribution with underlying probabilities of 0.01, 0.5 and 0.99 for reference homozygote, heterozygote and alternative homozygote, respectively. The overdispersion parameter of the beta-binomial distribution was estimated for each sample independently shared across all SNPs. The genotype data from 1000 Genomes samples were combined with the genotype data for our samples, and PCA was performed on the scaled genotype data (mean 0 and s.d. equal to 1 for each SNP locus). The ethnicity of each sample was determined by the Mahalanobis distance to the four major ethnic groups in the 1000 Genomes Project (African, East Asian, European and South Asian). The first three principal components were used to compute the cluster centre and the covariance matrix for each ethnic group.

**Cell type composition analysis.** The number of cells for each sample and cell type combination was modelled with a generalized linear mixed model with a Poisson outcome. The five clinical factors (age, sex, inferred ethnicity, tissue and the interaction of COVID-19 status and broad age group) and three technical factors (donor, 10x kit, sequencing batch and sample) were fitted as random effects to overcome the collinearity among the factors. The effect of each clinical/technical factor on cell type composition was estimated by the interaction term with the cell type. The glmer function in the lme4 package implemented on R was used to fit the model. The standard error of the variance parameter for each factor was estimated using the numDeriv package. The conditional distribution of the fold change estimate of a level of each factor was obtained using the ranef function in the lme4 package. The log-transformed fold change is relative to the grand mean and adjusted such that it becomes 0 when there is no effect. The statistical significance of the fold change estimate was measured by the local true sign rate (LTSR), which is the probability that the estimated direction of the effect is true, that is, the probability that the true log-transformed fold change is greater than 0 if the estimated mean is positive (or less than 0 if the estimated mean is negative). It is calculated on the basis of the estimated mean and s.d. of the distribution of the effect (log-transformed fold change), which is to an extent similar to performing a (one-sided) one-sample $Z$-test and showing $(1 - P)$.

**Differential expression analysis using metadata.** We performed differential gene expression analysis for both airway and PBMC data. We used the 7 clinical (donor, age group, sex, ethnicity, tissue, smoking status and COVID-19 status) and the 4 technical factors (batch, 10x kit version, the number of expressed genes and the number of mapped fragments) to adjust for confounding effects. For PBMC data, the tissue and 10x kit were identical across samples and not included in the model. We used the linear mixed model proposed in ref. [81] to adjust for the 11 confounding factor effects and the effect of cell type as a random effect in differential expression analysis. We fit the model on a gene-by-gene basis using the estimated variance parameters to test each factor $k$ explaining a significant amount of transcription variation. If the focal factor $k$ is a categorical variable with $L$ levels (for example, COVID-19 status with 3 levels), we partitioned the levels into one of two groups. There are $2L - 1$ contrasts that were tested against the null model (removing the focal factor $k$ in the model) to compute Bayes factors. Those Bayes factors were next used for fitting a finite mixture model to compute the posterior probability as well as the LTSR (see section 1.3 of the supplementary note of ref. [81] for more details). We used g:Profiler 2 implemented in R (v.2.0.1.5) to identify which pathways are enriched for differentially expressed genes for each contrast. We used genes of which the LTSR is greater than 0.5 to perform the analysis (both upregulated and downregulated genes separately).

**Single-cell VDJ-sequencing data analysis.** TCR and BCR sequencing data were processed using Cell Ranger and downstream analysis was performed using the scirpy package (v.0.6.1)[82]. In brief, we integrated

TCR and BCR data with gene expression from T cell and B cell subsets, respectively. After categorizing cells on the basis of the detection of productive antigen receptor chains, we selected cells with a single pair of productive chains for further analysis. T cell clonotypes were defined at the amino acid level, considering both receptor chains. B cell clonotypes were defined at the amino acid level while allowing for a Hamming distance of up to 10% of the sequence, considering both receptor chains.

## Reporting summary

Further information on research design is available in the Nature Research Reporting Summary linked to this paper.

## Data availability

The dataset from our study can be explored interactively through a web portal (https://covid19cellatlas.org). Quality control metrics for our single-cell data are provided at the web portal page. The data object, as a h5ad file, can also be downloaded from the portal page. The UK dataset is available at the European Genome–Phenome Archive under accession number EGAD00001007718. Counts matrices from bronchial brushings obtained from patients at Northwestern Memorial Hospital, Chicago, are available at the Gene Expression Omnibus under accession number GSE168215. As data are from living patients, these data are available under managed data access.

## Code availability

All data analysis scripts are available at GitHub (https://github.com/Teichlab/COVID-19paed).

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

**Acknowledgements** We acknowledge assistance from L. Thorne, P. S. Chia, R. Hynds, J. Eliasova, D. King, M. Heightmann, M. Marks, M. Avari, T. Mistry, M. Shaw-Taylor, R. Pereira, J. Machta, J. Lim, R. Prendecki, C. Frauenfelder, J. Rudd, A. Hall and the staff at the Sanger Institute Core Sequencing facility. We thank R. Jenner and the staff at the UCLH/UCL Biomedical Research Centre for the use of their 10x Chromium controller. We acknowledge funding from Wellcome (WT211276/Z/18/Z and Sanger core grant WT206194). M.Z.N., S.M.J. and K.B.M. have been funded by the Rosetrees Trust (M944, M35-F2) and from Action Medical Research (GN2911). This project has been made possible in part by grants 2017-174169 and 2019-202654 from the Chan Zuckerberg Foundation and has received funding from the European Union's Horizon 2020 Research and Innovation programme under grant agreement no. 874656. M.Z.N. acknowledges funding from the Rutherford Fund Fellowship allocated by the MRC, and M.Z.N. and S.M.J. from the UK Regenerative Medicine Platform 2 (MR/5005579/1), the Longfonds BREATH consortium and University College London Hospitals Biomedical Research Centre. M.Y. is funded by The Jikei University School of Medicine. K.B.W. acknowledges funding from University College London, Birkbeck MRC Doctoral Training Programme. C.M.S. and M.Z.N. acknowledge support from BBSRC (BB/V006738/1). We acknowledge support from the NIHR Great Ormond Street Biomedical Research Centre and the Great Ormond Street Hospital Children's Charity. S.S. was supported by a Japan Society for the Promotion of Science Overseas Fellowship (310072). R.G.W. was supported by NIH grant U19AI135964 and a GlaxoSmithKline Distinguished Scholar in Respiratory Health grant from the CHEST Foundation. A.V.M. was supported by NIH grant U19AI135964. This publication is part of the Human Cell Atlas (www.humancellatlas.org/publications/).

**Author contributions** M.Z.N. and K.B.M. conceived, set up and directed the study. C.R.B., E.K., A.d.W., B.C.J., A. Saleh, A. Saigal, H.Y., S.M.J., S.S., P.M., N.J.S., P.d.C., V.C., J.M.C., C.C., A.I. and M.Z.N. recruited patients, and collected samples (where applicable also through bronchoscopies) and clinical metadata. K.B.W. and M.Y. assisted with sample and metadata collection, isolated PBMCs and performed single-cell isolation of nasal, tracheal and bronchial brushings. K.B.W. and M.Y. performed 10x and CITE-seq and isolated DNA for genotyping. J.A.-H. collected samples, performed single-cell isolation and 10x (including CITE-seq) on post-COVID-19 samples. J.L.B. and I.T.H. helped with the study set-up, CITE-seq and isolated DNA for genotyping. L.M., L.B. and L.R. prepared sequencing libraries and conducted the sequencing. E.P. coordinated sample shipment and metadata collection. N.H., R.G.H.L., N.K., C.D.C., E.M., K.P., J.P.P. and J.W. performed bioinformatics analysis. M.Z.N., K.B.M., K.B.W., M.Y., R.G.H.L., N.H., N.K., C.D.C., E.M. and W.S. interpreted the data. K.P. facilitated online data hosting. G.R. and M.H. provided help with PBMC annotation. N.J.S., B.C.J. and S.S. provided stored healthy paediatric control nasal tissue blocks. M.C., A.W.-C., G.S.B. and M.R.C. performed experiments to collect and stain biopsies. F.J.C.-N. and B.G. designed the CITE-seq panel and advised on CITE-seq experimental design. K.B.M., M.Z.N., K.B.W., M.Y., R.G.H.L. and N.H. wrote the manuscript. E.M., S.A.T., B.G., W.S., S.M.J., J.P.P. and L.M. edited the manuscript. C.M.S., C.O., P.d.C., S.M.J. and C.R.B. provided support through ethics and patient recruitment. C.J. and A.-K.R. provided support in setting up and training for all CL3 work. For the Northwestern University (bronchial) samples, A.C.A., C.A.G., G.R.S.B., J.E.D., R.G.W., S.B.S. and T.A.P. performed bronchoscopies, collection of bronchial brushings and curation of clinical metadata. A.V.M. performed sample processing and analysis. H.K.D. obtained informed consent and coordinated sample collection. N.S.M. performed analysis. Z.L. performed sample processing and library construction.

**Competing interests** In the past three years, S.A.T. has worked as a consultant for Genentech, Roche and Transition Bio, and is a remunerated member of the Scientific Advisory Boards of Qiagen, GlaxoSmithKline and Foresite Labs and an equity holder of Transition Bio. P.M. is a Medical Research Council-GlaxoSmithKline (MRC-GSK) Experimental Medicine Initiative to Explore New Therapies (EMINENT) clinical training fellow with project funding, has served on an advisory board for SOBI, outside the submitted work, and receives co-funding by the National Institute for Health Research (NIHR) University College London Hospitals Biomedical Research Centre (UCLH BRC).

**Additional information**
**Correspondence and requests for materials** should be addressed to Marko Z. Nikolić or Kerstin B. Meyer.

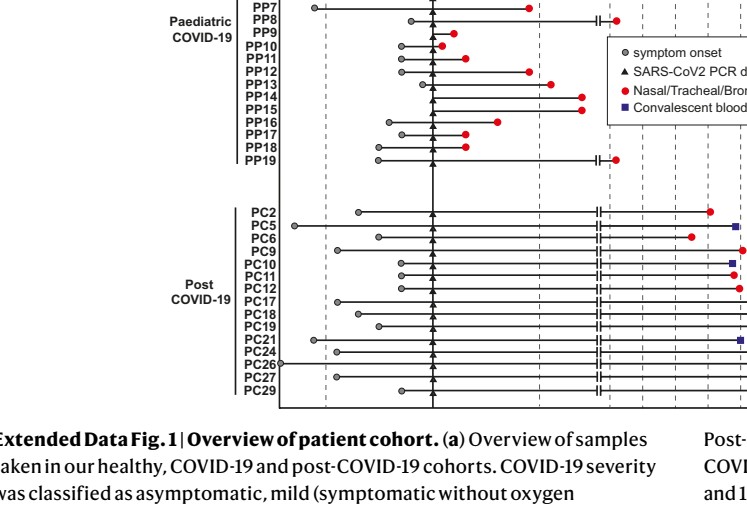

**Extended Data Fig. 1 | Overview of patient cohort. (a)** Overview of samples taken in our healthy, COVID-19 and post-COVID-19 cohorts. COVID-19 severity was classified as asymptomatic, mild (symptomatic without oxygen requirement or respiratory support), moderate (requiring oxygen without respiratory support) or severe (requiring non-invasive or invasive ventilation). Post-COVID-19 patients were sampled 3 months after recovering from severe COVID-19. **(b)** Timeline of sample collections from COVID-19 positive (18 adults and 19 paediatric) and post-COVID-19 (13 adults and 2 paediatric) patients enrolled in our study. Sample collections are shown relative to symptom onset and a SARS-CoV-2 positive RT–qPCR test, to which all patients are aligned.

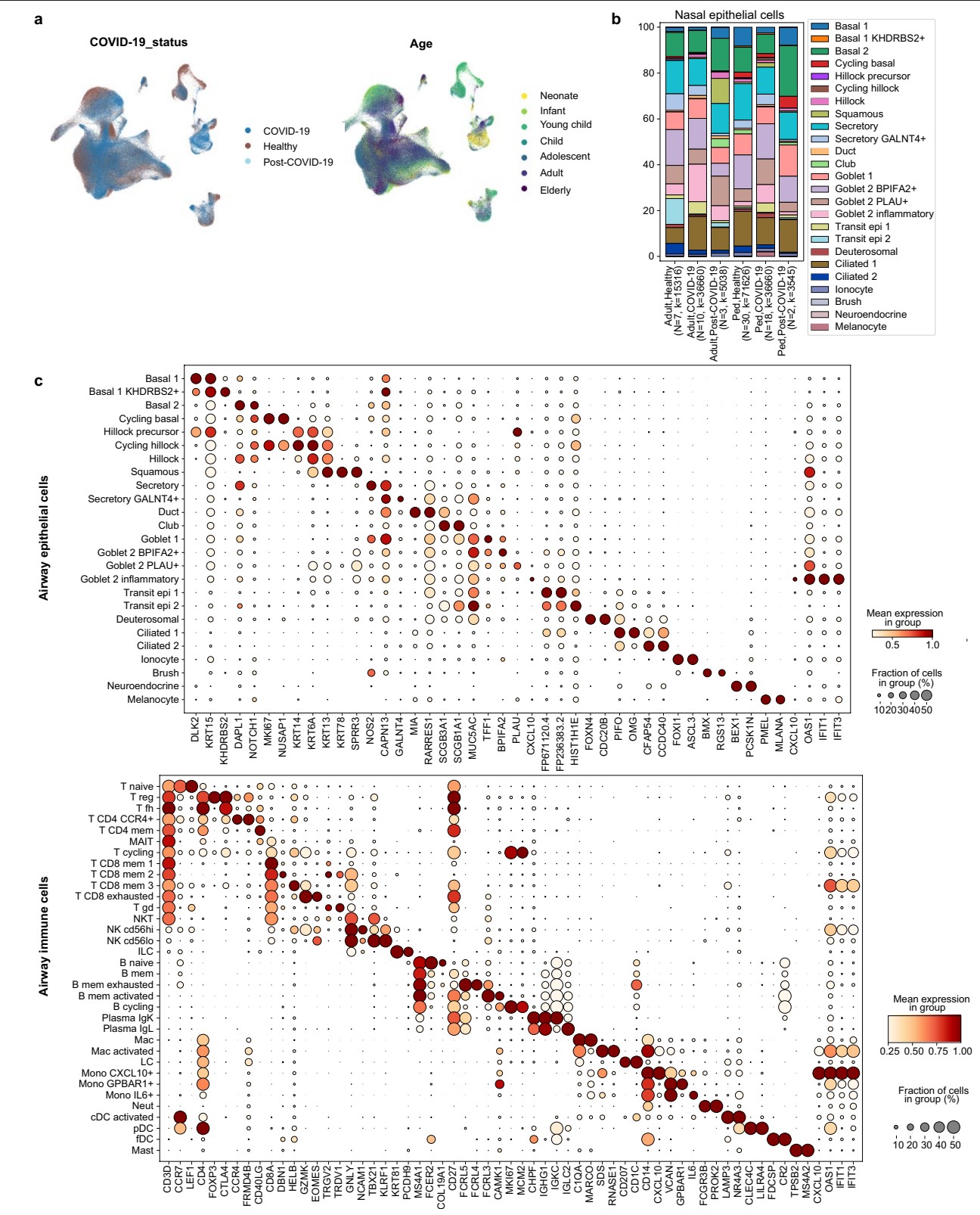

**Extended Data Fig. 2 | Airway single-cell metadata, proportions and cell type markers.** (**a**) UMAP visualization of annotated airway scRNA-seq dataset from Fig. 1b coloured by COVID-19 status and age groups. (**b**) Bar plot comparing nasal epithelial cell type compositions across COVID-19 status and age groups. (**c**) Dot plots showing marker genes for annotated airway epithelial and immune cell types, with fraction of expressing cells and average expression within each cell type indicated by dot size and colour, respectively.

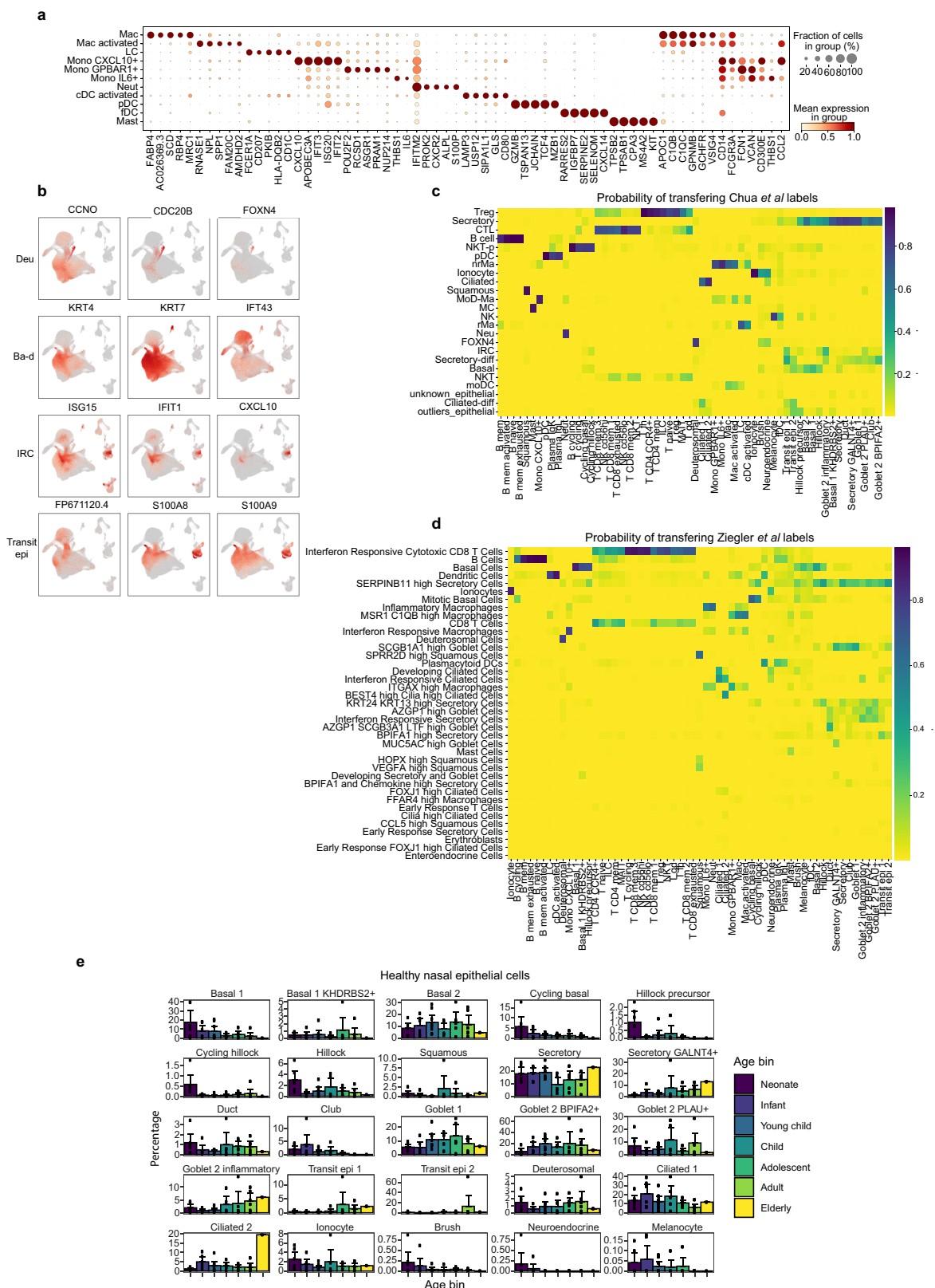

**Extended Data Fig. 3 | Supplementary information for airway cell type annotation.** (**a**) Detailed marker genes for distinct airway myeloid populations in our data set listing marker genes that are unique to each of the defined populations, whilst markers that are common to closely related myeloid cell types are shown on the right side of the panel. (**b**) Comparison of annotated cell types to published data sets. Marker genes for the three populations identified as differentiating to ciliated cells[28] and markers of transit epithelial cells (Transit epi 1 and 2). Deu; deuterosomal, Ba-d; basal differentiating, IRC; interferon responsive cell. (**c, d**) Logistic regression based label transfer for the data sets in (c) Chua et al[28] and (d) Ziegler et al[14]. (**e**) Bar chart showing changes in nasal epithelial cell type proportions observed across age within our paediatric and adult healthy cohorts. Error bars indicate two times standard error of the mean.

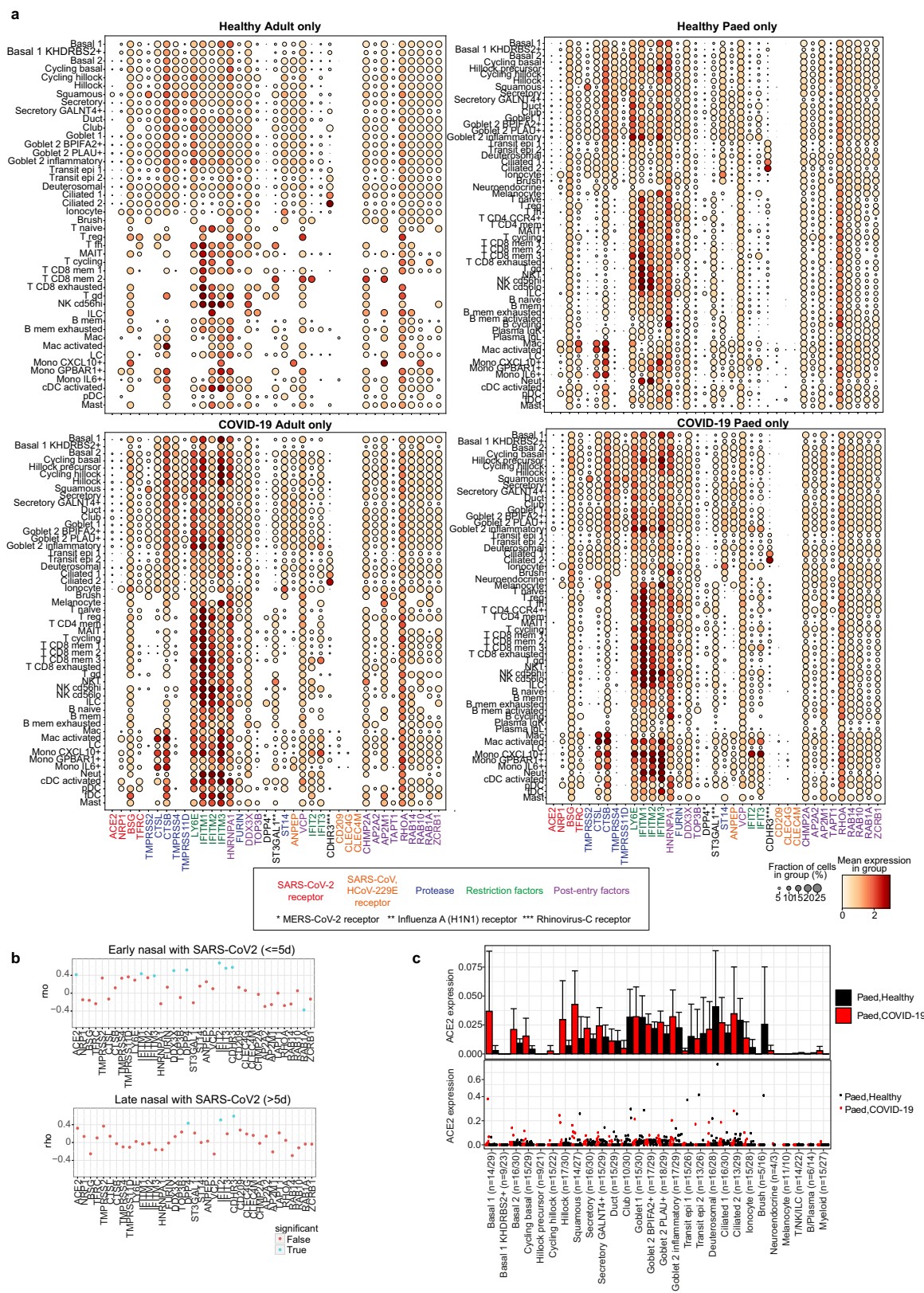

**Extended Data Fig. 4 |** See next page for caption.

**Extended Data Fig. 4 | Expression of viral entry-associated genes in the airways.** (**a**) Dot plots showing cell type expression of viral entry-associated genes within the upper airways of healthy adults (n = 7), healthy children (n = 30), COVID-19 adults (n = 10) and COVID-19 children (n = 18) respectively, included genes linked to SARS-CoV-2, SARS-CoV, MERS-CoV, Rhinovirus-C and Influenza A infections. The fraction of expressing cells and average expression within each cell type is indicated by dot size and colour, respectively. (**b**) Spearman correlation between the fraction of cells with detected viral RNA and the average expression of entry factors, as in (a), across cell types within the airways of COVID-19 patients samples (with viral reads ≥ 5) within 5 days of a positive SARS-CoV-2 qPCR test (Early) and those sampled longer than 5 days prior to onset of symptoms or positive SARS-CoV-2 qPCR test, whichever was longer (Late). Dots in blue indicate $p < 0.05$. (**c**) Expression of ACE2 in paediatric airway cells in each cell type averaged by donor (upper) and in each donor (lower) and coloured by COVID-19 status. Error bars indicate two times standard error of the mean across donors. Numbers in brackets indicate numbers of COVID-19 donors/healthy donors.

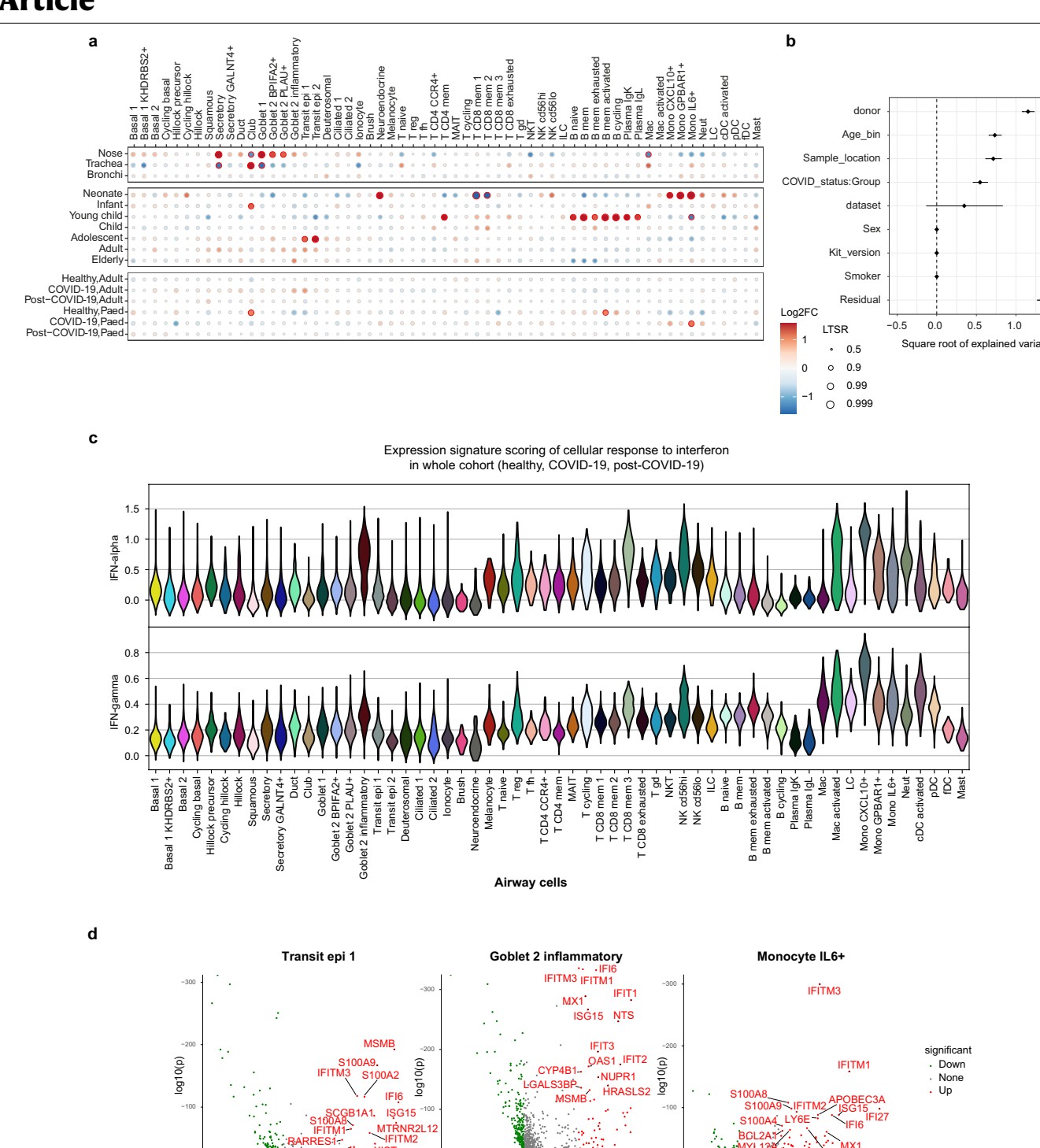

**Extended Data Fig. 5 | Airway cell type proportion analysis, interferon responses and differential gene expression.** (**a**) Dot plot showing fold change and statistical significance of all airway cell type proportions across location of sampling, age group and COVID-19 status, respectively, estimated by fitting Poisson generalized linear mixed models taking into account other technical and biological variables (see Methods). (**b**) Feature importance plot depicting the variance accounted for by each of the clinical and technical factors in our statistical analysis of cell type proportions within our airway scRNA-seq dataset. Factors were donor (patient), patients age (Age_bin), sample location (nasal, tracheal, bronchial), COVID-19 status group (COVID-19 positive, negative or post-COVID-19), dataset (UK cohort or Chicago Cohort) sex, 10x chromium 5′ single-cell sequencing kit version (kit_version) smoking status

(non-smoker, ex-smoke or current), date and other factors (residual). Note: Error bars were not able to be generated for sex, Kit_version and smoker. 97 samples contributed to the estimation of variances and their standard errors. (**c**) Response to interferon by airway cell type. Scores of GO term gene signatures for the terms: response to type 1 interferon (GO:0035455 or GO:0034340) and interferon-gamma (GO:0034341) across cell types. Scores were calculated with Scanpy as the average expression of the signature genes subtracted with the average expression of randomly selected genes from bins of corresponding expression values. (**d**) Differential gene expression contrasting COVID-19 and non-COVID-19 samples in transit epithelial 1 cells, inflammatory goblet 2 cells, and mono IL-6 cells.

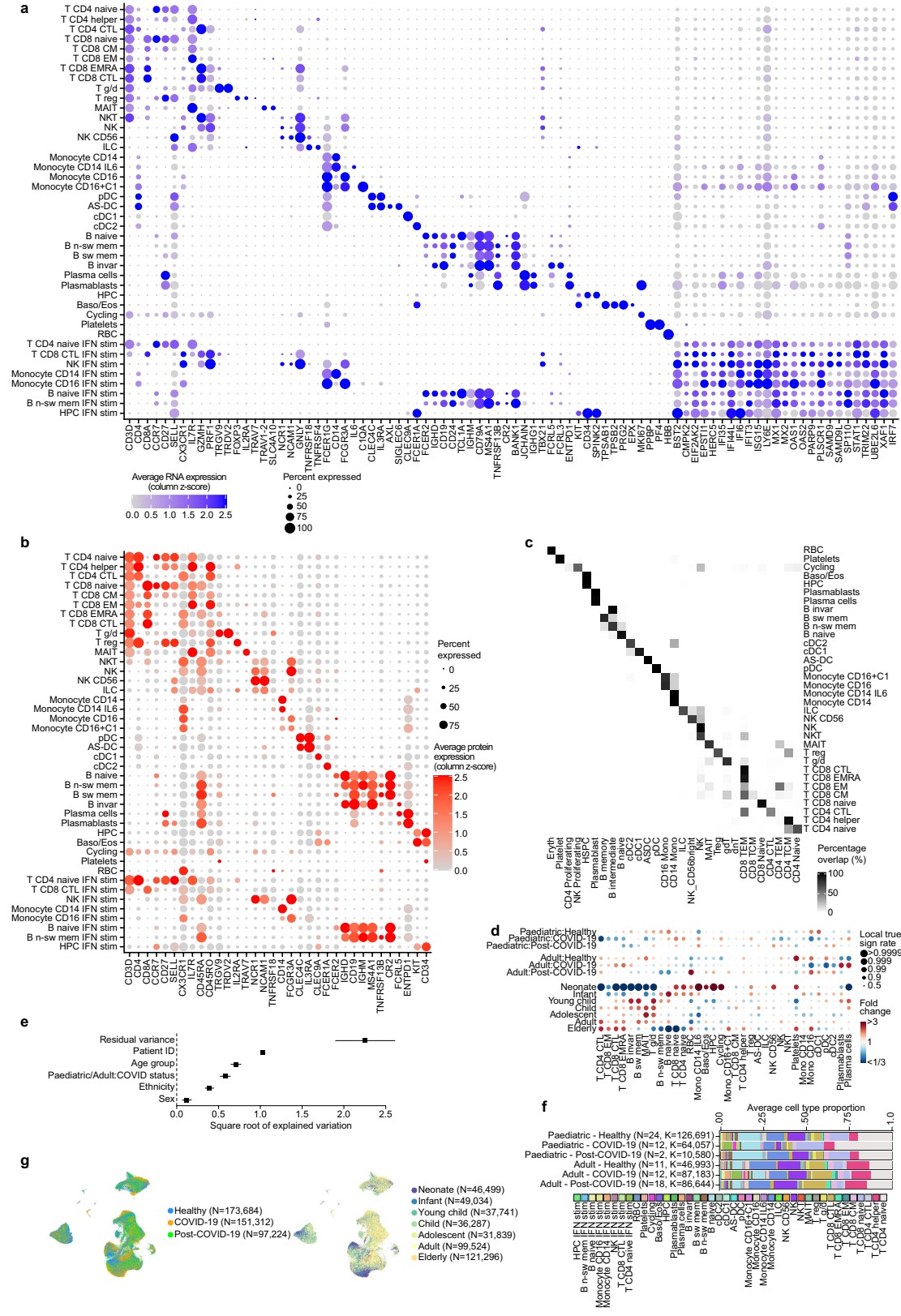

**Extended Data Fig. 6** | See next page for caption.

**Extended Data Fig. 6 | Expression of cell type markers and immune compartment dynamics.** (**a**) Expanded dot plot from Fig. 3d showing the RNA expression of cell type marker genes and interferon-stimulated genes. (**b**) Dotplot showing the cell surface protein expression of cell type marker proteins. In both **a** and **b** the size of the dot is scaled to the percentage of cells that have at least one count for each gene or protein, and the colour is scaled to the z-score normalized expression of each gene or protein. (**c**) Comparison of our manual cell type PBMC annotation vs an automated annotation performed by Azimuth. (**d**) Fold changes of immune cell type proportions across age group and disease status. Age and disease specific changes were deconvoluted by fitting Poisson generalized linear mixed models taking into account other confounders such as sex and ethnicity. (**e**) Feature importance plot showing the variance that can be explained by the different features that were included in the Poisson linear mixed model that was fitted on the cell type proportions in the PBMC data. 80 samples contributed to the estimation of variances and their standard errors. (**f**) Bar plots showing the average immune cell proportions in PBMC samples. Cell types are colour coded and grouped based on their age group and disease status. N denotes the amount of samples in each group, while K denotes the amount of cells per group. (**g**) UMAPs as in Fig. 3a in which the COVID-19 status (left panel) and the age group (right panel) is visualized for each cell.

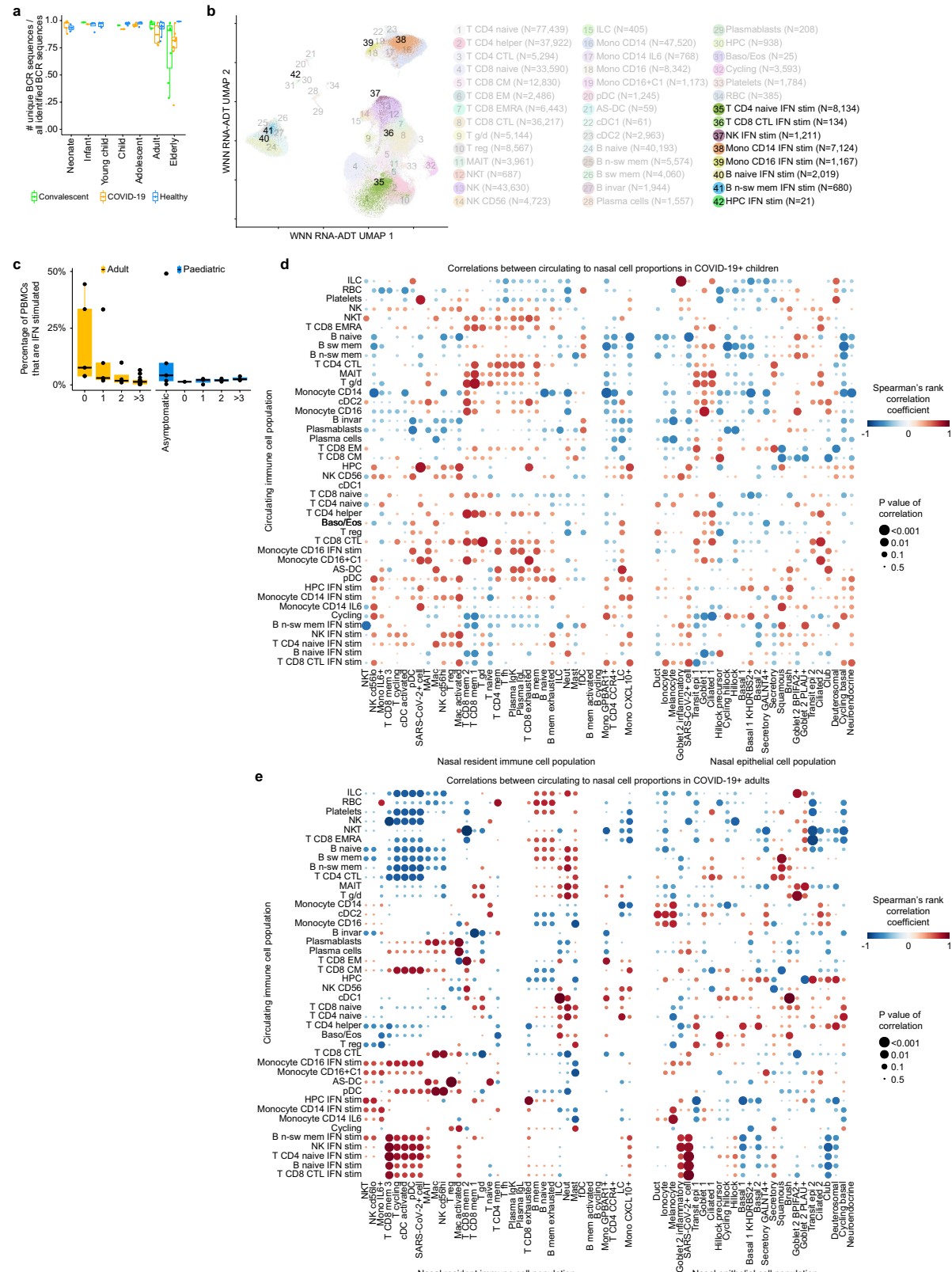

**Extended Data Fig. 7** | See next page for caption.

**Extended Data Fig. 7 | Immune cell population dynamics.** (**a**) Fractions of unique BCR sequences show the differences in immune repertoire diversity over age and disease. (**b**) UMAP visualization as in Fig. 3a showing the annotated interferon-stimulated subpopulations in clusters 35–42. (**c**) Boxplot showing the percentage of PBMCs that are interferon-stimulated in asymptomatic or symptomatic COVID-19 patients, grouped by the weeks since the onset of symptoms, and separated for adults (left) and children (right). (**d**) Dotplot of Spearman correlations between nasal and blood cell type proportions in paediatric COVID-19 patients and (**e**) in adult COVID-19 patients. In both **d** and **e**, cell type proportions in the nose (x-axis) are compared to the blood (y-axis). Correlations shown in Fig. 3g present a zoom in of the adult panel. Rows and columns in both dotplots are clustered by hierarchical clustering on the combined matrices. The size of the dots is scaled by the significance of each correlation. Colour is scaled by the Spearman rank-correlation coefficient. If a blood - nose cell type combination shows a positive correlation, this is indicative that if the blood cell type changes in proportion, the nasal cell type changes accordingly, and vice versa. Dots in **a** and **c** represent independent patient samples. Box plots were drawn with the centre line as the median of the data distribution, the hinges as the first and third quartiles, and with the whiskers extending to the lowest and highest values that were within $1.5 \times$ interquartile range of the upper or the lower hinge.

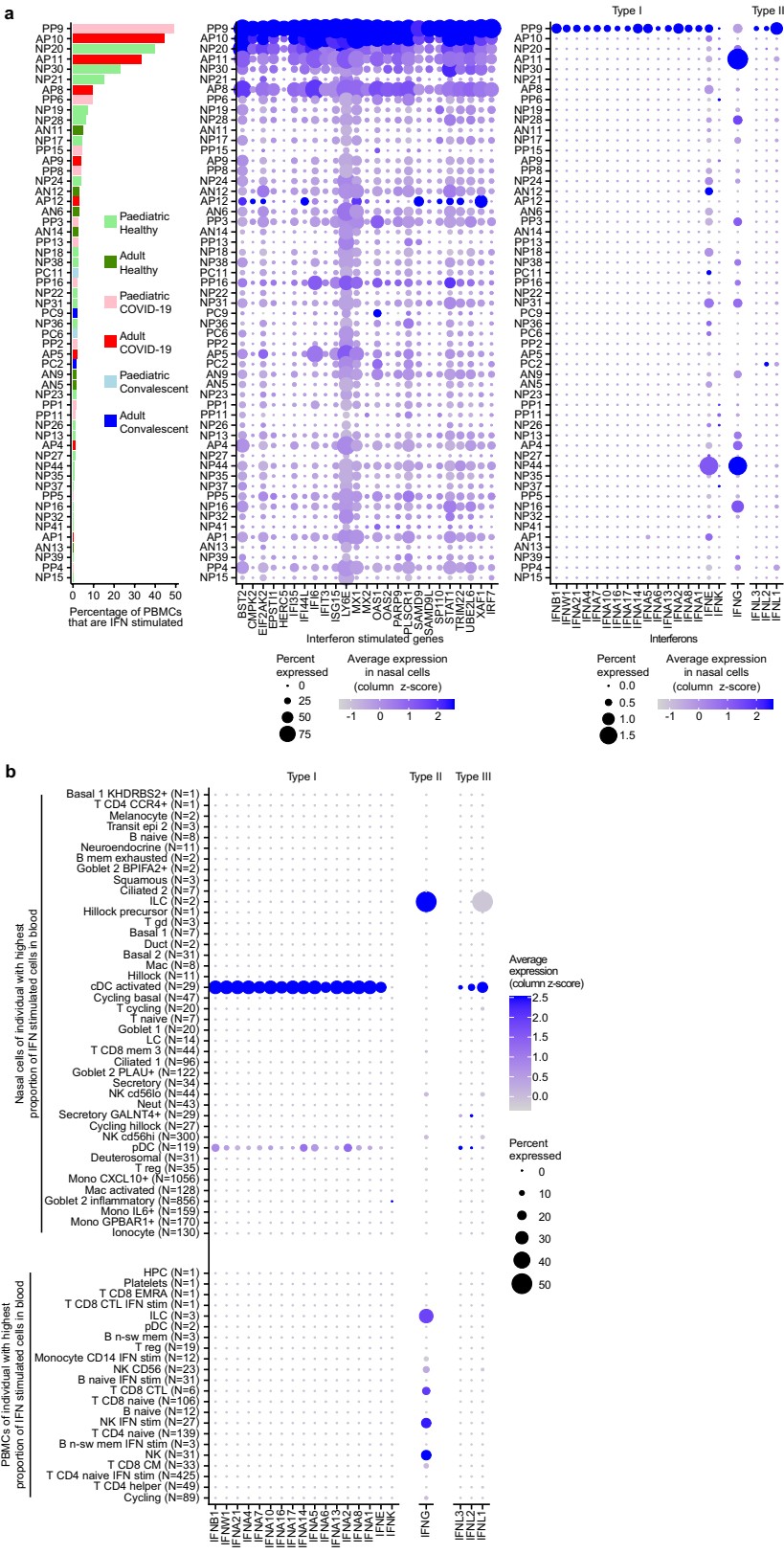

**Extended Data Fig. 8 | Interferon expression in COVID-19 patient with highest amount of interferon-stimulated blood cells. (a)** Ranked barplot and matched dotplots as in Fig. 3h, but showing the expression of all genes that make up the interferon-stimulated gene signature (middle) and the expression of all interferons (right) in all cells, instead of averaged signatures gene expression signatures in specific cell types. **(b)** Dotplot related to Fig. 3h

showing the expression of all interferons in all nasal resident (top) and circulating (bottom) cell types that were present in this individual. The size of the dot is scaled to the percentage of cells that have at least one count for each gene or protein, and the colour is scaled to the z-score normalized expression of each gene or protein.

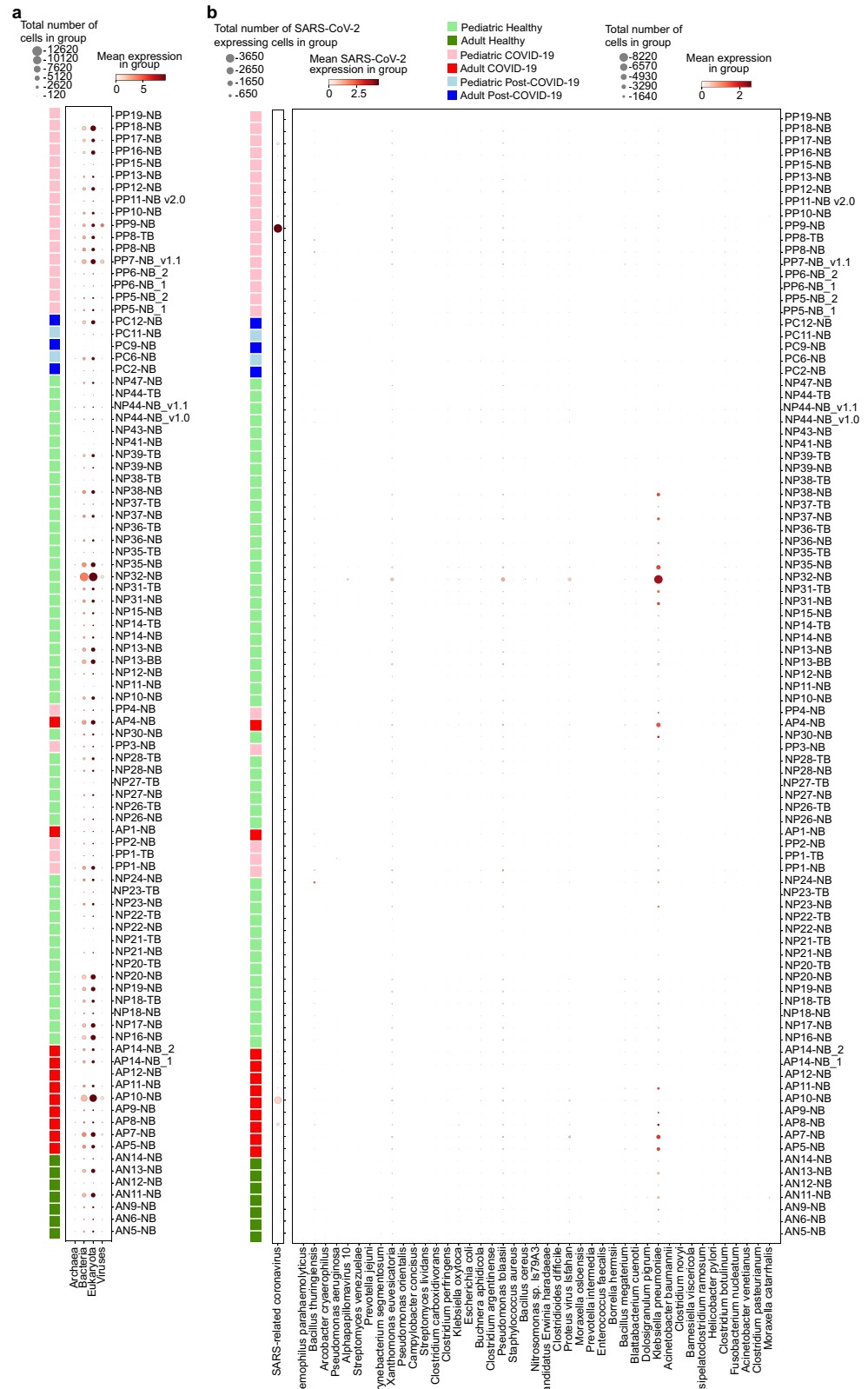

**Extended Data Fig. 9 | Metagenomic analysis of patient sample reads that were not mapped to the human genome.** (**a**) Dotplot showing the amount of cells that harbour reads aligned to archaea, bacteria, eukaryota (including human reads that initially did not align to the human transcriptome by STARsolo) and viruses. (**b**) Dotplot showing the amount of cells that harbour reads to a selection of disease-relevant bacteria and viruses. Apart from SARS-CoV-2 and non-specific signal found in most samples, we did not detect any pathogens that were highly abundant in samples of interest.

# Extended Data Table 1 | Summary of patient metadata

| | SARS-CoV-2 negative children (n = 30) | SARS-CoV-2 negative adults (n = 11) | SARS-CoV-2 positive children (n = 19) | SARS-CoV-2 positive adults (n = 18) | Post-COVID19 (n = 15) |
|---|---|---|---|---|---|
| **Median age** | 2.87 (3 days - 16 yrs) | 43 (26 - 67 yrs) | 4 (3 days - 16 yrs) | 66 (25 - 92 yrs) | 47 (4 months - 82 yrs) |
| **Sex** | | | | | |
| Male (%) | 13 (43.3) | 4 (36.4) | 14 (73.7) | 10 (55.6) | 12 (80.0) |
| Female (%) | 17 (56.7) | 7 (63.6) | 5 (26.3) | 8 (44.4) | 3 (20.0) |
| **Ethnicity** | | | | | |
| White (%) | 20 (66.7) | 8 (72.7) | 10 (52.6) | 7 (38.9) | 10 (66.7) |
| Black (%) | 4 (13.3) | 1 (9.1) | 3 (15.8) | 1 (5.6) | 0 (0) |
| Hispanic (%) | 0 (0) | 0 (0) | 0 (0) | 3 (16.7) | 0 (0) |
| South Asian (%) | 1 (3.3) | 1 (9.1) | 0 (0) | 5 (27.8) | 1 (6.7) |
| Middle Eastern or Central Asian (%) | 1 (3.3) | 0 (0) | 5 (26.3) | 0 (0) | 2 (13.3) |
| East Asian/ Pacific Islander (%) | 0 (0) | 1 (9.1) | 0 (0) | 0 (0) | 1 (6.3) |
| Other (%) | 1 (3.3) | 0 (0) | 1 (5.3) | 1 (5.6) | 0 (0) |
| Unspecified (%) | 3 (10) | 0 (0) | 0 (0) | 1 (5.6) | 1 (6.7) |
| **Peripheral blood test at sampling** | | | | | |
| Lymphocyte count (cells/µl) | 2,026 ± 684.6 | - | 3,309 ± 2,703 | 1,185 ± 504.1 | 2,163 ± 1,056 |
| Neutrophil count (cells/µl) | 7,870 ± 6,881 | - | 6,699 ± 5,918 | 6,275 ± 3,268 | 5,354 ± 1,811 |
| **Reported symptoms** | | | | | |
| Fever, Fatigue (%) | 0 (0) | 0 (0) | 9 (47.4) | 10 (55.6) | 5 (33.3) |
| Digestive symptom (%) | 2 (6.7) | 0 (0) | 8 (42.1) | 3 (16.8) | 3 (20) |
| Upper respiratory tract symptom (%) | 0 (0) | 0 (0) | 5 (26.3) | 9 (50) | 6 (40.0) |
| Respiratory failure (%) | 0 (0) | 0 (0) | 6 (31.6) | 12 (66.7) | 13 (86.7) |
| Other (%) | 0 (0) | 0 (0) | 3 (15.8) | 0 (0) | 0 (0) |
| **Respiratory Support** | | | | | |
| None (%) | - | - | 9 (47.4) | 3 (16.7) | 0 (0) |
| Low flow oxygen (%) | - | - | 1 (5.3) | 4 (22.2) | 0 (0) |
| HFNC / NIPPV (%) | - | - | 2 (10.5) | 4 (22.2) | 7 (46.7) |
| IMV (%) | - | - | 7 (36.8) | 7 (38.9) | 8 (53.3) |
| **COVID-19 severity** | | | | | |
| Asymptomatic (%) | NA | NA | 5 (26.3) | 0 (0) | 0 (0) |
| Mild (%) | NA | NA | 4 (21.1) | 5 (27.8) | 0 (0) |
| Moderate (%) | NA | NA | 1 (5.3) | 3 (16.7) | 0 (0) |
| Severe (%) | NA | NA | 9 (47.4) | 10 (55.5) | 15 (100) |
| **MIS-C** | NA | NA | 2 (10.5) | 0 (0) | 0 (0) |
| **Detected Co-infection** | | - | | | |
| Bacterial (%) | 0 (0) | - | 1 (5.3) | 1 (5.7) | 1 (6.7) |
| Viral (%) | 0 (0) | - | 1 (5.3) | 0 (0) | 0 (0) |
| Fungal (%) | 0 (0) | - | 0 (0) | 1 (5.7) | 0 (0) |
| Mulitple (%) | 0 (0) | - | 0 (0) | 1 (5.7) | 2 (13.3) |
| | | | | | 0 (0) |
| **COVID-19 Treatment** (Prior to sample collection) | NA | NA | 3 (15.8) | 0 (0) | - |

Patients were divided into columns according to COVID-19 status. Metadata on median age, sex, ethnicity, peripheral blood counts at the time of sampling, reported symptoms, respiratory support, COVID-19 severity, diagnosis of multisystem inflammatory syndrome in children (MIS-C), detected co-infection and specific anti-COVID-19 treatment prior to sampling, are shown. Abbreviations: HFNC = high flow nasal cannula, NIPPV = non-invasive positive pressure ventilation, IMV = invasive mechanical ventilation, NA = not assessed.

# Reporting Summary

## Statistics

For all statistical analyses, confirm that the following items are present in the figure legend, table legend, main text, or Methods section.

| n/a | Confirmed | |
|---|---|---|
| ☐ | ☒ | The exact sample size (*n*) for each experimental group/condition, given as a discrete number and unit of measurement |
| ☐ | ☒ | A statement on whether measurements were taken from distinct samples or whether the same sample was measured repeatedly |
| ☐ | ☒ | The statistical test(s) used AND whether they are one- or two-sided *Only common tests should be described solely by name; describe more complex techniques in the Methods section.* |
| ☐ | ☒ | A description of all covariates tested |
| ☐ | ☒ | A description of any assumptions or corrections, such as tests of normality and adjustment for multiple comparisons |
| ☐ | ☒ | A full description of the statistical parameters including central tendency (e.g. means) or other basic estimates (e.g. regression coefficient) AND variation (e.g. standard deviation) or associated estimates of uncertainty (e.g. confidence intervals) |
| ☐ | ☒ | For null hypothesis testing, the test statistic (e.g. *F*, *t*, *r*) with confidence intervals, effect sizes, degrees of freedom and *P* value noted *Give P values as exact values whenever suitable.* |
| ☐ | ☒ | For Bayesian analysis, information on the choice of priors and Markov chain Monte Carlo settings |
| ☐ | ☒ | For hierarchical and complex designs, identification of the appropriate level for tests and full reporting of outcomes |
| ☐ | ☒ | Estimates of effect sizes (e.g. Cohen's *d*, Pearson's *r*), indicating how they were calculated |

*Our web collection on statistics for biologists contains articles on many of the points above.*

## Software and code

Policy information about availability of computer code

| Data collection | No specific code was used in the data collection |
|---|---|
| Data analysis | The following open access algorithms were used in the data analysis. Azimuth bbknn 1.3.12 bedtools v.2.30 Cell Ranger 3.0.2 EmptyDrops g:profiler toolkit Harmony Kraken 2 Scanpy 1.6.0 Scirpyy Scrublet 0.2.1 Scanpy 1.6.0 scvelo 0.2.2 Seuratt SoupX 1.5.0 and 1.4.8 as specified in methods SouporCell STARsolo functionality of STAR 2.7.3 All data analysis scripts are available on https://github.com/Teichlab/COVID-19paed. |

For manuscripts utilizing custom algorithms or software that are central to the research but not yet described in published literature, software must be made available to editors and reviewers. We strongly encourage code deposition in a community repository (e.g. GitHub). See the Nature Portfolio guidelines for submitting code & software for further information.

## Data

Policy information about availability of data

All manuscripts must include a data availability statement. This statement should provide the following information, where applicable:

- Accession codes, unique identifiers, or web links for publicly available datasets
- A description of any restrictions on data availability
- For clinical datasets or third party data, please ensure that the statement adheres to our policy

Data availability:
The data set from our study can be explored interactively through a web portal: https://covid19cellatlas.org. Quality control metrics for our single cell data can be found at the web portal page. The data object, as a h5ad file, can also be downloaded from the portal page. The UK data set is available under accession number EGAD00001007718. Counts matrices from bronchial brushings obtained from patients at Northwestern Memorial Hospital, Chicago, are available at GEO, accession number GSE168215. As data is from living patients, these data will be available under managed data access.

The EGA link is:
https://urldefense.proofpoint.com/v2/url?
u=https-3A__ega-2Darchive.org_datasets_EGAD00001007718&d=DwIDaQ&c=D7ByGjS34AllFgecYw0iC6Zq7qlm8uclZFI0SqQnqBo&r=UkvGllMAxxOrRLImmtb8_8aL9
f8dRmw6ZZOconDDoI&m=ms4g_hTiCC1177yddG023CrSlQfvZR3LHJ-3aHcbNfLqrMJ30dvc2iSSkzVsMJH2&s=yCvFfXAlnXSAk41YM7Fn2afxwbaPZxTYYJDExRQGVLA&e=

The applicant requests specific dataset access via : https://www.sanger.ac.uk/legal/DAA/MasterController

# Field-specific reporting

Please select the one below that is the best fit for your research. If you are not sure, read the appropriate sections before making your selection.

[✗] Life sciences        [ ] Behavioural & social sciences        [ ] Ecological, evolutionary & environmental sciences

For a reference copy of the document with all sections, see nature.com/documents/nr-reporting-summary-flat.pdf

# Life sciences study design

All studies must disclose on these points even when the disclosure is negative.

| Sample size | No sample size calculations were carried out. The following statement was sent to the reviewers: |
| --- | --- |
| | Due to the complexity of single-cell datasets, there are not yet any widely accepted methods available to perform power calculations for studies such as ours. However, the statistical framework that is employed to perform cell type composition analyses in this study specifically fits random effects to model any unexplained variance in a rigorous manner. |
| | Single cell sequencing is a technique that gives great in depth insight, but at high financial cost. The total number of patients enrolled in this study was 93, which is in line with or larger than comparable recent studies (see references 10, 11, 12 and 14 in the manuscript). |
| Data exclusions | All samples for which sequencing data was generated have been submitted to EGA. For the airway data set, 7 samples were excluded from analysis, out of which 1 (AP13-NB) had almost no reads at all, 4 (AN2-NB, AN3-NB, AN7-NB, PP14-NB) had too few reads, 1 (PP7-NB_v2.0) had low mapping rate, and 1 (PC21-NB) failed cell calling. For the PBMC data set, PC7 was of insufficient quality and therefore not included in the analysis. |
| Replication | All findings were based on statistical analysis of a large patient cohort. There was no replication cohort. |
| Randomization | As this was not a clinical trial, randomisation was not relevant for our study. |
| Blinding | As this was not a clinical trial, blinding was not relevant as the statistical tests were performed in a single analysis with all relevant samples included. |

# Reporting for specific materials, systems and methods

We require information from authors about some types of materials, experimental systems and methods used in many studies. Here, indicate whether each material, system or method listed is relevant to your study. If you are not sure if a list item applies to your research, read the appropriate section before selecting a response.

## Materials & experimental systems

| n/a | Involved in the study |
|---|---|
| ☐ | ☒ Antibodies |
| ☒ | ☐ Eukaryotic cell lines |
| ☒ | ☐ Palaeontology and archaeology |
| ☒ | ☐ Animals and other organisms |
| ☐ | ☒ Human research participants |
| ☐ | ☒ Clinical data |
| ☒ | ☐ Dual use research of concern |

## Methods

| n/a | Involved in the study |
|---|---|
| ☒ | ☐ ChIP-seq |
| ☒ | ☐ Flow cytometry |
| ☒ | ☐ MRI-based neuroimaging |

## Antibodies

| | |
|---|---|
| Antibodies used | anti-human S100A9 conjugated to FITC (clone: MRP14, Biolegend cat. # 350703); anti-human EpCam conjugated to APC (clone: 9C4, Biolegend cat # 324207);   192 TotalSeq-C antibodies (Biolegend, cat. # 99814). The latter was a pre-diluted commercial panel. S100A9 validation: Recombinant human S100A8 protein (Cat. No. 719902, lane 1), S100A9 protein( lane 2) and total lysates (15 µg protein) from HeLa (negative control, lane 3) , PBMC (lane 4) were resolved by 4-20% Tris-Glycine electrophoresis, transferred to nitrocellulose, and probed with 1:5000 (0.1 µg/mL) diluted purified anti-MRP-14 (S100A9) (clone A10105J). Proteins were visualized by chemiluminescence detection using 1:3000 diluted HRP anti-mouse-IgG secondary antibody (Cat. No. 405306). 1:2000 dilution of Direct-Blot™ HRP anti-β-actin antibody (clone 2F1-1, Cat. No. 643807) was used as a loading control (lower). Lane M: Molecular weight ladder. The electrophoresis gel shows clear staining in lane2 and 4, but not lane 1 and 3. validation of EpCam: Each lot of this antibody is quality control tested by immunofluorescent staining with flow cytometric analysis. For flow cytometric staining, the suggested use of this reagent is ≤ 0.5 µg per 106 cells in 100 µL volume or 100 µL of whole blood. It is recommended that the reagent be titrated for optimal performance for each application. |
| Validation | All antibodies employed were commercial antibodies. |

## Human research participants

Policy information about studies involving human research participants

| | |
|---|---|
| Population characteristics | Population characteristics are listed in Extended Data Table 1. |
| Recruitment | Recruitment of patients was in line with research ethics permissions listed below. Experienced clinicians assessed each patient and exclusion criteria noted in the methods were applied. |
| Ethics oversight | Ethical approval was given through the Living Airway Biobank, administered through UCL Great Ormond Street Institute of Child Health (REC reference: 19/NW/0171, IRAS project ID 261511, North West - Liverpool East Research Ethics Committee), REC reference 18/SC/0514 (IRAS project 245471, South Central - Hampshire B Research Ethics Committee) administered through University College London Hospitals NHS Foundation Trust and REC reference 18/EE/0150 (IRAS project ID 236570, East of England - Cambridge Central Research Ethics Committee) administered through Great Ormond Street Hospital NHS Foundation Trust, REC reference 08/H0308/267 administered through Cambridge University Hospitals NHS Foundation Trust, as well as by the local R&D departments at all hospitals. All study participants or their surrogates provided informed consent.<br><br>Ethical approval for sample collection from patients with severe pneumonia was given by Northwestern Institutional Review Board, study STU00204868 (PI Richard Wunderink). Samples from patients with COVID-19, viral pneumonia and other pneumonia, and non-pneumonia controls were collected from participants enrolled in the Successful Clinical Response in Pneumonia Therapy (SCRIPT) study STU00204868 and admitted to the ICU at Northwestern Memorial Hospital, Chicago. |

Note that full information on the approval of the study protocol must also be provided in the manuscript.

## Clinical data

Policy information about clinical studies

All manuscripts should comply with the ICMJE guidelines for publication of clinical research and a completed CONSORT checklist must be included with all submissions.

| | |
|---|---|
| Clinical trial registration | na |
| Study protocol | Note where the full trial protocol can be accessed OR if not available, explain why. |
| Data collection | Describe the settings and locales of data collection, noting the time periods of recruitment and data collection. |

Outcomes

*Describe how you pre-defined primary and secondary outcome measures and how you assessed these measures.*

