## [Peer Review File · Nature]

Manuscript Title: Local and systemic responses to SARS-CoV-2 infection in children and adults

Reviewer Comments & Author Rebuttals

Reviewer Reports on the Initial Version:

Referee #1 (Remarks to the Author):

A. Summary of the key results

Yoshida et al describe a single-cell multiomics analysis of 4 children and 18 adults infected with SARS-CoV-2 that included both PBMCs, nasal and tracheal brushings. Similar sample triads were also collected from 30 healthy children of different age, allowing comparisons of immune cell states during childhood, and in response to SARS-CoV-2 infection.

B. Originality and significance: if not novel, please include reference

The study involves state-of-the art methodologies for single cell analyses of both mRNA and proteins. Another strength is the matched samples from blood, nasal and tracheal mucosa. The main limitation of the study is the very small sample size, both from different age groups of healthy children and from pediatric COVID-19 cases. As a consequence of these very small numbers the authors fail to draw interesting novel conclusions about age-associated differences and between COVID-19 infected and healthy children and adults. Most of the reported findings involve correlates between age and individual genes and cells and sweeping statements such as "...plasma cells and gamma delta (g/d) T cells all showed a striking increase in young children (2 to 6 years of age), presumably reflecting greater antigen exposure of children in this age group, for example in pre-school nurseries". The children were healthy when sampled and plasmablast responses are typically transient in the blood but since this finding is based upon 7 blood samples collected from children between 2-6 years of age, this might be a completely random event without any biological relevance.

C. Data & methodology: validity of approach, quality of data, quality of presentation

I think in general the developmental changes in blood of healthy children are too severely underpowered to be informative. There is a great deal of variability among children and that is simply not possible to capture with 5-7 blood samples collected from different children in each of the developmental stages during childhood.

D. Appropriate use of statistics and treatment of uncertainties

There are no obvious technical errors or unwarranted uses of statistical tests but there is a general lack of statistical power due to a very small sample set overall. Especially among the healthy children there are about ~6000 cells sequenced per PBMC sample, which means that many rare immune cell types are missed, and others are represented by a handful cells or so, leading to general problems with uncertainties in all conclusions drawn. This is particularly damning given that the authors aim to perform so many individual comparisons, across tissue types, across ages and between healthy and SARS-CoV-2 infected.

E. Conclusions: robustness, validity, reliability

See D above. I worry that many of the conclusions are simply overinterpretations of random fluctuations in the data. I think the authors should perform power-analyses to first determine what conclusions are possible to draw from this extremely high-dimensional analysis performed in a small number of cells from a handful of samples per group.

F. Suggested improvements: experiments, data for possible revision

I suggest the authors focus their story on the most robust, and in my opinion most interesting aspect of the paper which are the analyses across matched tissue sample types and the

similarities/differences between blood and airway mucosa in the COVID-19 patients vs healthy children and adults.

Also, the analyses between adult and pediatric COVID-19 patients are hampered by lack of details in the sampling time from symptom onset. For example the differences seen with respect to clonal expansion in the mucosa between children and adults with COVID-19 could be explained by differences in the time from symptom onset but such information is not provided and no longitudinal samplings are done for the mucosal samples. Thus, I find it hard to assess whether these differences are of biological relevance or not.

G. References: appropriate credit to previous work?
Yes I find this appropriate.

H. Clarity and context: lucidity of abstract/summary, appropriateness of abstract, introduction and conclusions

The paper is well written, and figures are overall complex but well described. Conclusions are mostly speculative and not followed up on with additional experiments making it difficult to verify any of the conclusions made by the authors.

Referee #2 (Remarks to the Author):

Yoshida et al. assembled a cohort of 30 healthy children from neonate to adolescent to sample nasal, tracheal brushings and matched PBMCs. They also assembled a disease cohort with 4 pediatric COVID-19 patients, and 18 adult COVID-19 patients with a range of disease severities; from all patients, nasal brushings and PBMCs were sampled, and tracheal samples were obtained in one infant and 4 adults. Nasal, tracheal and bronchial tissue were processed for 10x droplet-based single cell RNA sequencing, and PBMCs were processed using a 192-antibody CITE-seq panel. From their 5' 10X single cell libraries they also amplified TCR and BCR sequences and performed clonotype analysis for the lymphocytes. In total, they obtained ~460,000 high quality single cell transcriptomes, and annotated 42 airway and 31 blood cell populations, including most of the major epithelial and immune cell types of the conducting airways and the blood. They performed trajectory and RNA velocity analysis to infer developmental trajectories in airway and immune cell types.

The authors describe age-related changes in the immune landscape of child development, in terms of cell type composition (but see major comment below), and show effector- or innate-immunity related gene expression in the naïve lymphocytes. They also describe an inflammatory epithelial transit cell present in COVID-19 samples demonstrating inflammatory phenotype that is, by RNA velocity, developmentally connected and thus suggested to differentiate into ciliated cells. These are interesting findings, but in contrast to the detailed characterization of age-related changes in normal development, the authors do not perform the same in-depth comparison with respect to COVID-19. For most of the COVID-19 immune analysis, the comparisons collapsed the patient cohorts into two categories: pediatric or adult, and is limited to comparing cell type abundances, without discussion of the gene expression programs driven in response to SARS-CoV-2 infection. This severely reduces the impact of this analysis and undermines the rationale to perform broad single cell RNA sequencing – the ability to study the gene expression program in transcriptionally defined cell populations, and single cell dynamics. If it is true that the immune landscape is dynamic and varied between age groups, a more appropriate analysis would be to compare the pediatric patients to age-matched healthy counterparts.

In summary, the dataset is a multi-national, multi-center clinical cohort with high quality single cell transcriptomes that represents a valuable resource to the community of researchers in pulmonary biology, systems immunology, and SARS-CoV-2 virology. The analysis presented was

performed competently but is disjointed, perhaps because they have attempted to cover child development as well as pediatric and adult COVID-19 infection, and at three different sites (nasal brushings, tracheal brushings, blood) in a single paper. The analysis lacks organization with respect to the logical progression of major findings, validation of any of the biological hypotheses generated from the bioinformatic analysis, and a focused comparison characterizing the diverse range of host response to SARS-CoV-2 infection that is specific to age, and thus fails to explain why pediatric COVID-19 presents so differently from adult disease. The single cell RNA-seq dataset is valuable, but inadequate analysis addressing the most important questions about pediatric COVID-19 disease, and lack of any follow-up experiments, weaken the few takeaway points and hence interest and impact for the Nature readership.

Major comments

1. The authors state they “track(ed) the developmental changes for 42 airway and 31 blood cell populations from infancy, through childhood to adolescence” but the discussion of gene expression changes across ages is extremely limited. The authors might consider publishing the COVID data separately and expand on their analysis across development, or remove the developmental/childhood data and instead focus the analysis on adult COVID infection.
2. The author describe an inflammatory epithelial transit cell (IETC). However, the concept that airway epithelial cells respond to foreign stimulus or microbes through upregulating interferon or inflammatory cytokines is long recognized (Geiss et al PNAS 2002), as is plasticity in the identities of the major airway epithelial cell types (ciliated, basal, goblet) (Rock et al PNAS 2009) and alternative differentiation trajectories (Plasschaert et al Nature 2018, Montoro et al Nature 2018). It would be important if IETCs could be shown to represent a novel, disease relevant trajectory.
3. On histology in Figure 2i, the double positive staining of S100A8/9 and EPCAM is hard to evaluate without higher power imaging. There doesn't seem to be much double positive staining in the low-power magnification. If the hypothesized function of S100A8/9 expression in these IETCs is neutrophil recruitment is true, the authors should show staining of infiltrating neutrophils in the vicinity of IETCs. Better staining of IETCs is also needed to exclude the possibility that these transcriptional profiles represent a technical artifact, such as ambient RNA from neutrophils or monocytes (which also highly express S100A8/9).
4. Inferring cellular abundance from single-cell RNAseq data, even when care is taken to use “general” dissociation procedures, can be misleading. No isolation procedure will perfectly represent the actual cellular abundances due to differences in ease of dissociation, viability, and maintenance of their expression program, all of which change with development. Furthermore, the samples in this study that came from nasal or bronchial brushing procedures can be extremely variable with respect to the particular physician performing the procedure, and different degrees of epithelial shedding in different patients and sites. The observed differences in cell type abundance in Figure 2e (and other similar figures) are hard to have confidence in because of large variation in the number of subjects per group and between cell type abundances among individuals (exemplified in Figure 3f and Extended Data Figure 9c).
5. In Figure 2f, the authors used a generalised linear mixed model to compute the fold change estimate and an LTSR statistic with respect to each clinical/technical factor, but this is not explained well – what is the fold change being computed being compared to? The authors need to provide evidence that this chosen statistical test reflects the underlying true difference in cell type proportion, due to the technical caveats mentioned above, using a complementary quantitative method to confirm population abundances (eg, immunohistochemistry, FACS).
6. Could the small differences in interferon responses presented in Figure 2g and ACE2 expressing cells in Fig 1h and Extended Data Figure 3c be explained simply by differences in the number of individuals, ages, and/or ethnicities represented in each group?
7. An interesting and potentially important conclusion is the presence of naive B and T lymphocytes in neonates and infants with a unique gene expression signature bearing hallmarks of innate immunity in Figure 3d, including RNA expression of GZMA in CD4/CD8 T cells, and NKG7 in CD8+ cells, suggesting effector immune function in naïve T lymphocytes only in neonates. However, how this relates to COVID-19 disease is not clear in the subsequent data analysis, as the immune analysis in both airway and blood of COVID-19 patients (Figures 3e-h, Figures 4a-b) is

restricted to cell type abundance comparisons, and virtually no analysis or discussion of the gene expression program underlying the phenotypic changes with disease.

8. The Abstract statement that the dataset enables study of "the spatial dynamics of infection" seems too strong.

9. The results section of the manuscript is ineffectively organized. Sections are organized by sample (healthy or COVID-19), technique (gene expression, CITE-seq, or immune repertoire), tissue location (airway, blood), but not by major conclusions, especially as it pertains to COVID-19 disease relevance. For example, the discovery of innate lymphoid and non-clonally expanded naïve T cells in peripheral blood in COVID-19 began in "Immune cell states in healthy paediatric blood", and ended in the last results section, "Lymphocyte clonality". This makes the paper difficult to penetrate and hinders evaluation of the results.

Minor

1. The authors alignment of the single cell transcriptomes included the SARS-CoV-2 reference genome, and wrote in the results that they detected viral reads, defined as > 10 viral reads, in only 4 of the individuals (3 nasal and 1 bronchial), then go on to show in Figure 1f the sum of viral expression across epithelial and immune cell types, with goblet and ciliated cells being the major targets. This should be elaborated – what is the correlation with disease severity and progression, and are these individuals pediatric or adult? Why was a sum of viral expression showed, and not a per-cell distribution, stratified by individual? It is hard to interpret the sum of viral expression (units were not given in the y-axis) as it is confounded by total number of detected cells in each type, aggregated across patients. Is there a difference in gene expression in these cells from which SARS-COV-2 reads were detected? Is there a difference in the individuals from which SARS-CoV-2 reads were detected in their airway? This would be important in all their downstream analysis of host gene expression changes in COVID-19, to distinguish direct viral effects of infection, versus indirect effect from inflammatory and antiviral signals.

2. This impressive cohort of patients included patients from the UK (London) and US (Chicago) – were they infected by different (B.1.1.7 vs D614G) or the same strains? Did the diagnostic PCR test resolve this? Given differences between strains in their virulence related to cell entry, and immunoreactivity, this may be useful scientifically and epidemiologically to resolve.

Referee #3 (Remarks to the Author):

Interesting and potentially impactful work on immune landscape in pediatric versus adult COVID-19. While the scope of the work and findings are interesting and potentially important there are substantive issues to consider in its current form. The major issue here is the relative paucity of COVID-19 samples, especially pediatric samples, and the lack of orthogonal validation of any of the key findings.

1. The most substantive concern relates to the small sample number, especially the 4 pediatric COVID-19 patients who were profiled. Without larger numbers of samples (whether fully profiled or used to validate key findings), one is left with the concern that this work is mainly hypothesis generating in its current form.

2. At times it is hard to say if the results (e.g. detection of SARS-CoV-2 reads) were from adult, pediatric, or from age-unselected samples. This makes the work at times hard to follow especially for specific aspects based on a subset of profiled samples.

3. It would be important to validate the presence of IETCs using a second modality, such as flow/RT-PCR including on additional pediatric healthy/covid-19 samples.

4. Can the differential interferon response in children with covid-19 be validated on a second set of samples, even by assessing key target gene expression? Small sample numbers again raise

concerns as to the generalizability of this interesting observation. Moreover, this seems to be something that could be studied in the peripheral blood (alongside cellular composition) in a large number of adult vs. pediatric covid-19 samples given the availability of covid-19 PB samples. This is a missed opportunity to validate and extend their work.

5. A lot of this paper is focused on looking at age-dependent changes in cellular composition and gene expression in normal cells. While this is an interesting question, it is not nearly as novel or impactful as the covid-19 findings yet they are the bulk of the samples and analyses presented. As such this does reduce the immediacy and novelty of the findings substantively.

6. As presented the T and B cell clonality studies are descriptive and of unclear relevance to COVID-19 pathogenesis, immune response or disease severity.

Author Rebuttal to Initial Comments:

Referee #1 (Remarks to the Author)	Authors' Response
A. Summary of the key results Yoshida et al describe a single-cell multiomics analysis of 4 children and 18 adults infected with SARS-CoV-2 that included both PBMCs, nasal and tracheal brushings. Similar sample triads were also collected from 30 healthy children of different age, allowing comparisons of immune cell states during childhood, and in response to SARS-CoV-2 infection. B. Originality and significance: The study involves state-of-the art methodologies for single cell analyses of both mRNA and proteins. Another strength is the matched samples from blood, nasal and tracheal mucosa. The main limitation of the study is the very small sample size, both from different age groups of healthy children and from pediatric COVID-19 cases. As a consequence of these very small numbers the authors fail to draw interesting novel conclusions about age-associated differences and between COVID-19 infected and healthy children and adults. Most of the reported findings involve correlates between age and individual genes and cells and sweeping statements such as "...plasma cells and gamma delta (g/d) T cells all showed a striking increase in young children (2 to 6 years of age), presumably reflecting greater antigen exposure of children in this age group, for example in pre-school nurseries". The children were healthy when sampled and plasmablast responses are typically transient in the blood but since this finding is based upon 7 blood samples collected from children between 2-6 years of age, this	We are grateful for the referee's comments and have addressed sample size by increasing sample numbers and changing the focus of our comparisons on changes between SARS-CoV-2 infected children and adults. We agree that the sample size in our paediatric COVID-19 cohort in our original submission limited our ability to draw strong conclusions. To address this, we have quintupled this cohort by including 15 additional paediatric patients with COVID-19. To strengthen the comparisons that are made in this study even further, we have also included a healthy adult cohort of 11 individuals and 15 additional paediatric or adult individuals that have recovered from COVID-19, giving us the following numbers in total:  - 30 paediatric healthy - 19 paediatric COVID-19 - 18 adult COVID-19 - 15 post-COVID-19 - 11 adult healthy individuals. (See Figure 1b, Extended Data Figure 1a, Extended Data Table 1) Overall, we have now expanded our patient numbers significantly and analysed data from 93 patients (Figure 1b) 177 samples and 659,000 high-quality sequenced single cells. This is to our knowledge the largest multi-site (nose, trachea, bronchi, blood) single cell paediatric COVID-19 study to date. Importantly, the expanded and more balanced dataset enabled us to confidently identify a number of mechanisms that explain why children are at a lower risk of developing severe COVID-19 as outlined below:  1. We made the novel observation that the airway of healthy children is in a pre-activated interferon response state (Figure 2f,g), which could enable them to respond with a stronger initial immune response to restrict viral spread. 2. The paediatric immune response to COVID-19 is characterised by a naive (increased naive B and T cells) and tolerised state (T reg) (Figure 3c), while adults manifest with a much stronger and highly cytotoxic immune response in the blood (Figure 3c,g). The cytotoxic character of the systemic response could lead to a higher risk of immune-related damage across organs in adults compared to children.

might be a completely random event without any biological relevance.	 3. We observe higher T cell immune repertoire diversity in children (Figure 3d) which could contribute to a more efficient establishment of an adaptive immune response against SARS-CoV-2. 4. We find novel interferon-stimulated subpopulations in blood (particularly B, NK, and T cells; Figure 3g,h, Extended Data Figure 11) that persist beyond the early phase of COVID-19 in adults (Figure 3i), but not in children. These persistent, inflamed circulating cells increase the cytotoxic character (increased IFN-NK and IFN-T CD8 CTL; Figure 3g) of the adult-specific immune response even further, possibly accounting for the more severe symptoms in adults. In addition, we have further refined the annotation of the single cell airway and blood landscape and now identify 93 distinct cell populations, including multiple novel cell types such as epithelial cells with an inflammatory gene expression (SA100A8/A9) which are found enriched in COVID-19 patients.
C. Data & methodology (validity of approach, quality of data, quality of presentation) I think in general the developmental changes in blood of healthy children are too severely underpowered to be informative. There is a great deal of variability among children and that is simply not possible to capture with 5-7 blood samples collected from different children in each of the developmental stages during childhood.	In our revised manuscript, we no longer focus on the developmental changes of healthy children in order to present a more focussed story as suggested by the reviewer in point F. Naturally these individuals continue to be part of our cohort, but we mainly use these as healthy controls for our COVID-19 patients. We only discuss the changes across age in the context of the differences between paediatric and adult COVID and do not claim extensive novel findings from comparisons between the different age groups.
D. Appropriate use of statistics and treatment of uncertainties There are no obvious technical errors or unwarranted uses of statistical tests but there is a general lack of statistical power due to a very small sample set overall. Especially among the healthy children there are about ~6000 cells sequenced per PBMC sample, which means that many rare immune cell types are missed, and others are represented by a handful cells or so, leading to general problems with uncertainties in all conclusions drawn. This is particularly damning given that the authors aim to perform so many individual comparisons,	We expanded our sample set, as outlined above, and hence statistical power has increased significantly. The current study has sequenced over 237,000 airway cells and 422,000 blood cells. Given that rare cell types can often be detected with as few as 50 cells, we have a very good ability to detect rare cell types, as exemplified by the detection of neuroendocrine cells in the nasal samples and the identification of a novel interferon-stimulated subset of 21 hematopoietic progenitor cells. As requested by the reviewer, we reduced the number of individual comparisons. Hence, in the revised manuscript, we focus our results on the comparison of children versus adult COVID-19 patients. However, we wish to point out that in cell type composition analysis we are using a statistical model that is able to examine the whole data set and model the different variables at once, which reduces the number of comparisons. In places where multiple comparisons are performed, an FDR procedure is used to control for false positives.

across tissue types, across ages and between healthy and SARS-CoV-2 infected.	
E. Conclusions (robustness, validity, reliability) See D above. I worry that many of the conclusions are simply overinterpretations of random fluctuations in the data. I think the authors should perform power-analyses to first determine what conclusions are possible to draw from this extremely high-dimensional analysis performed in a small number of cells from a handful of samples per group.	As discussed above, we have greatly expanded and balanced our dataset to draw more confident conclusions. And while our COVID-19 study is unique in its multi-organ profiling approach and paediatric specific angle, the size of our data set (659K single cells) is substantially larger than the other COVID-19 airway profiling studies in adults such as Chua et al. (160K single cells) and Ziegler et al. (33K single cells). Due to the complexity of single-cell datasets, there are not yet any widely accepted methods available to perform power calculations for studies such as ours. However, the statistical framework that is employed to perform cell type composition analyses in this study specifically fits random effects to model any unexplained variance in a rigorous manner. In addition, we have now restricted our analyses to answer the key question of what the paediatric-specific response to COVID-19 is, which has made the interpretation of the data much simpler and the resulting conclusions more robust.
F. Suggested improvements (experiments, data for possible revision) I suggest the authors focus their story on the most robust, and in my opinion most interesting aspect of the paper which are the analyses across matched tissue sample types and the similarities/differences between blood and airway mucosa in the COVID-19 patients vs healthy children and adults.	We thank the reviewer for this suggestion and have now focussed our manuscript comparing the differences between paediatric and adult COVID-19 patients and on relationships between cells the nasal mucosa and blood. We report the novel finding that paediatric epithelial cells have higher basal interferon stimulated gene (ISG) signatures and show greater induction of ISG responses in their nasal immune response, in particular in the innate immune compartment (Figure 2f). In contrast, in adults there is a strong induction of an epithelial interferon (IFN) response. In blood, there are more extensive differences between children and adults (Figure 3). In children the innate immune response dominates, whilst in adults it is a more cytotoxic one. In addition, IFN-stimulated immune populations are much more abundant in adults. Interestingly, in adults these populations correlate with a strong IFN response in the nose.
Also, the analyses between adult and pediatric COVID-19 patients are hampered by lack of details in the sampling time from symptom onset. For example the differences seen with respect to clonal expansion in the mucosa between children and adults with COVID-19 could be explained by differences in the time from symptom onset but such information is not provided and no longitudinal samplings are done for the mucosal samples. Thus, I find it hard to assess whether these differences are of biological relevance or not.	We agree that sampling time from symptom onset is very important to consider. The details of sampling and timing of sampling are given in Extended Data Figure 1, which now includes data for all the new patients. The overall distribution of time of sampling with respect to onset of symptoms is similar in the paediatric and adult cohorts. With regard to longitudinal sampling, we do include a number of longitudinal samples that are highlighted in the same figure by the use of different colours (red, first sample, blue: second sample). In addition, we now include an analysis that examines changes in the interferon response in blood with respect to the onset of symptoms and find a gradual reduction in these responses with time in adults but not in children.

G. References (appropriate credit to previous work?) Yes I find this appropriate.	In addition, the current revision includes additional relevant, but distinct papers (including preprints) that have been published recently.
H. Clarity and context (lucidity of abstract/summary, appropriateness of abstract, introduction and conclusions) The paper is well written, and figures are overall complex but well described. Conclusions are mostly speculative and not followed up on with additional experiments making it difficult to verify any of the conclusions made by the authors.	We are grateful for acknowledging that our paper was well written. In our revised manuscript we have increased our sample size significantly and carried out an analysis that is more focused on COVID-19 specific questions. Therefore the structure of the manuscript has changed and our conclusions are much better supported.
Referee #2 (Remarks to the Author):	Authors' Response
A. Summary of the key results. Yoshida et al. assembled a cohort of 30 healthy children from neonate to adolescent to sample nasal, tracheal brushings and matched PBMCs. They also assembled a disease cohort with 4 pediatric COVID-19 patients, and 18 adult COVID-19 patients with a range of disease severities; from all patients, nasal brushings and PBMCs were sampled, and tracheal samples were obtained in one infant and 4 adults. Nasal, tracheal and bronchial tissue were processed for 10x droplet-based single cell RNA sequencing, and PBMCs were processed using a 192-antibody CITE-seq panel. From their 5' 10X single cell libraries they also amplified TCR and BCR sequences and performed clonotype analysis for the lymphocytes. In total, they obtained ~460,000 high quality single cell transcriptomes, and annotated 42 airway and 31 blood cell populations, including most of the major epithelial and immune cell types of the conducting airways and the blood. They performed trajectory and RNA velocity analysis to infer developmental trajectories in airway and immune cell types.	We are delighted to provide a much improved manuscript with the following improvements:  1) We have expanded the data set by including 15 additional paediatric COVID-19 patients (ranging from asymptomatic/mild to severe cases), 11 healthy adult controls and 15 post-COVID-19 patients, giving us the following numbers in total: and now have  ● 30 paediatric healthy ● 19 paediatric COVID-19 ● 18 adult COVID-19 ● 15 post-COVID-19 ● 11 adult healthy individuals in our study. Overall, we have now expanded our patient numbers significantly and analysed data from 93 patients (Fig. 1b) 177 samples and 659,000 high-quality sequenced single cells. This is a very large study and to our knowledge the largest multi-omic single cell paediatric COVID-19 study to date. The increased paediatric sample numbers as well as the inclusion of healthy adults allowed us to generate more robust results and conclusions. 2) We focused our revised manuscript on changes between SARS-CoV-2 infected adults and children to ensure that sample sizes are large enough to support our conclusions. 3) We have further improved the already highly granular cell type annotation, allowing us to define 59 different cell types in the airways and 34 different cell types and states in blood. 4) Additional changes are listed in our point-by-point responses to the referees.

B. Originality and significance (if not novel, please include reference) The authors describe age-related changes in the immune landscape of child development, in terms of cell type composition (but see major comment below), and show effector- or innate-immunity related gene expression in the naïve lymphocytes. They also describe an inflammatory epithelial transit cell present in COVID-19 samples demonstrating inflammatory phenotype that is, by RNA velocity, developmentally connected and thus suggested to differentiate into ciliated cells. These are interesting findings, but in contrast to the detailed characterization of age-related changes in normal development, the authors do not perform the same in-depth comparison with respect to COVID-19. For most of the COVID-19 immune analysis, the comparisons collapsed the patient cohorts into two categories: pediatric or adult, and is limited to comparing cell type abundances, without discussion of the gene expression programs driven in response to SARS-CoV-2 infection. This severely reduces the impact of this analysis and undermines the rationale to perform broad single cell RNA sequencing – the ability to study the gene expression program in transcriptionally defined cell populations, and single cell dynamics. If it is true that the immune landscape is dynamic and varied between age groups, a more appropriate analysis would be to compare the pediatric patients to age-matched healthy counterparts.	As outlined above we have substantially changed the structure of our manuscript to perform in-depth comparisons between paediatric and adult COVID-19 patients, in line with the referee’s recommendations. This analysis was facilitated by the increased numbers of paediatric COVID-19 patients as well as the inclusion of healthy adults. We carry out a detailed analysis of (i) changes in cell type proportions in health versus disease, using a Poisson linear mixed model that allows us to take into account changes that can be attributed to age alone and (ii) now also include a detailed analysis of gene expression changes associated with disease. We conclude that the major perturbed pathways are responses to interferon signaling, TNFalpha signaling, and neutrophil chemotaxis signatures. We also provide detailed DE analysis in those cell types that underwent the most prominent changes in response to COVID-19. We further make use of our single cell data to examine each of these responses by cell type and find multiple differences between the adult and paediatric responses, as well as differences in the local versus the systemic response. In short, in children the local immune responses are stronger, with notable induction of interferon responsive genes in innate immune cell types such as monocytes and in helper T cells, but lower cytotoxic responses. On the other hand, in the systemic immune response, we see the induction of specific IFN-induced subpopulations (T, NK B mono etc) that are much stronger in the adult rather than paediatric patients. These findings may account for fewer symptoms in paediatric versus adult patients. The reviewer also wonders about cell dynamics: in diseased adults we report increased levels of transit epithelial cells which lie on a developmental trajectory towards ciliated cells, a cell type shown to die in response to COVID infection (Zhu et al (2020)Nat. Commun. 11, 3910). The changes of the immune system with time are challenging to analyse. It is exactly for this reason that we have used a Poisson linear mixed model that is able to deconvolute the changes that can be attributed to age alone from the changes that are due to COVID-19. (For a more detailed description see Referee 1, point D)
In summary, the dataset is a multi-national, multi-center clinical cohort with high quality single cell transcriptomes that represents a valuable resource to the community of researchers in pulmonary biology, systems immunology, and SARS-CoV-2 virology. The	We appreciate that the reviewer already viewed our first data set as a valuable resource for the community. As outlined above, the structure and analysis of our manuscript has been substantially improved and our conclusions are more strongly supported than in the original submission.

analysis presented was performed competently but is disjointed, perhaps because they have attempted to cover child development as well as pediatric and adult COVID-19 infection, and at three different sites (nasal brushings, tracheal brushings, blood) in a single paper. The analysis lacks organization with respect to the logical progression of major findings, validation of any of the biological hypotheses generated from the bioinformatic analysis, and a focused comparison characterizing the diverse range of host response to SARS-CoV-2 infection that is specific to age, and thus fails to explain why pediatric COVID-19 presents so differently from adult disease. The single cell RNA-seq dataset is valuable, but inadequate analysis addressing the most important questions about pediatric COVID-19 disease, and lack of any follow-up experiments, weaken the few takeaway points and hence interest and impact for the Nature readership.	Our conclusions with respect to gene expression changes are already enumerated above. In addition, we for the first time have the ability to link the local immune response to the systemic immune response through the analysis of paired local airway and blood samples. We found a strong correlation between the induction of a strong epithelial immune response and the inductions of specific IFN-stimulated blood cell populations in adults. In contrast, in children only a mild epithelial IFN response was observed and only few cells in the periphery became IFN-stimulated. These findings are entirely consistent with the milder symptoms seen in children versus adults. In its entirety, our results suggest that there is a stronger local IFN response in children that is likely to be able to constrain the viral infection. In contrast, in adults the infection leads to a systemic activation of a range of IFN-induced cell populations that may contribute to the immune-related organ and tissue damage seen in more severe COVID-19. These findings have implications for the nuanced targeted manipulation of the interferon response over the course of the infection. However, clinical studies validating this, are beyond the scope of this paper.
Major comments	
1. The authors state they “track(ed) the developmental changes for 42 airway and 31 blood cell populations from infancy, through childhood to adolescence” but the discussion of gene expression changes across ages is extremely limited. The authors might consider publishing the COVID data separately and expand on their analysis across development, or remove the developmental/childhood data and instead focus the analysis on adult COVID infection.	We would like to thank the reviewer for these suggestions and we have indeed now focussed our paper on the differences between paediatric and adult COVID patients as detailed above. We re-iterate the response to referee 1 detailing our new findings: Overall, we have now expanded our patient numbers significantly and analysed data from 93 patients (Figure 1b) 177 samples and 659,000 high-quality sequenced single cells. This is to our knowledge the largest multi-site (nose, trachea, bronchi, blood) single cell paediatric COVID-19 study to date. Importantly, the expanded and more balanced dataset enabled us to confidently identify a number of mechanisms that explain why children are at a lower risk of developing severe COVID-19 as outlined below:  1. We made the novel observation that the airway of healthy children is in a pre-activated interferon response state (Figure 2f,g), which could enable them to respond with a stronger initial immune response to restrict viral spread.

	 2. The paediatric immune response to COVID-19 is characterised by a naive (increased naive B and T cells) and tolerised state (T reg) (Figure 3c), while adults manifest with a much stronger and highly cytotoxic immune response in the blood (Figure 3c,g). The cytotoxic character of the systemic response could lead to a higher risk of immune-related damage across organs in adults compared to children. 3. We observe higher T cell immune repertoire diversity in children (Figure 3d) which could contribute to a more efficient establishment of an adaptive immune response against SARS-CoV-2. 4. We find novel interferon-stimulated subpopulations in blood (particularly B, NK, and T cells; Figure 3g,h, Extended Data Figure 11) that persist beyond the early phase of COVID-19 in adults (Figure 3i), but not in children. These persistent, inflamed circulating cells increase the cytotoxic character (increased IFN-NK and IFN-T CD8 CTL; Figure 3g) of the adult-specific immune response even further, possibly accounting for the more severe symptoms in adults. In addition, we have further refined the annotation of the single cell airway and blood landscape and now identify 93 distinct cell populations, including multiple novel cell types such as epithelial cells with an inflammatory gene expression (SA100A8/A9) which are found enriched in COVID-19 patients.
2. The author describe an inflammatory epithelial transit cell (IETC). However, the concept that airway epithelial cells respond to foreign stimulus or microbes through upregulating interferon or inflammatory cytokines is long recognized (Geiss et al PNAS 2002), as is plasticity in the identities of the major airway epithelial cell types (ciliated, basal, goblet) (Rock et al PNAS 2009) and alternative differentiation trajectories (Plasschaert et al Nature 2018, Montoro et al Nature 2018). It would be important if IETCs could be shown to represent a novel, disease relevant trajectory.	In our latest data set the IETCs fall into two subtypes, which we have remained Transit epi 1 and 2. Of these, transit epi 2 in particular appears to be associated with both COVID-19 and with age. We also compare our transit epithelial populations linking secretory and ciliated cells to those that have been described in a previous analysis of developmental trajectories in the nose, that identified deuterosomal cells (Deprez et al (2020)Am. J. Respir. Crit. Care Med.) as well as additional trajectories (Chua et al (2020) Nat. Biotechnol.). Deuterosomal cells are clearly present. However, the markers for the additional “bridging cell types” do not map to our transit epithelial populations particularly well. In contrast, our marker FP671120.4 is more specific to these cells, suggesting that these are indeed novel populations (Extended Data Figure 7a). In addition, we carry out logistic regression label transfer (in Extended Data Figure 7) that shows that the probability of correct label transfer of previously defined populations is relatively low for the transit epi 1 and 2 cells (Extended Data Figure 7b,c). We now define this cell type better and find that it overlaps with previously described Secretory-Diff, IRC and other secretory cells.
3. On histology in Figure 2i, the double positive staining of S100A8/9 and EPCAM is hard to evaluate without	We have provided a better image in Fig 2k showing the S100A9+EpCam+ cells clearer, amidst S100A9-EpCam+ and S100A9+EpCam- cells, which are likely to be infiltrating immune cells, such as the

higher power imaging. There doesn't seem to be much double positive staining in the low-power magnification. If the hypothesized function of S100A8/9 expression in these IETCs is neutrophil recruitment is true, the authors should show staining of infiltrating neutrophils in the vicinity of IETCs. Better staining of IETCs is also needed to exclude the possibility that these transcriptional profiles represent a technical artifact, such as ambient RNA from neutrophils or monocytes (which also highly express S100A8/9).	monocytes the referee refers to These cells are present in a distinct spatial location (ie. not along the surface epithelium). Examining the H&E section our histopathologist did not detect extensive evidence of neutrophil infiltration. We note that our data set is generally low in neutrophils, but much higher in other data sets that have analysed BAL (Pandolfi et al (2020)BMC Pub Med 20:301; PMID: 33198751) or aspirate suggesting that neutrophils may migrate into the lumen.
4. Inferring cellular abundance from single-cell RNAseq data, even when care is taken to use “general” dissociation procedures, can be misleading. No isolation procedure will perfectly represent the actual cellular abundances due to differences in ease of dissociation, viability, and maintenance of their expression program, all of which change with development. Furthermore, the samples in this study that came from nasal or bronchial brushing procedures can be extremely variable with respect to the particular physician performing the procedure, and different degrees of epithelial shedding in different patients and sites. The observed differences in cell type abundance in Figure 2e (and other similar figures) are hard to have confidence in because of large variation in the number of subjects per group and between cell type abundances among individuals (exemplified in Figure 3f and Extended Data Figure 9c).	In order to carry out single cell sequencing study, cellular dissociation is essential. Whilst this clearly induces some changes in gene expression (Van den Brink et al. (2017) Nature Methods)), as long as the same dissociation procedure is carried out for all samples, changes in cell type composition with age will be able to be identified. For dissociation we used a modified protocol previously published in the literature looking at characterizing the human airway epithelium at the single cell level (Deprez.M, et. al 2019). The majority of the healthy pead and COVID-19 positive samples (those collected at GOSH) were performed by the same ENT clinicians. The samples were generally collected by the same two clinicians based at either the Royal Free of University College London. Variation in cell type composition between individuals is an issue that all scRNAseq studies have to contend with. As we have now increased our sample size considerably, we are able to obtain statistically significant results.
5. In Figure 2f, the authors used a generalised linear mixed model to compute the fold change estimate and an LTSR statistic with respect to each clinical/technical factor, but this is not explained well – what is the fold change being computed being compared to? The authors need to provide evidence that this chosen statistical test reflects the underlying true difference in cell type proportion, due to the technical caveats mentioned	The log fold change is relative to the grand mean and adjusted so that it becomes 0 when there is no effect. LTSR is the probability that the estimated direction of the effect is true, i.e. the probability that the true log fold change is greater than 0 if the estimated mean is positive (or less than 0 if the estimated mean is negative). It is calculated based on the estimated mean and standard deviation of the distribution of the effect (log fold change), which is to an extent similar to performing a (one-sided) one-sample Z-test and showing (1 - p_value).

above, using a complementary quantitative method to confirm population abundances (eg, immunohistochemistry, FACS).	This text has been included in the Methods section. In addition, we highlight the benefits of such a statistical analysis in the relevant results sections.
6. Could the small differences in interferon responses presented in Figure 2g and ACE2 expressing cells in Fig 1h and Extended Data Figure 3c be explained simply by differences in the number of individuals, ages, and/or ethnicities represented in each group?	Our current data is very different from the originally presented data. We show strong induction of interferon responses, for example for all epithelial and all immune cells in Figure 2f with strong statistical significance. In blood, we chose to use a different analysis, as we found that iFN-stimulated populations clustered separately on UMAPs. Here we saw an increase in these IFN-stimulated populations in COVID-19, with p values as low as 0.001. This numerical analysis shows that these are strong and significant results.
7. An interesting and potentially important conclusion is the presence of naïve B and T lymphocytes in neonates and infants with a unique gene expression signature bearing hallmarks of innate immunity in Figure 3d, including RNA expression of GZMA in CD4/CD8 T cells, and NKG7 in CD8+ cells, suggesting effector immune function in naïve T lymphocytes only in neonates. However, how this relates to COVID-19 disease is not clear in the subsequent data analysis, as the immune analysis in both airway and blood of COVID-19 patients (Figures 3e-h, Figures 4a-b) is restricted to cell type abundance comparisons, and virtually no analysis or discussion of the gene expression program underlying the phenotypic changes with disease.	As suggested by this referee, we have removed some of the results relating to the analysis of healthy children only.
8. The Abstract statement that the dataset enables study of "the spatial dynamics of infection" seems too strong.	We have re-worded our abstract.
9. The results section of the manuscript is ineffectively organized. Sections are organized by sample (healthy or COVID-19), technique (gene expression, CITE-seq, or immune repertoire), tissue location (airway, blood), but not by major conclusions, especially as it pertains to COVID-19 disease relevance. For example, the discovery of innate lymphoid and non-clonally expanded naïve T cells in peripheral blood in COVID-19 began in "Immune cell states in healthy paediatric blood", and	We agree with the reviewer and have now completely restructured the results section, with subtitles highlighting our findings.  - Study cohort and experimental overview - Detection of SARS-CoV-2 reads in airway epithelial cells - Novel cellular subtypes in nasal, tracheal and bronchial epithelia - Changes of airway cell type proportions in paediatric and adult COVID-19 patients - Distinct COVID-19 gene expression changes in airways of children and adults - Multi-omic data reveals immune cell landscape of paediatric and adult blood - Reduced cytotoxic immune response to COVID-19 in paediatric blood

ended in the last results section, “Lymphocyte clonality”. This makes the paper difficult to penetrate and hinders evaluation of the results.	 - Expansion of naive T cells in paediatric COVID-19 patients - Immune repertoire in children is more diverse compared to adults - Interferon stimulation results in defined immune cell subtypes in blood - Circulating interferon-stimulated cells strongly associate with early COVID-19 in adults, but not in children
Minor comments	
1. The authors alignment of the single cell transcriptomes included the SARS-CoV-2 reference genome, and wrote in the results that they detected viral reads, defined as > 10 viral reads, in only 4 of the individuals (3 nasal and 1 bronchial), then go on to show in Figure 1f the sum of viral expression across epithelial and immune cell types, with goblet and ciliated cells being the major targets. This should be elaborated – what is the correlation with disease severity and progression, and are these individuals pediatric or adult? Why was a sum of viral expression showed, and not a per-cell distribution, stratified by individual? It is hard to interpret the sum of viral expression (units were not given in the y-axis) as it is confounded by total number of detected cells in each type, aggregated across patients. Is there a difference in gene expression in these cells from which SARS-COV-2 reads were detected? Is there a difference in the individuals from which SARS-CoV-2 reads were detected in their airway? This would be important in all their downstream analysis of host gene expression changes in COVID-19, to distinguish direct viral effects of infection, versus indirect effect from inflammatory and antiviral signals.	We have made changes to the way in which we analysed viral reads. Firstly, the viral reads are now shown as a fraction of airway cells with detected SARS-CoV2 mRNA in each cell type. In addition, we also examined the number of SARS-CoV-2 reads per individual prior to SoupX correction, as this might include actual viral particles and viral reads released from dying cells. We show the fraction of viral reads as a percentage of total reads in Figure 1h. Analysis of the interval between symptom onset and sample collection indicated that high viral reads are only detected in the early phase of the infection, for both paediatric and adult patients. This also correlated with a strong induction of the interferon response with 3 out of the top 4 individuals being identical between the two responses (Figure 1h; Extended Data Figure 13). To perform an analysis per cell type, ambient RNA-corrected reads (by SoupX) have to be used, and we found that only three individuals had high reads, making it difficult to reach firm conclusions. To account for possible confounders of the inflammatory response mediated by other pathogens that might be present in some patients, we also carried out a metagenomic analysis to detect reads from other pathogens in the transcriptomes of our patient cohort (See Methods; metagenomic analysis). Apart from SARS-CoV-2 and non-specific signals found in most samples, we did not detect any pathogens that were abundant in samples of interest.
2. This impressive cohort of patients included patients from the UK (London) and US (Chicago) – were they infected by different (B.1.1.7 vs D614G) or the same strains? Did the diagnostic PCR test resolve this? Given differences between strains in their virulence related to	Thank you very much for acknowledging that the cohort of patients we managed to include in our work is impressive. The question about strains is an important one. We are unable to look at viral strain as we only have 5’ tag sequencing and do not have access to the clinical diagnostic PCR tests. However, we have mapped a timeline showing the frequency of SARS-CoV-2 variants recorded in the UK and the USA at the time of SARS-CoV-2 PCR testing of each patient included in our COVID-19 cohort (paediatric samples are shown in red and adult ones in purple):

cell entry, and immunoreactivity, this may be useful scientifically and epidemiologically to resolve.

The SARS-CoV-2 variant frequency panel, was generated by GISAID and taken from their website using the following parameters; UK Dataset: ncov, gisaid, Europe.

Filtered on United kingdom. USA Dataset: ncov, gisaid, North America. Coloured by Clade (Source: GISAID - NextStrain. (n.d.). Retrieved July 12, 2021, from <https://www.gisaid.org/phylogenetics/global/nextstrain>).

Although a slight difference in the SARS-CoV-2 variant frequencies can be seen between the USA and UK around the time of sampling, the majority of the samples were collected when 20B and 20A, plus 20E (EU1) in the UK, were the dominant SARS-CoV-2 variants sequenced.

	In addition, we wish to point out that as yet no strain is known to be more pathogenic or result in a particular clinical syndrome, making this information less relevant in the context of our study.
Referee #3 (Remarks to the Author):	Authors' Response
A. Summary of the key results: Interesting and potentially impactful work on immune landscape in pediatric versus adult COVID-19. While the scope of the work and findings are interesting and potentially important there are substantive issues to consider in its current form. The major issue here is the relative paucity of COVID-19 samples, especially pediatric samples, and the lack of orthogonal validation of any of the key findings. 1. The most substantive concern relates to the small sample number, especially the 4 pediatric COVID-19 patients who were profiled. Without larger numbers of samples (whether fully profiled or used to validate key findings), one is left with the concern that this work is mainly hypothesis generating in its current form.	We acknowledge that the small paediatric COVID-19 number was a problem and have addressed this as outlined above (see referee 1+2), but reiterated below. Overall, we have now expanded our patient numbers significantly and analysed data from 93 patients (Figure 1b) 177 samples and 659,000 high-quality sequenced single cells. This is to our knowledge the largest multi-site (nose, trachea, bronchi, blood) single cell paediatric COVID-19 study to date. Importantly, the expanded and more balanced dataset enabled us to confidently identify a number of mechanisms that explain why children are at a lower risk of developing severe COVID-19 as outlined below:  1. We made the novel observation that the airway of healthy children is in a pre-activated interferon response state (Figure 2f,g), which could enable them to respond with a stronger initial immune response to restrict viral spread. 2. The paediatric immune response to COVID-19 is characterised by a naive (increased naive B and T cells) and tolerised state (T reg) (Figure 3c), while adults manifest with a much stronger and highly cytotoxic immune response in the blood (Figure 3c,g). The cytotoxic character of the systemic response could lead to a higher risk of immune-related damage across organs in adults compared to children. 3. We observe higher T cell immune repertoire diversity in children (Figure 3d) which could contribute to a more efficient establishment of an adaptive immune response against SARS-CoV-2. 4. We find novel interferon-stimulated subpopulations in blood (particularly B, NK, and T cells; Figure 3g,h, Extended Data Figure 11) that persist beyond the early phase of COVID-19 in adults (Figure 3i), but not in children. These persistent, inflamed circulating cells increase the cytotoxic character (increased IFN-NK and IFN-T CD8 CTL; Figure 3g) of the adult-specific immune response even further, possibly accounting for the more severe symptoms in adults.

	In addition, we have further refined the annotation of the single cell airway and blood landscape and now identify 93 distinct cell populations, including multiple novel cell types such as epithelial cells with an inflammatory gene expression (SA100A8/A9) which are found enriched in COVID-19 patients.
2. At times it is hard to say if the results (e.g. detection of SARS-CoV-2 reads) were from adult, pediatric, or from age-unselected samples. This makes the work at times hard to follow especially for specific aspects based on a subset of profiled samples.	We apologise for this and have now improved the results section to make it easier to follow, clearly indicating the source of the samples.
3. It would be important to validate the presence of IETCs using a second modality, such as flow/RT-PCR including on additional pediatric healthy/covid-19 samples.	These two epithelial populations are very similar and from our experience it is very unlikely that flow cytometry would be able to distinguish this population. However, in unsupervised clustering they fall into two clearly distinct domains, and can, for example, be distinguished by the expression of a nuclear marker gene (Extended Data Figure 3), which is why we have annotated these as distinct populations.
4. Can the differential interferon response in children with covid-19 be validated on a second set of samples, even by assessing key target gene expression? Small sample numbers again raise concerns as to the generalizability of this interesting observation.	We now have much larger sample numbers and present a more detailed analysis of the interferon responses.
Moreover, this seems to be something that could be studied in the peripheral blood (alongside cellular composition) in a large number of adult vs. pediatric covid-19 samples given the availability of covid-19 PB samples. This is a missed opportunity to validate and extend their work.	In our revised manuscript we have indeed included an analysis of interferon responses in peripheral blood. In fact, the responses were so clear that IFN-activated cells formed distinct clusters on UMAPs (see Figure 3). Analysis of the size of these subcluster allowed us to draw interesting conclusions:  1. Not all immune cells are activated to the same extent, suggesting that cells need to be stimulated in a specific environment where a critical signalling threshold is reached. 2. The proportion of IFN-stimulated blood population is much larger in adults than in paediatric COVID-19 patients, providing a possible explanation for the much weaker symptoms and milder disease course in children. 3. We observe a correlation between the induction of a nasal epithelial IFN response and the presence of distinct IFN-stimulated populations in the blood, in line with the disease starting off as a respiratory infection.
5. A lot of this paper is focused on looking at age-dependent changes in cellular composition and gene expression in normal cells. While this is an interesting question, it is not nearly as novel or impactful as the covid-19 findings yet they are the bulk of the samples	We agree with the reviewer and have therefore restructured our manuscript that now focuses specifically on the changes associated with COVID-19.

and analyses presented. As such this does reduce the immediacy and novelty of the findings substantively	
6. As presented the T and B cell clonality studies are descriptive and of unclear relevance to COVID-19 pathogenesis, immune response or disease severity.	We have reduced the complexity of the analysis of clonality with the clear conclusion that children show fewer expanded clones and are therefore expected to have a more diverse adaptive immune repertoire.

Reviewer Reports on the First Revision:

Referee #1 (Remarks to the Author):

In the revised manuscript the authors have added additional samples and sequenced more cells as compared to the initial version of the paper. This is important and makes the conclusions much more robust. Even more important is the change in focus away from healthy immune development using data from only very few children, towards a more focused paper comparing anti-SARS-coV-2 immune responses in children and adults, both in blood and in the airway mucosa. The revised paper is much improved and the information generated largely confirms prior work comparing immune responses in children and adults.

Referee #2 (Remarks to the Author):

A. Summary of key results

In this extensive revision of the original manuscript, Yoshida et al. expanded their original dataset to include nasal brushings and matched peripheral blood (PBMC) samples from 15 additional pediatric COVID-19 patients and 15 adult patients that had recovered from COVID-19, bringing the total study size to 49 pediatric (30 healthy, 19 with COVID-19), and 44 adult (11 healthy, 18 with COVID-19, and 15 post-COVID-19) patients. They analyzed the samples using a battery of single cell genomics technologies, mainly 10x droplet-based single cell RNA sequencing but also a 192-antibody CITE-seq panel and TCR and BCR sequencing with clonotype analysis for lymphocytes. The substantial increase in pediatric COVID-19 samples allowed the authors to switch the focus of the manuscript to a comparison of the effects of COVID-19 infection on pediatric vs adult patients, a clinically (and societally) important difference that is widely recognized but poorly understood mechanistically. The analysis was also extended to include not only a comparison of the effects of COVID-19 on the cell composition of nasal brushing and peripheral in pediatric vs adult patients, but also in depth analysis of the most prominent disease-associated changes in gene expression in the cell types. They find that the major disease-associated changes are responses to interferon signaling, TNF-alpha signaling, and neutrophil chemotaxis, but the most interesting results are the differences they report between the pediatric and adult responses including differences in local (nasal brushing) vs systemic (blood) responses. Children have higher basal (pre-COVID-19) interferon pathway signatures (pathway is "pre-activated") across many types of nasal epithelial cells, and children also show greater induction of the local (nasal) immune response, especially among innate immune cell types. The authors suggest that these may provide greater protection against viral infection and spread. In contrast, adults show a stronger systemic (blood) immune response, notably in cytotoxic T cells and with more abundant interferon-activated immune cell subpopulations (e.g., MK, B, and T cells) that correlate with the interferon response in the nose. This suggests greater systemic infection and inflammatory response in adults, which could cause or contribute to immune-related damage across many organs. Their analysis of COVID-19 patients stratified by time since symptom onset suggests an interesting hypothesis that dendritic cells initiate interferon signaling in the early stages of infection.

B. Originality and significance

This is a very large study and to our knowledge the largest multi-omic single cell pediatric COVID-19 study to date. The comparison between pediatric and adult responses to COVID-19 and of matched nasal and blood samples from most patients are key strengths of the revised manuscript. The paper now provides extensive data on the pressing question of differences in pediatric and adult responses and outcomes to the infection. Their data identify interesting and important cellular and molecular differences in the local and systemic responses between children and adults, which suggest hypotheses to explain the well appreciated clinical differences. It is a rich new resource that likely will be mined by the many investigators that wish to understand differences in the pediatric vs adult forms of COVID-19.

C. Data and methodology

Approach and quality of the data and analyses appear excellent. The quality of the presentation is greatly improved and is now also very good.

D. Appropriate use of statistics and treatment of uncertainties

Appropriate

E. Conclusions (robustness, validity, reliability)

The conclusions appear statistically robust. The intriguing hypotheses suggested will of course require experimental test but are appropriate for this stage of the analysis (and the clinical and social urgency of public access to their dataset and initial analyses).

F. Suggested improvements (experiments, data for possible revision)

There are still several pertinent new issues along with ones raised in the original review that remain unresolved and should be addressed.

1. The authors state find the airway of healthy children is in a "pre-activated" interferon response state compared to adults (Figure 2f,g). Is it possible that a subset of pediatric patients with upper respiratory infections common in their age group are confounding the aggregated interferon score? If so, this could be unrelated to COVID-19 or might provide cross protection against COVID-19.

2. The authors report higher TCR diversity in children (Figure 3d). The age-related decline in immune repertoire diversity is well appreciated (Naylor et al J Immunol 2005, Britanova et al J Immunol 2014), but importantly here for individual patients in the study was immune diversity correlated to disease severity/outcome?

3. Inferring cellular abundance from single-cell RNAseq data can be misleading, even when care is taken to use similar tissue sampling and dissociation procedures. This original concern was not sufficiently addressed in the revision. The authors state that the "majority" of samples were collected by the same ENT clinicians. How big of a majority? Are the identities of the collecting clinicians documented in the metadata and examined for variability? The biases that are intrinsic to the method of scRNA-seq cannot be resolved by increasing sample size for scRNA-seq. In Figure 2e, for the molecular types that are claimed to be statistically overrepresented, can validation be added by an orthogonal technology (e.g., cell counts of stained tissue sections)?

4. In Figure 2f, the genes that constitute IFN α , IFN γ , and TNF α signatures should be detailed. It is important to document the nature of the characterized IFN responses especially because an imbalanced/inappropriate interferon response has been proposed (Blanco-melo Cell 2020) to drive development of COVID.

5. The revised manuscript now notes the discovery of two novel inflammatory epithelial transit cell types (Transit Epi 1 and Transit Epi 2), instead of just one (IETCs) in the original submission. Proximal airway epithelial cells are known to be plastic at least in mice (Rock et al PNAS 2009) and exhibit complex developmental trajectories (Plasschaert et al Nature 2018, Montoro et al Nature 2018). Could these newly identified cell populations be developmental intermediates rather than stable cell types? Describing them as novel cell types should probably be de-emphasized, and the alternative possibility of a developmental trajectory considered or added.

6. Related to the prior point, on histology in Figure 2k, the double positive staining of S100A9/EPCAM is technically improved but sections should be co-stained with known proximal airway markers since mucous cells and basal cells can express S100A9, to further support that the populations are truly novel.

G. References (appropriate credit to previous work?)

Yes

H. Clarity and context (lucidity of abstract/summary, appropriateness of abstract, introduction and conclusions)

Yes, greatly improved and now very accessible

Referee #3 (Remarks to the Author):

This paper is significantly improved, especially with the increased sample size and increased focus on COVID-19 vs. normal immune cell types. Some important questions remain.

1. What is the basis for the increased activation/pre-activation of IFN signaling in COVID19 esp pediatric disease? is this due to increased local ligand production, increased response to ligand, or relief of transcriptional feedback resulting in increased gene expression output in response to the same stimulus?
2. A lot of the paper is interesting but descriptive WRT which cell types are altered in adult and pediatric COVID-19. What would be important is whether the authors could delineate key immune cell types/biomarkers which could be used to diagnose, follow and risk stratify patients using a clinically accessible test. Which things should clinicians at the bedside measure in their patients based on this elegant science?
3. Did any of the patients in this cohort have COVID-19 variants and if so did any of the immune system changes differ in delta or other variant contexts?
4. The peripheral blood analysis in particular is very descriptive and not novel compared to other work in this space. The authors should present how their multi-mic analysis leads to new insights, and not just confirms what others have seen with less expensive/intensive approaches.
5. Can the authors show that local IFN production by key cell types is significantly increased in specific covid-19 contexts (statistically, not just in one outlier case).
6. The introduction is quite long and a lot of it is not essential.
7. What do the authors make of the low recovery rate of viral RNA reads in covid-19 positive cases (36%). that seems surprisingly low

Referee 2	Authors' Response
1. The authors state find the airway of healthy children is in a "pre-activated" interferon response state compared to adults (Figure 2f,g). Is it possible that a subset of pediatric patients with upper respiratory infections common in their age group are confounding the aggregated interferon score? If so, this could be unrelated to COVID-19 or might provide cross protection against COVID-19.	The referee is raising an important question which we have actually extensively considered. We have concluded that the pre-activated IFN response state is very unlikely to be due to upper respiratory tract infections because of the following three reasons:  1. All the healthy children who gave samples were asymptomatic, i.e. they did not have any upper respiratory tract symptoms and/or tested negative for COVID-19 and other respiratory viruses (there were no detected co-infections) (Extended Data Table 1). 2. We looked for evidence of asymptomatic infections using Kraken in our dataset and did not find any in our healthy paediatric cohort, with the exception of low levels of Klebsiella pneumoniae in a few samples (a normal component of the microflora found in the upper respiratory and GI tract), although we were able to identify SARS-CoV-2 in our COVID-19 cohorts (Extended Data Figure 9). 3. When we looked at healthy immune cells (see Figure below, Figure R2.1), the pre-activated IFN state can be seen throughout childhood, not only in school age children.   Figure R2.1 (referee 2, point1)

	Overall, this suggests an innately pre-activated IFN state throughout childhood, rather than secondary to infections, although the latter might contribute and provide cross-protection as pointed out by the referee.
2. The authors report higher TCR diversity in children (Figure 3d). The age-related decline in immune repertoire diversity is well appreciated (Naylor et al J Immunol 2005, Britanova et al J Immunol 2014), but importantly here for individual patients in the study was immune diversity correlated to disease severity/outcome?	We thank the reviewer for pointing this out and have included citations for the mentioned work in the revised manuscript. The age-related decline in immune repertoire diversity has indeed been described below as a contributing factor to an ineffective establishment of an adaptive immune response against (viral) pathogens other than SARS-CoV-2 (for example in influenza: Yager et al J Exp Med. 2008). It is important to note that the main aim of our revised manuscript is to provide a comprehensive overview of molecular insights that could explain the changing disease phenotype over age. And while its relevance for disease in general has been reported before, to our knowledge there have been no other studies that describe reduced immune repertoire diversity as a relevant feature in COVID-19 during ageing. We carried out the analysis suggested by the referee where we compared TCR pool diversity between age and symptom severity (Figure R2.2). As hypothesized by the referee and in-line with our report, we observe a trend where reduced TCR diversity appears to be associated with disease severity in adults but not in children (where TCR diversity is not limiting). However, we decided to not include this expanded analysis in our revised manuscript as the difference we observe between severity is not significant, and because we cannot exclude the possible confounding effect of expanding anti-SARS-CoV-2 clones which could be more prominent in patients with severe symptoms.  Figure R2.2: Boxplot showing TCR diversity over disease severity in adults (left plot) and children (right plot). TCR diversity was quantified as a fraction of unique alpha and beta CDR3 sequences detected in all TCR expressing cells within each individual.

3. Inferring cellular abundance from single-cell RNAseq data can be misleading, even when care is taken to use similar tissue sampling and dissociation procedures. This original concern was not sufficiently addressed in the revision. The authors state that the “majority” of samples were collected by the same ENT clinicians. How big of a majority? Are the identities of the collecting clinicians documented in the metadata and examined for variability? The biases that are intrinsic to the method of scRNA-seq cannot be resolved by increasing sample size for scRNA-seq. In Figure 2e, for the molecular types that are claimed to be statistically overrepresented, can validation be added by an orthogonal technology (e.g., cell counts of stained tissue sections)?

4. In Figure 2f, the genes that constitute IFN α , IFN γ , and TNF α signatures should be detailed. It is important to document the

It is common practice to infer cellular abundance from single cell data, and some recent COVID-19 papers analysing nasal/nasopharyngeal swab samples (Ziegler *et al* Cell 2021, Chua *et al* Nat Biotech 2020, Loske *et al* Nat Biotech 2021) use this method to delineate the difference between healthy and COVID-19.

To minimise the technical variation in sampling, a detailed protocol for nasal and tracheal brushings has been used. The majority (42/50, 84%) of the healthy paediatric and COVID-19 nasal brushes (those collected at GOSH) and all of the tracheal/bronchial brushes were performed by the same experienced ENT clinician. As suggested by the referee, we have examined sample variability with regard to cell viability and number and have not identified a significant difference when looking across sites:

Figure R2.3

We do not have stored tissue sections from these patients as we were not able to do nasal or tracheal biopsies (especially in severe, ventilated children - many of whom were extremely unwell and anticoagulated, making a research biopsy a very risky procedure and parental consent almost impossible; brushings have a much lower risk of bleeding compared to biopsies) and therefore the requested work is simply not possible. Especially when samples are taken from young children, the entirety of the sample is required for single cell sequencing and it was not possible to store additional material.

We have now included the list of interferon stimulated genes that contributed to the analysis in the methods section or have cited appropriate gene lists.

nature of the characterized IFN responses especially because an imbalanced/inappropriate interferon response has been proposed (Blanco-melo Cell 2020) to drive development of COVID.

In the paper by Blanco-Melo *et al* Cell 2020 a different gene set was analysed. However, we expect that all interferon genes respond similarly. We show below (**Figure R2.4**) the gene expression of the genes highlighted in this publication in the airways of children and adults. This list also includes a number of cytokine genes.

For the interferon genes, the analysis largely replicates what we have found. Upon infection, there is a stronger ISG activation in epithelial cells in adults, whilst in airway immune cells the response is stronger in children. The analysis also replicated the pre-activated ISG signature in the airway immune cells in children. The cytokine genes in this list are expressed at lower levels, but mostly follow the patterns of expression that we observe in the ISG response and we do not observe an “imbalance” between the ISG and cytokine genes here. Given the much stronger ISG response, we have not commented separately on the cytokine response in our manuscript.

Figure R2.4

5. The revised manuscript now notes the discovery of two novel inflammatory epithelial transit cell types (Transit Epi 1 and Transit Epi 2), instead of just one (IETCs) in the original submission. Proximal airway epithelial cells are known to be plastic at least in mice (Rock et al PNAS 2009) and exhibit complex developmental trajectories (Plasschaert et al Nature 2018, Montoro et al Nature 2018). Could these newly identified cell populations be developmental intermediates rather than stable cell types? Describing them as novel cell types should probably be de-emphasized, and the alternative possibility of a developmental trajectory considered or added.	We absolutely agree with the referee's comment that these newly identified cell populations are likely to be developmental intermediates rather than cell types, as suggested by our own developmental trajectory analysis and by the literature. This is why we had already named them "transit" cells to emphasise that these are transitory cells between secretory and ciliated cell types rather than stable cell types. As suggested, we have de-emphasised this throughout the manuscript (including the abstract) by referring to them as novel cell states rather than cell types. Many thanks for bringing this to our attention.
6. Related to the prior point, on histology in Figure 2k, the double positive staining of S100A9/EPCAM is technically improved but sections should be co-stained with known proximal airway markers since mucous cells and basal cells can express S100A9, to further support that the populations are truly novel.	In Figure 2g, we show clear double positive EPCAM+S100A9+ staining in cells located next to the lumen, suggesting that these are surface epithelial cells of the airways. The nasal cavity is largely lined by pseudostratified columnar epithelium, interspersed with mucus-secreting goblet cells, typically within the apical epithelium. As we were able to detect multiple, adjacent S100A9/EPCAM positive cells on histology, in which cells can be seen located next to the lumen, we can predict with some certainty these are not all likely to be goblet cells. In Extended Data Figure 3b we show that S100A9 is expressed in a range of cell types and states, including goblet, hillock, squamous, duct and other cell types. Basal cells express only low levels of S100A9 (see violin plot below, Figure R2.6) and will be located at the base of the epithelium, not next to the lumen where we detect staining. Thus, our main finding of EPCAM+ epithelial cells co-staining with S100A9 is in line with our scRNAseq data and this result stands firmly even without any co-staining with known proximal airway markers.

	 Figure R2.6 Lastly, we regret to say that we no longer have any tissue available for further staining and further collection would not be possible in a realistic time frame, especially considering that the collection of research biopsies is a very risky procedure in acutely unwell, usually anticoagulated COVID-19 patients.
Referee 3 1. What is the basis for the increased activation/pre-activation of IFN signaling in COVID19 esp pediatric disease? Is this due to increased local ligand production, increased response to ligand, or relief of transcriptional feedback resulting in increased gene expression output in response to the same stimulus?	Authors' Response Our discovery of type I and III interferon production in nasal resident dendritic cells shows that we are technically able to quantify interferon ligand production. Nevertheless, we find a strikingly low / absent local production of these ligands, as visualised in detail in the following Figure:

Figure R3.1

This is in line with earlier studies that investigated the cellular response in the airways of COVID-19 patients and that detected no or very rare type I/III interferon producing cells, with no expression reported in epithelial cells [Ziegler *et al* Cell 2021, Loske *et al* 2021]. Because the local IFN production is

	nearly completely absent, it is unlikely that differences in IFN production in the airway can explain the strong differences in activation of IFN signalling in children. While the referee proposes interesting mechanisms, the cause and effect are very difficult to distinguish when analysing snapshot data. Firstly, interferon signalling acts as a positive feed forward loop, where many of the activators and receptors of interferon signalling are also upregulated upon interferon stimulation (see http://www.interferome.org/ for examples). While we do for example observe that children have higher expression of viral sensor / interferon activator DDX58, we do not want to make claims about this as we cannot distinguish if higher DDX58 is the cause of the increased interferon activation, or a consequence of it. The second issue which also relates to transcriptional feedback, is that SARS-CoV-2 is known to very efficiently suppress interferon signalling (e.g. the local SARS-CoV-2 IFN response is much lower than that induced by influenza) [Cao et al 2020]. Therefore, we would likely find higher interferon target gene expression in children, purely because we and others have shown that viral infection is more efficiently contained in children, which limits the suppressive effects of the virus on interferon signalling. This again makes it difficult to distinguish the cause and consequence of more effective interferon target gene expression. Altogether, while we agree that our results open up extremely interesting new research avenues such as finding the mechanistic basis for interferon (pre)activation, we believe that dissecting the molecular mechanisms that underpin this phenomenon cannot be achieved with the data we have generated. Such an analysis would require substantial in vitro models or human challenge experiments which are outside the scope of the present study.
2. A lot of the paper is interesting but descriptive WRT which cell types are altered in adult and pediatric COVID-19. What would be important is whether the authors could delineate key immune cell types/biomarkers which could be used to diagnose, follow and risk stratify patients using a clinically accessible test. Which things should clinicians at the bedside measure in their patients based on this elegant science?	The aim of our work was indeed to provide the basis for the development of novel clinical applications in our joint efforts to fight this disease. In terms of risk stratification, our data suggests that identifying a pre-activated IFN state in newly diagnosed SARS-CoV-2 infected adults (by taking a nasal sample and identifying increased IFN levels, akin to the simplicity of taking a lateral flow test), might be able to risk stratify these patients into a “low risk of severe disease” group, considering that they will likely be able to fight the virus efficiently at the site of infection, hence preventing systemic spread and immune related damage - similar to what we have shown in children compared to adults. Similarly, the absence of a pre-activated IFN state might put patients into a “high risk of severe disease” group. Current clinical options for newly infected,

	seronegative patients include pre-emptive monoclonal antibody therapy (Mahase BMJ 2021; doi: https://doi.org/10.1136/bmj.n2083) to reduce the risk of progression to severe disease. Hence, it is conceivable that IFN response status measurement could refine risk groups further to direct this and other future expensive and limited therapies to greatest benefit. More specifically, there is emerging data demonstrating a potential for therapeutic benefit of inhaled interferon 1-beta in adults (Monk et al, Lancet Respir Med 2021; https://pubmed.ncbi.nlm.nih.gov/33189161/) but this currently lacks an appropriately stratified target population. Identification of those with absent IFN pre-activation could identify those more likely to benefit from such a therapy early after infection (Peiffer-Smadja et al, Lancet Respir Med 2021; https://www.ncbi.nlm.nih.gov/pmc/articles/PMC7833737/), potentially before the development of symptomatic disease following contact tracing or earlier in the course of symptomatic disease. We focused our answer on adults who are at much greater risk than children, but children without a pre-activated IFN state are also likely to be at an increased risk of severe disease. However, the risk of severe disease is already so low in children, even in most of those with other known risk factors for severe disease, that we do not think there would be a practical application for this in children. We have added a sentence to illustrate potential clinical applications arising from our work in the discussion and thank the referee for raising this point.
3. Did any of the patients in this cohort have COVID-19 variants and if so did any of the immune system changes differ in delta or other variant contexts?	We agree that this is an important question considering the appearance of new variants throughout the pandemic so far. We believe that we have already adequately addressed this in our initial rebuttal, where we mapped a timeline showing the frequency of SARS-CoV-2 variants recorded in the UK and the USA at the time of SARS-CoV-2 PCR testing of each patient included in our COVID-19 cohort (paediatric samples are shown in red and adult ones in purple):

Figure R3.3: The SARS-CoV-2 variant frequency panel, was generated by GISAID and taken from their website using the following parameters; UK Dataset: ncov, gisaid, Europe. Filtered on United Kingdom. USA Dataset: ncov, gisaid, North America. Coloured by Clade (*Source: GISAID - NextStrain. (n.d.). Retrieved July 12, 2021, from <https://www.gisaid.org/phylogenetics/global/nextstrain>*).

Although a slight difference in the SARS-CoV-2 variant frequencies can be seen between the USA and UK around the time of sampling, the majority of the samples were collected when 20B and 20A, plus 20E (EU1) in the UK, were the dominant SARS-CoV-2 variants sequenced. We have now added a sentence to

	the methods section, stating: Given the timing of sample collections the viral strains in our cohort are most likely represent early viral variants (20B, 20A, 20E (EU1)).
4. The peripheral blood analysis in particular is very descriptive and not novel compared to other work in this space. The authors should present how their multi-omic analysis leads to new insights, and not just confirms what others have seen with less expensive/intensive approaches.	The novelty and focus of our work lies in the comparison between COVID-19 in paediatric and adult patients. Comparing the composition of the immune compartment in blood in Figure 3b reveals that the abundance of more than a dozen different immune cell types changes differently when children get COVID-19 compared to adults. To our knowledge, none of these changes have been reported before. Such broad differences can contribute to the difference in disease outcome between children and adults in multiple ways, making these observations very valuable for both the COVID-19 scientific community and society. Furthermore, in Figure 3d-f we functionally dissect the identity of eight immune cell types into interferon-stimulated cell states that we found to be present in all major immune cell compartments. We found these stimulated subpopulations to be present alongside unstimulated cells of the same cell type within the same patient. These subpopulations associate stronger with COVID-19 than any other previously known blood cell type (Figure 3f). Importantly, while others have reported global upregulation of IFN signaling before using differential expression analyses, this response has to our knowledge not been annotated into multiple defined blood subpopulations. These novel cell states therefore give a much more detailed and quantitative view of the immune cell response to COVID-19 than reported before. This new insight is highly relevant to COVID-19 research as the interferon signaling pathway has been identified by us and many others to be the key defense mechanism against infection, and the most important modulator of disease severity. Finally, our study generated a number of additional novel insights due to the unique comparison we perform between the cellular response in the airway and blood. Comparing the cell type proportion dynamics in the blood with the airway reveals a large disconnect between the immune response in the blood and at the site of infection (Extended Data Figure 7d-e), which underscores the importance of our approach to profile both sites in the same patient. Investigating these relationships in detail revealed that the interferon stimulated subpopulations that we discovered in the blood data, strongly correlate with the abundance of dendritic cells in the nose, which turn out to be the only interferon type I and III producing cells in our dataset. Again, we believe that this is an extremely valuable observation due to the relevance of interferon signaling in COVID-19 as described above, where to our knowledge, we are the first to identify interferon type I/III producing dendritic cells in COVID-19.

5. Can the authors show that local IFN production by key cell types is significantly increased in specific covid-19 contexts (statistically, not just in one outlier case).	We would like to stress that we are the first to identify dendritic cells with robust interferon type I/III ligand production in COVID-19 patients. While we emphasise in the main text that this is indeed limited to only the patient that was sampled at the earliest time point of disease onset, we do believe that this observation is very important, especially considering the role of interferon signaling in COVID-19 and the possible implications of discovering the producing cell type. The referee asks us to test the statistical significance of the observed IFN producing cell types and their association to COVID-19 contexts. First, our observation of type I and III IFN production in nasal resident dendritic cells is very significant as evidenced by the high expression and cell-type specificity shown in Figure R3.1, Figure 3h and Extended Data Figure 8. In other words, we have found cells with significant IFN ligand expression, and we are certain that this production is specific for nasal resident pDCs and cDCs. Second, we observe this cell-type specific and strong IFN production only in the patient that was sampled at the earliest time point of infection and we hypothesise that IFN production by DCs in the airway is temporally restricted to very early disease (i.e. a specific COVID-19 context). Because the time-since-infection metric is not normally distributed, it is statistically most appropriate to use a non-parametric rank-based significance test such as the Mann-Whitney U test to obtain the probability for an association between time and presence of IFN producing DCs. Comparing the timing of this one patient to the 24 patients where we did not detect any IFN producing DCs yields a p-value = 0.000027, meaning that it is statistically extremely unlikely that we would observe this by chance even with only one positive individual. Nevertheless, we prefer not to include these p-values in our manuscript to strengthen our conclusions, as some readers might overinterpret a significant statistical test into proof for causality, which in our view would require independent validation.
6. The introduction is quite long and a lot of it is not essential.	We agree and have shortened the introduction to half the initial length.
7. What do the authors make of the low recovery rate of viral RNA reads in covid-19 positive cases (36%). That seems surprisingly low.	In order to provide a more complex and comprehensive analysis of the adaptive immune response as well as a comparison in the T and B cell immune repertoires in children versus adults we decided to process our samples using 5' single cell 10X Chromium Next GEM technology instead of 3' technology. The two assays are similar, differing in location of the polydT sequence location (found on the gel bead in 3' assays and supplied as a RT primer in the 5' assay). A template switching oligo (TSO) is used in both workflows to reverse transcribe the full-length transcript. Whilst the 3' assay generates sequences close to gene transcription termination and polyadenylation sites, which may reveal 3' UTRs, their genomic location and alternative terminal sequences the 5' assay allows generation of reads at or close to the

transcription start site (TSS) of each gene. This provides information regarding the TSS location and alternative promoter usage making it a more valuable assay for studying promoters, transcription start sites, splice variants etc. and therefore preferable for studying the SARS-CoV-2 immune responses in children and adults within our study.

Whilst we were still able to detect low levels of SARS-CoV-2 virals reads within 10 of our COVID-19 positive patients (where ≥ 10 reads were detected), in line with other studies (Chua *et al* 2020 and Ziegler *et al* 2021), we found that recovery rate was lower than in other similar studies. We speculate that the use of **5' versus 3' technology**, the latter of which most other studies have used, may be responsible for the slightly lower sensitivity of detection. Since the genome of SARS-CoV-2 is a single and positive RNA strand, genes close to the 3' end of the genome are expected to have a higher detection rate in the presence of subgenomic transcripts. To our knowledge there are no studies where both methods have been used on the same samples and the detection rates compared. However, Ren X *et al* Cell 2021 used both 3' and 5' scRNA-seq SARS-CoV-2 datasets and reported a 3' enriched detection pattern in viral positive cells with both. This is in agreement with what we have observed and the known nested transcription process of coronaviruses, described by Masters *et al* 2006, where all genomic and subgenomic RNA molecules share the same 3' end.

Other factors which are known to affect the detection of viral reads include the **a)** site of sampling, **b)** time since onset of infection, **c)** disease severity and **d)** the way in which the samples are processed; such as experimental protocols including the version of 10X Next GEM chromium technology and bioinformatic parameters used in analysis.

- **Site of sampling:** Whilst the majority of other COVID-19 studies looking at the upper airways have sampled using nasopharyngeal swabs (Chua *et al* 2020 and Ziegler *et al* 2021) we sampled from a different region of the nose (inferior nasal concha) with a cytological brush which provided us with better cell capture than nasopharyngeal swabs. To our knowledge no SARS-CoV-2 viral transcripts have been detected or analysed in the nasal samples of children by scRNA-seq (Loske *et al* 2021 or Winkely *et al* 2021).
- **Time since onset of infection:** In line with other studies, where the majority of viral transcripts was detected in the early phase of infection (e.g. within the first 11 days of symptom onset (Chua *et al* 2020)), few viral reads were detected beyond 14 days of infection, with an average

	infection collection interval of those in which ≥ 10 SARS-CoV-2 reads were detected of 7.1 days ± 1.38 (mean \pm SEM). - Disease severity: In the 10 COVID-19 positive donors, which includes patients with a range of COVID-19 severities from asymptomatic to severe and adults and children, we are able to detect an average of 28 ± 6.3 UMI per million (geometric mean \pm sem) pre-filter and 25 ± 7.6 post-filter. These numbers are slightly lower, but comparable to both those reported in the Chua et al 2020 and Ziegler et al 2021, whilst including patients across a wider range of severities (including asymptomatic) as well as paediatric COVID-19 patients.- Sampling processing: Furthermore, distinct 10x chromium technology can affect viral read recovery. As we first started this project back in March 2020 all our COVID-19 samples were processed using the V1.1 5' Next GEM 10x chromium technology. Since then the company has released a newer version (V2 assay) which was shown to be more sensitive compared to the V1.1. The technology used in Next GEM Single Cell 3' v3.1 and the latest version of Next GEM Single Cell 5' (v2) are more similar in sensitivity and mapping rates.
--	---

Reviewer Reports on the Second Revision:

Referee #2 (Remarks to the Author):

The revised manuscript is significantly improved and many of the points raised in the previous review are well addressed in this revision. There are certainly major logistical and physical challenges of obtaining more tissue for staining to confirm some of the scRNA-seq results, and agree with the authors that the current data and analysis is extensive and provides a highly valuable resource for the research and medical community.

Referee #3 (Remarks to the Author):

No further comments